# Overtrained Language Models Are Harder to Fine-Tune

**Jacob Mitchell Springer** [1]  **Sachin Goyal** [1]  **Kaiyue Wen** [2]  **Tanishq Kumar** [3]  **Xiang Yue** [1]
**Sadhika Malladi** [4]  **Graham Neubig** [1]  **Aditi Raghunathan** [1]

## Abstract

Large language models are pre-trained on ever-growing token budgets under the assumption that better pre-training performance translates to improved downstream models. In this work, we challenge this assumption and show that extended pre-training can make models harder to fine-tune, leading to degraded final performance. We term this phenomenon **catastrophic overtraining**. For example, the instruction-tuned OLMo-1B model pre-trained on 3T tokens leads to over 2% worse performance on multiple standard LLM benchmarks than its 2.3T token counterpart. Through controlled experiments and theoretical analysis, we show that catastrophic overtraining arises from a systematic increase in the broad sensitivity of pre-trained parameters to modifications, including but not limited to fine-tuning. Our findings call for a critical reassessment of pre-training design that considers the downstream adaptability of the model.

## 1. Introduction

Language models have achieved widespread success following a two-stage paradigm: (1) pre-training on a vast corpus of uncurated data, followed by (2) post-training on high-quality task-specific data, often to confer targeted abilities such as instruction-following, multi-modality, or reasoning. Under the maxim *"more data is better"*, there have been massive investments in scaling both pre-training and post-training.

Hoffmann et al. (2022) proposed a compute-optimal ratio of roughly 20 tokens per model parameter, yet recent models have far exceeded this. For example, Llama-2-7B (Touvron et al., 2023) was trained on 1.8T tokens—13× the recom-

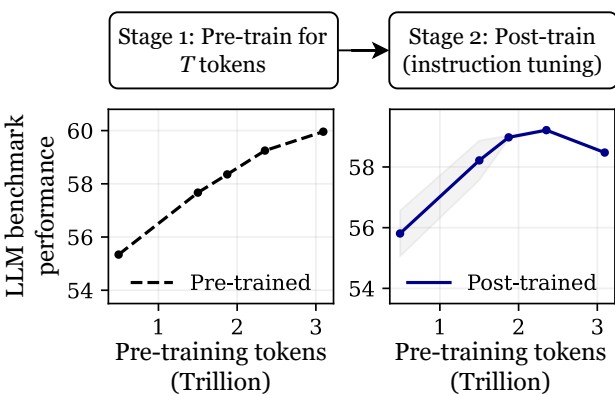

*Figure 1.* **Language models with extensive pre-training can exhibit *catastrophic overtraining*, where the performance of post-trained models degrades as the pre-training stage is extended.** We report the average performance of five common LLM benchmarks (ARC-Easy, ARC-Challenge, PIQA, HellaSwag) for OLMo-1B intermediate checkpoints before and after instruction fine-tuning, with additional results in Section 2. We argue that catastrophic overtraining arises as a result of a progressive increase throughout pre-training of model sensitivity to parameter transformations, leading to greater forgetting of the capabilities acquired during pre-training after fine-tuning (Section 3). Overall, our results challenge the notion that scaling pre-training is strictly beneficial.

mended ratio—and Llama-3-8B scaled this further to 15T tokens. This trend is driven by consistent gains in zero-shot performance (Gadre et al., 2024; Sardana et al., 2024), with few exceptions where scaling up is *not* helpful (Wei et al., 2022; McKenzie et al., 2022a;b; 2023).

In this paper, we demonstrate that the widely adopted strategy of **scaling up language model pre-training does not universally translate to better performance after post-training.** Through both theory and experiments, we uncover a phenomenon we term *catastrophic overtraining*, where longer pre-training harms final model performance after instruction tuning or other forms of post-training (Figure 1).

Catastrophic overtraining is not an isolated curiosity; rather it emerges consistently across a range of models and tasks. As shown in Section 2, extensive empirical evaluations demonstrate the prevalence of this phenomenon in exist-

[1]Carnegie Mellon University [2]Stanford University [3]Harvard University [4]Princeton University. Correspondence to: Jacob Mitchell Springer <jspringer@cmu.edu>.

*Proceedings of the 42nd International Conference on Machine Learning*, Vancouver, Canada. PMLR 267, 2025. Copyright 2025 by the author(s).

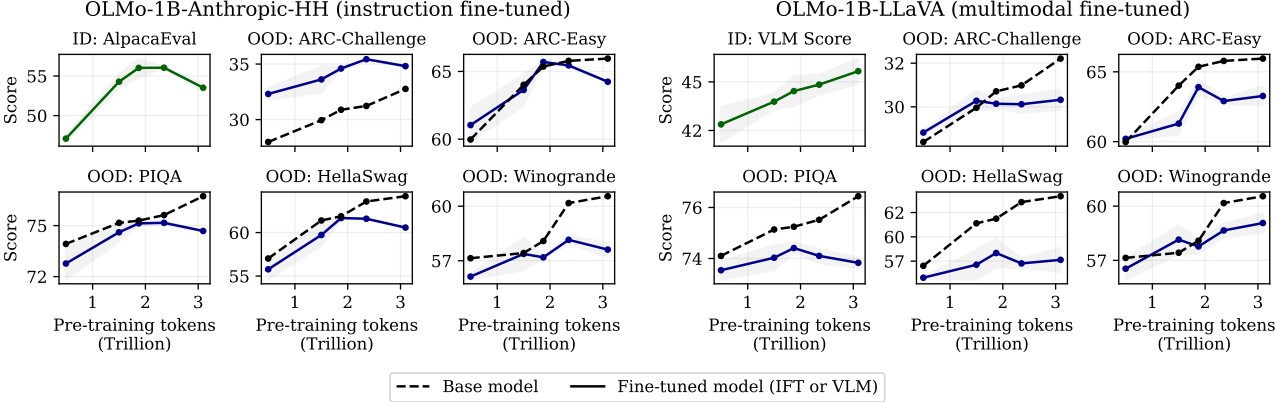

*Figure 2.* **Extending pre-training can degrade performance after fine-tuning on Anthropic-HH (left) and LLaVA (right).** We consider fine-tuning on various intermediate checkpoints from OLMo-1B pre-training. While the base model performance (before fine-tuning) improves with the pre-training token budget (black dashed curve), the performance after fine-tuning drops as we pre-train on more tokens. In the instruction-tuning setting (left), we observe degradation on the ID task (green)—AlpacaEval—as well as on OOD benchmarks (blue)—ARC, PIQA, and HellaSwag. In the multimodal tuning setting, we observe degradation with overtraining on PIQA, and a larger gap between the fine-tuned and base model for ARC, HellaSwag, and Winogrande. We report average over three independent fine-tuning runs, plus error bars. Refer to Appendix F for additional models (OLMo-2-7B, LLM360-Amber) and instruction-tuning datasets (extended results for Anthropic-HH, TULU).

ing models. For instance, we show that the OLMo-1B model (Groeneveld et al., 2024a), pre-trained on 3T tokens and post-trained on Anthropic-HH (Bai et al., 2022), performs 3% worse on AlpacaEval (Li et al., 2023b) and 2% worse on ARC (Clark et al., 2018) compared to an intermediate checkpoint trained on just 2.3T tokens (Figure 2).

To understand *why* catastrophic overtraining occurs, we turn to carefully controlled experiments (Section 3). We find that modifying the parameters of a pre-trained model leads to forgetting of previously acquired capabilities, where the extent of this forgetting depends on the magnitude of the parameter modifications. However, another key factor influencing forgetting is what we term progressive sensitivity: for modifications of equal magnitude, models that have undergone longer pre-training exhibit greater forgetting (Figure 4). Catastrophic overtraining arises when this increased forgetting due to post-training modifications overtakes the improvement during pre-training. While constraining the magnitude of the parameter modifications that arise from post-training can mitigate this degradation, it can also limit the pre-trained model's capacity to adapt and learn. This reveals an inherent trade-off that shapes the feasibility of preventing catastrophic overtraining in practice (Figure 7).

Finally, we present a theoretical analysis of a linear transfer learning setting in Section 4 that admits a precise characterization of catastrophic overtraining and progressive sensitivity. We study how *incremental feature learning* leads to progressive sensitivity and inevitable catastrophic overtrain-

ing. Regularization during fine-tuning can delay the onset, albeit at the cost of downstream performance.

## 2. Extended pre-training can hurt post-training

We study the effect of extended pre-training on two common post-training setups—instruction tuning for instruction following capability, and multimodal fine-tuning (visual instruction tuning) with LLaVA (Liu et al., 2023a).

### 2.1. Experimental setup

To analyze the effect of overtraining, we experiment on three language models with open-sourced intermediate checkpoints: OLMo-1B (Groeneveld et al., 2024a), OLMo-2-7B (OLMo et al., 2024), and LLM360-Amber-7B (Liu et al., 2023b). For each model, we perform post-training on intermediate checkpoints. We investigate instruction tuning with two datasets: Anthropic-HH (Bai et al., 2022) and TULU (Wang et al., 2023), and we perform multimodal fine-tuning with the LLaVA visual instruction tuning framework (Liu et al., 2023a). We train each intermediate checkpoint on each dataset.

We evaluate model performance along two key dimensions: the *ID performance*, evaluated on the fine-tuning task of interest (for e.g. instruction following), and the *OOD performance*, computed on a suite of ten common LLM evaluation benchmarks, covering reasoning, QA, commonsense, and

knowledge extraction. For each checkpoint, we tune the learning rate and select the model with the best ID performance.

We refer the reader to Appendix C for further information on the pre-trained models, the specification of the fine-tuning process, and for details of evaluation.

## 2.2. Results

Figure 2 compares the performance of various OLMo-1B models, trained to different pretraining budgets (x axis).

**Extended pre-training always improves base models.** In line with past work, we find that extended pre-training yields a monotonic improvement in the base models. The performance keeps improving on all the downstream tasks we evaluate (dashed line in Figure 2).

**Extended pre-training can hurt post-trained counterparts.** While the base model improves, we find a surprising degradation when the base models are post-trained. Specifically, after fine-tuning on the Anthropic-HH dataset for instruction following, a base model pre-trained on 3T tokens shows up to 3% lower response rate (AlpacaEval score) than one pre-trained on just 2.3T tokens ($\sim 23\%$ fewer tokens). We see a similar drop on various OOD tasks such as reasoning and question answering, as evaluated on benchmarks such as ARC-Easy, ARC-Challenge, HellaSwag, and PIQA. Overall, after instruction tuning, models pre-trained on 3T tokens underperform compared to those pre-trained on 2.3T tokens, dropping to the level of models pre-trained with just 1.5T tokens (50% fewer tokens).

For multimodal fine-tuning, we see that extended pre-training translates to continuous improvements in the VLM score. However, models pre-trained on more tokens show greater forgetting and larger drops in performance across the various OOD benchmarks. On some datasets such as PIQA, the drop is so severe that extended pre-training actively hurts performance after post-training (Figure 2, right).

We present evaluations of additional pre-trained models on different fine-tuning setups in Appendix F. Overall, while extended pre-training always improves the pre-training performance, these gains do not always translate to post-training. There are several settings where extended pre-training actively hurts post-training performance.

## 3. Catastrophic overtraining

In Section 2, we made a surprising observation where extended pre-training can hurt post-training. In this section, we dig deeper into this phenomenon to understand *why* and *when* expending more compute by pre-training on more tokens can counterintuitively degrade performance.

We begin by defining the phenomenon, which we call catastrophic overtraining.

> **Catastrophic overtraining** is the phenomenon where extending pre-training beyond a certain token budget results in a decrease in the model's performance after subsequent modifications.

We call this token budget where performance first begins to degrade the **inflection point**. Catastrophic overtraining can refer to a decrease of the pre-training performance or of the performance of other downstream tasks as pre-training is extended. Note that this performance drop can manifest differently across various downstream evaluation tasks, even for the same model.

In Section 2, we see catastrophic overtraining when post-training OLMo-1B for instruction tuning or multimodal fine-tuning and evaluating on standard benchmarks. In the rest of this paper, we aim to answer two central questions:

1. *When and why does catastrophic overtraining occur?*

2. *What factors influence the inflection point?*

To address these questions, we systematically study and build an intuitive picture of the effect of overtraining in the presence of Gaussian perturbations (Section 3.2) and then expand to fine-tuning in a controlled but real-world setup (Section 3.3).

### 3.1. Catastrophic overtraining in a controlled setup

We documented several instances of catastrophic overtraining in real-world scenarios. To gain a deeper understanding and explore more extreme degrees of overtraining, we investigate a simpler, controlled setup described below. Note that our real-world experiments used publicly available checkpoints from a single training run, which meant that each pre-training budget corresponded to a different final learning rate due to the annealing schedule. In this section, we remove that confounding factor.

**Pre-training setup.** We pre-train models from scratch with sizes ranging from 15M to 90M parameters, spanning token budgets from 4B to 128B, on C4 web data (Raffel et al., 2019). We train with a cosine annealing schedule that anneals every model to zero. In the main paper, we present results from the 30M model; see Appendix G for results with 15M and 90M parameter models.

**Modifications to the pre-trained model.** We fine-tune the pre-trained models above. We fine-tune each model on various classification and language modeling datasets spanning QA, sentiment analysis, math, and code. Details on the datasets and hyperparameter choices are provided in Appendix D. We also consider a simple modification of

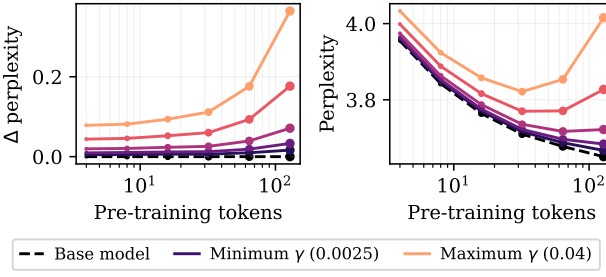

*Figure 3.* ***Progressive sensitivity of Gaussian perturbations (left):*** **extending pre-training progressively increases the degree to which a Gaussian parameter perturbation degrades perplexity.** ***Catastrophic overtraining (right):*** **eventually, this leads to overall worse pre-training perplexity.** We perturb OLMo-30M models trained on various pre-training token budgets with Gaussian noise scaled by the factor $\gamma$ (color). The left plot shows the difference in perplexity between the perturbed and unperturbed models, while the right plot shows the absolute perplexity of the perturbed models.

adding Gaussian perturbations to the pre-trained weights as a warm-up in Section 3.2.

Our intuitive picture views post-training as some modification to the pre-trained model that is trained on large amounts of broad data. Such modifications are aimed at improving some targeted performance (such as VLM score). However, as argued in (Kumar et al., 2022), such modifications can inadvertently distort the pre-trained knowledge, leading to degraded performance on out-of-distribution or unrelated tasks.

**Downstream evaluation.** While we evaluate real-world benchmarks in Section 2, we focus here on measuring the C4 perplexity of the modified downstream model as an indicator of how well the original pre-trained knowledge is preserved. A decline in C4 perplexity may signal a loss of this knowledge, potentially resulting in both out-of-distribution performance degradation (due to forgetting or distortion). We also measure ID performance as perplexity on held-out set from the same distribution as the fine-tuning data. We use perplexity rather than accuracy because it is a smoother and less noisy metric, and can often offer a better measure of model quality than accuracy for small models (Schaeffer et al., 2023; 2024). Although our analysis centers on pre-training perplexity, we acknowledge that other factors may also contribute to downstream performance losses—a topic we leave for future work.

### 3.2. Warmup: Gaussian perturbations

We take base models pre-trained to various token budgets and add Gaussian noise of the following form. Let $\theta \in \mathbb{R}^{\mathbf{d}}$

denote the base model weights, then we get

$$\tilde{\theta} = \theta + \epsilon \text{ where } \epsilon \sim \mathcal{N}(\mathbf{0}, \gamma^{\mathbf{2}}\Sigma), \qquad (1)$$

where $\Sigma$ is the covariance matrix of the initialization distribution of the parameters (prior to pre-training) and $\gamma$ controls the magnitude of the perturbation.

First, we plot the change in C4 perplexity due to Gaussian noise, i.e. the difference between the C4 perplexity of $\theta$ and $\tilde{\theta}$ in Figure 3 (left). We observe an interesting trend as we track the change in perplexity between the base model and the perturbed model as a function of the number of pre-training tokens:

> **Progressive sensitivity to noise:** For a fixed magnitude of perturbation, the change in perplexity between the base model and the perturbed model increases monotonically with the number of pre-training tokens.

Simultaneously, we plot the absolute C4 perplexity of the base model (Figure 3, right, dashed line). We observe that the base model's perplexity decreases with the number of pre-training tokens.

In this setting, **catastrophic overtraining arises from the interaction between the progressive sensitivity to noise and the monotonic improvement of the base model** as pre-training progresses. Early in training, the base model improves faster than the rate at which sensitivity increases, leading to a net decrease in perplexity after Gaussian parameter perturbations. Beyond a certain point, the rate at which sensitivity increases surpasses the rate at which the base model improves, leading to an increase in perplexity after the perturbation. This results in a **U-shaped trend** of the C4 perplexity after perturbation (Figure 3, right).

**Tracking the inflection point.** In Figure 3, larger perturbations are associated with a larger and more quickly increasing degradation of the pre-training loss. Thus, the point at which the degradation from sensitivity surpasses the improvement in the base model is accelerated for larger perturbations, leading to an earlier inflection point.

**Intuitive picture.** Pre-training on more tokens improves the base model (as expected) but also makes the base models more sensitive to noise. Progressive sensitivity leads to catastrophic overtraining as the increase in perplexity due to noise eventually overwhelms improvements in the model. For large magnitude perturbations, this degradation sets in at lower token budget, while for smaller magnitudes of perturbations, catastrophic overtraining may not be observed until a large token budget.

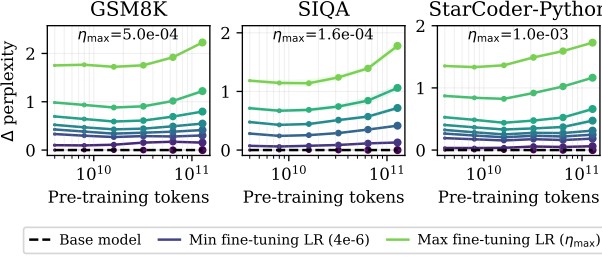

*Figure 4.* ***Progressive sensitivity of fine-tuning:*** **Extending pre-training progressively increases the degree to which fine-tuning degrades perplexity.** OLMo-30M models trained on various pre-training token budgets are fine-tuned on downstream tasks using fixed hyperparameters: math (GSM8k), code (Starcoder-Python), and QA (SIQA). Lines connect models sharing hyperparameters, differing only in pre-training tokens. Learning rates range from 4e-06 to the dataset-specific maximum ($\eta_{max}$). We report the difference in perplexity between the fine-tuned and pre-trained models, as a function of the number of pre-training tokens.

## 3.3. Fine-tuning pre-trained models

In the previous section, we studied how catastrophic over-training arises when adding noise to pre-trained models. While noise can be seen as a canonical modification, it is different from fine-tuning that might involve more structured updates to the models. However, we see in this section that the intuitive story above also holds when we fine-tune models on real-world language datasets described above.

### 3.3.1. FINE-TUNING WITH FIXED LEARNING RATE

First, analogous to how we quantify performance drop for a fixed magnitude of Gaussian perturbation ($\gamma$), we similarly need to regularize the fine-tuning in some way to ensure a consistent degree of change across the pre-trained checkpoints. Fixing the learning rate is a simple and effective way to do so. While we do not provide a formal justification, we discuss our reasoning in Appendix D.

For each learning rate, we plot the change in C4 perplexity from the pre-trained model to the fine-tuned model in Figure 4. In this plot, we track how the degradation in C4 perplexity evolves with the number of pre-training tokens. First, larger learning rates distort the model more and thus exhibit a greater increase in perplexity. Second, we observe a trend over pre-training tokens analogous to the behavior seen with Gaussian noise, but this time for fine-tuning.

> **Progressive sensitivity when fine-tuning:** For a fixed learning rate, the change in perplexity increases monotonically with the number of pre-training tokens.

At the inflection point at which sensitivity increases surpasses the rate at which the base model improves, we observe catastrophic overtraining. This results in a U-shaped trend of the C4 perplexity after fine-tuning (Figure 5, top).

**Tracking the inflection point for fine-tuning.** Analogous to the Gaussian setting, since the rate of increase of degradation is accelerated for larger learning rates, models trained with larger learning rates exhibit an inflection point at lower token budgets, and the degradation is more pronounced.

**ID perplexity.** While smaller learning rates generally result in less degradation to the C4 perplexity, the ID perplexity of the fine-tuned models shows a different trend: larger learning rates, up to a point, result in a lower ID perplexity, though sometimes also exhibit a U-shaped trend in ID perplexity (Figure 5, bottom). This implies that tuning the learning rate can sometimes mitigate degradation only at the cost of fine-tuning performance. We explore in Section 3.3.2 when tuning the learning rate to minimize the ID perplexity can mitigate the degradation of C4 perplexity that arises as pre-training is extended, and when it cannot.

**Intuitive picture.** The intuition from the Gaussian perturbation setting carries over to fine-tuning with a fixed learning rate. Pre-training on more tokens will improve the quality of the base model and at the same time make the model degrade more when fine-tuned. Beyond a certain point, pre-training on additional tokens will degrade the resulting fine-tuned model's C4 perplexity, and often the ID perplexity of the fine-tuning task.

### 3.3.2. BALANCING FINE-TUNING GAINS WITH DEGRADATION

In Section 3.3, we showed that for a fixed learning rate, the sensitivity of pre-trained models increases with the number of pre-training tokens, leading to catastrophic overtraining. In practice, however, the learning rate is tuned on a validation set from the in-domain (ID) task. This tuning process may yield different optimal learning rates across pre-trained checkpoints, which can potentially mitigate catastrophic overtraining. The degradation depends on both the learning rate as well as the sensitivity. So if a model pre-trained on more tokens can admit a smaller learning rate when fine-tuning to achieve good ID performance, it can compensate the increase in sensitivity.

However, this smaller rate does restrict the extent of necessary parameter updates, and might be insufficient to achieve good ID performance. This presents an interesting trade-off that we investigate empirically. We tune the learning rate to maximize fine-tuning ID performance. We track the optimal value as a function of the pre-training token budget, and plot the ID performance and pre-train perplexity corresponding

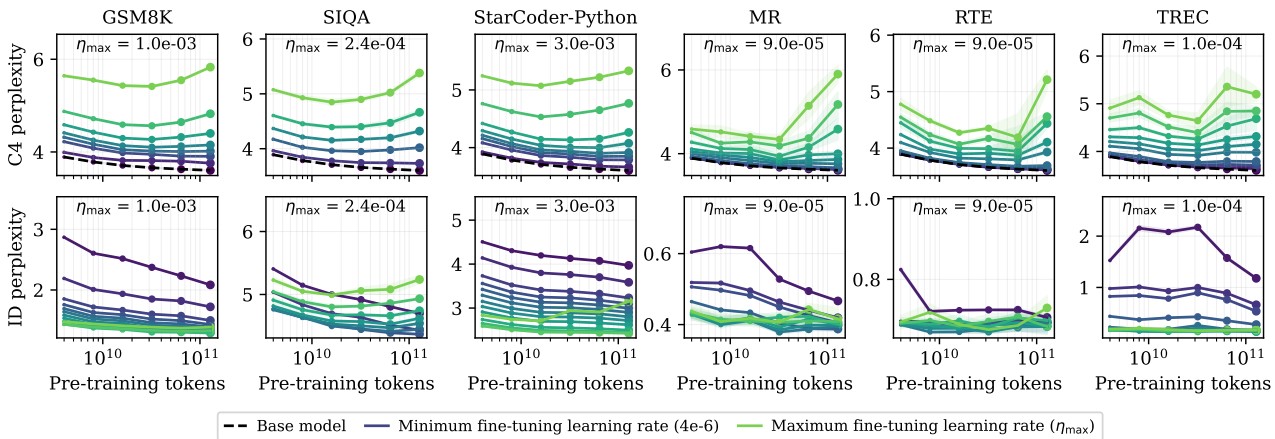

*Figure 5.* ***Catastrophic overtraining for fine-tuning with fixed hyperparameters:*** **extending pre-training can lead to an overall increase in the C4 perplexity (top), and ID perplexity (fine-tuning task; bottom), when fine-tuning with fixed hyperparameters.** OLMo-30M models pre-trained with varying token budgets are fine-tuned on downstream tasks using fixed hyperparameters: math (GSM8k), code (Starcoder-Python), QA (SIQA), and classification (MR, RTE, TREC). Lines connect models sharing hyperparameters, differing only in pre-training tokens. Learning rates range from 4e-06 to the dataset-specific maximum ($\eta_{max}$). At sufficiently large learning rates (lighter colors), we observe performance degradation in both ID and pre-training metrics beyond certain pre-training budgets. (See Appendices D and G for ablations.)

to this optimal learning rate in Figure 6.

Our findings indicate that the emergence of catastrophic overtraining depends on how the optimal learning rate evolves. We conceptualize this trade-off between ID performance and pre-train perplexity degradation into three scenarios, illustrated in Figure 7:

1. **Constant optimal learning rate:** A constant optimal learning rate across token budgets leads to degradation in both ID and out-of-domain (OOD) performance for large pre-training budget $T$ (Figure 7, left).

2. **Slowly decreasing optimal learning rate:** A slowly decreasing optimal learning rate may improve ID performance while OOD performance degrades (Figure 7, center).

3. **Quickly decreasing optimal learning rate:** A quickly decreasing optimal learning rate enables improvements in both ID and OOD performance as the pre-training budget increases (Figure 7, right).

**Using a non-optimal learning rate to mitigate degradation.** In cases where catastrophic overtraining emerges when fine-tuning with the optimal learning rate, using a non-optimal learning rate can sometimes mitigate the degradation or delay the inflection point. For example, in both cases where tuning leads to eventual degradation of the OOD loss in Figure 7, choosing to train with the smallest

learning rate would delay the inflection point. However, this would also result in a lower ID performance.

**Summary.** Overall, our experiments reveal that progressive sensitivity manifests under two types of modifications: unstructured Gaussian noise and structured fine-tuning, leading us to conjecture that *progressive sensitivity is a universal phenomenon*. With fixed perturbation magnitude or learning rate, this sensitivity causes catastrophic overtraining. In practice, tuning the learning rate introduces a trade-off: its evolution determines if extended pre-training results in performance degradation.

## 4. A theoretical perspective of overtraining

Catastrophic overtraining contradicts the common belief that more pre-training improves model quality. To investigate this, we analyze catastrophic overtraining for two-layer linear networks, focusing on identifying the *inflection point* (Definition 4.2)—the point after which more pre-training harms final performance on the pre-training task. We first examine catastrophic overtraining via Gaussian perturbations, paralleling our empirical results (Section 3.2), and then demonstrate progressive sensitivity during extended pre-training in a canonical fine-tuning scenario (Theorem 4.6). Next, we formalize how restricting the magnitude of the updates can alleviate performance degradation, using regularization rather than reduced learning rates as in earlier experiments (Section 3.3.2). Without regularization, catastrophic overtraining inevitably emerges (Theorem 4.7).

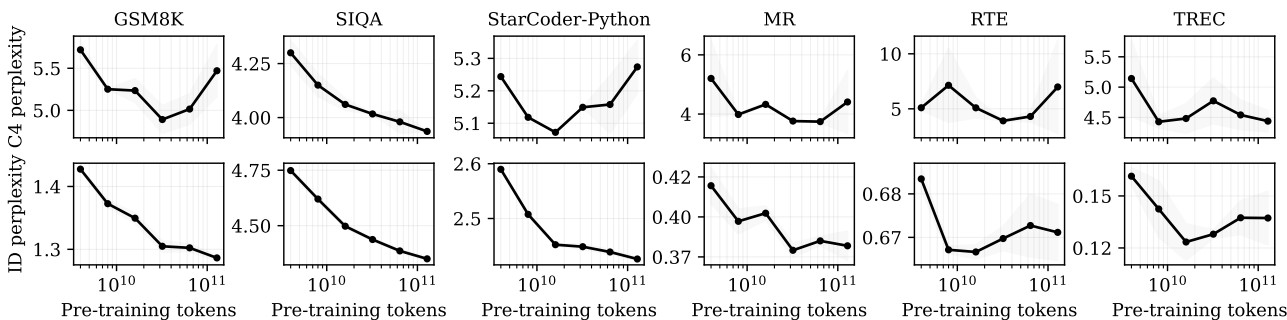

*Figure 6.* ***Catastrophic overtraining after hyperparameter tuning:*** **extending pre-training can lead to eventual degradation of the C4 perplexity (top) and ID perplexity (fine-tuning task; bottom), even after hyperparameter tuning.** OLMo-30M models pre-trained with varying token budgets are fine-tuned on downstream tasks: math (GSM8k), code (Starcoder-Python), QA (SIQA), and classification (MR, RTE, TREC). Lower is better. We tune the learning rate to optimize ID performance. ID perplexity degrades with extensive overtraining (RTE, TREC); C4 perplexity degrades in GSM8k, Starcoder-Python, MR, and RTE. Results averaged over three fine-tuning runs. (Additional ablations in Appendices D and G.)

While regularization can prevent this phenomenon, it can also impair fine-tuning performance by limiting adaptation (Theorem 4.7).

### 4.1. Pre-training setting

We adopt the two-layer linear regression setting proposed by Saxe et al. (2018) as a case where pre-training performance improves monotonically with training time via incremental feature learning. Precisely, we consider a regression problem where the data is generated by a full rank linear map $\boldsymbol{y} = \boldsymbol{A}^{\mathrm{pre}}\boldsymbol{x}$ for $\boldsymbol{x}, \boldsymbol{y} \in \mathbb{R}^d$, with $\boldsymbol{A}^{\mathrm{pre}} \in \mathbb{R}^{d \times d}$, and where we sample $\boldsymbol{x} \sim \mathcal{N}(0, \boldsymbol{I})$. Denote the SVD of $\boldsymbol{A}^{\mathrm{pre}}$ as $\boldsymbol{U}\boldsymbol{\Sigma}^{\mathrm{pre}}\boldsymbol{V}^T$, with the diagonal elements of $\boldsymbol{\Sigma}^{\mathrm{pre}}$ being strictly positive and monotonically decreasing. We will call these singular values the *pre-training features*, and denote them $\sigma_1^{\mathrm{pre}} > \cdots > \sigma_d^{\mathrm{pre}}$. Let $\boldsymbol{\Sigma}_{:i}^{\mathrm{pre}}$ be a diagonal matrix with the first $i$ singular values equal to those of $\boldsymbol{\Sigma}^{\mathrm{pre}}$ and the remaining set to 0.

We learn a two-layer network $\boldsymbol{\theta} = \boldsymbol{W}_1 \boldsymbol{W}_2$ with $\boldsymbol{W}_1, \boldsymbol{W}_2 \in \mathbb{R}^{d \times d}$ that minimizes the mean squared error $\mathcal{L}_{\mathrm{pre}}$ on the population of Gaussian inputs.

$$\mathcal{L}_{\mathrm{pre}}(\boldsymbol{\theta}(t)) = \|\boldsymbol{W}_1(t)\boldsymbol{W}_2(t) - \boldsymbol{A}^{\mathrm{pre}}\|_F^2.$$

We initialize $\boldsymbol{W}_1$ and $\boldsymbol{W}_2$ with small values and train using gradient flow. Prior work has established that, as training proceeds in this setting, the model $\boldsymbol{\theta}$ incrementally learns the spectrum of $\boldsymbol{A}^{\mathrm{pre}}$ (Saxe et al., 2018; Gidel et al., 2019).

**Theorem 4.1** (Informal statement of Saxe et al. (2018); Gidel et al. (2019)). *There exists a sequence of timesteps* $t_1 < \ldots < t_i < \ldots t_d$ *such that at timestep* $t_i$,

$$\boldsymbol{\theta}(t_i) \approx \boldsymbol{U}\boldsymbol{\Sigma}_{:i}^{\mathrm{pre}}\boldsymbol{V}^T.$$

This theorem implies that $\boldsymbol{\Sigma}(t) = \boldsymbol{U}^\top \boldsymbol{\theta}(t)\boldsymbol{V}$ is approximately diagonal, and the vector of its diagonal entries $\sigma(t)$ tracks which pre-training features have been learned by time $t$. In the ideal case, which we use in the main paper for brevity, we expect the first $n$ elements of $\sigma(t_n)$ are $\sigma_1^{\mathrm{pre}}, \ldots, \sigma_n^{\mathrm{pre}}$ and the remaining elements are zero.[1] Therefore, studying the evolution of $\sigma$ over time and its impact on the fine-tuning procedure allow us to characterize how elongating the pre-training period affects the pre-training and downstream performance of the final model. We will generally study progressive sensitivity and catastrophic overtraining by characterizing the model at time steps $t_1, ..., t_d$. We focus on studying the inflection point, the time at which catastrophic overtraining with respect to the pre-training loss emerges.

**Definition 4.2** (Inflection point). Fix a post-training modification to the model $\mathcal{A}$. The inflection point with respect to the pre-training loss is defined as the smallest $r$ such that $\mathcal{L}_{\mathrm{pre}}(\mathcal{A}(\boldsymbol{\theta}(t_r))) < \mathcal{L}_{\mathrm{pre}}(\mathcal{A}(\boldsymbol{\theta}(t_{r+1})))$.

In the following two sections, we study the inflection point for two different post-training modifications: Gaussian parameter perturbations and fine-tuning on a canonical family of tasks.

### 4.2. Gaussian perturbation setting

As a warm-up, we set $\mathcal{A}$ to be isotropic Gaussian parameter perturbations, mirroring Section 3.2. Formally, let $\mathcal{A}(\boldsymbol{\theta}(t_n)) = \widetilde{\boldsymbol{\theta}}(t_n) = (\boldsymbol{W}_1(t_n) + \boldsymbol{Z}_1)(\boldsymbol{W}_2(t_n) + \boldsymbol{Z}_2)$ where $\boldsymbol{Z}_1, \boldsymbol{Z}_2 \sim \mathcal{N}(0, \gamma^2 \mathrm{I}_{d^2 \times d^2})$, and let $\widetilde{\mathcal{L}}_{\mathrm{pre}}(t_n) = \mathbb{E}\left[\mathcal{L}_{\mathrm{pre}}(\widetilde{\boldsymbol{\theta}}(t_n))\right]$. We characterize how the perturbed model

---

[1] Appendix A contains the case when these coordinates are small but not exactly zero.

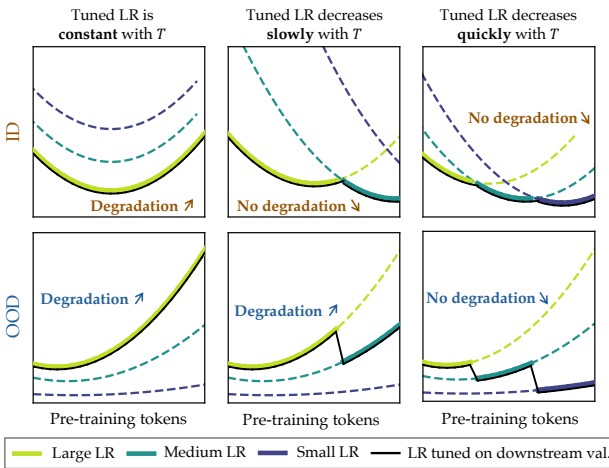

Tuned LR is **constant** with $T$ | Tuned LR decreases **slowly** with $T$ | Tuned LR decreases **quickly** with $T$

ID — Degradation ↗ | No degradation ↘ | No degradation ↘

OOD — Degradation ↗ | Degradation ↗ | No degradation ↘

Pre-training tokens | Pre-training tokens | Pre-training tokens

— Large LR  — Medium LR  — Small LR  — LR tuned on downstream val.

*Figure 7.* **Schematic to illustrate how the scaling of the optimal learning rate can affect model evaluations as a function of the pre-training tokens $T$.** The dashed lines indicate the hypothetical performance of a fixed learning rate, while solid lines indicate the performance when using the learning rate that optimizes the ID performance. **(Left)** When the optimal learning rate is constant, we expect to observe degradation of both ID and OOD performance. **(Center)** When the optimal learning rate decreases slowly with $T$, we may observe a degradation of only the OOD performance. **(Right)** When the optimal learning rate decreases quickly, we will not observe degradation of either metric of performance.

pre-training loss $\widetilde{\mathcal{L}}_{\mathrm{pre}}(t_n)$ evolves as pre-training is extended.

**Proposition 4.3** (Informal version of Lemma A.4). *Let $t_1, \ldots, t_d$ be defined as in Theorem 4.1. Then,*

$$\widetilde{\mathcal{L}}_{\mathrm{pre}}(t_n) - \widetilde{\mathcal{L}}_{\mathrm{pre}}(t_{n-1}) \geq (2d\gamma^2 - \sigma_n^{\mathrm{pre}})\sigma_n^{\mathrm{pre}}. \quad (2)$$

The formal proof in Appendix A demonstrates that elongating pre-training introduces a newly non-zero feature $\sigma_n$ introduces a new dimension along which the perturbation degrades loss. The above proposition allows us to characterize the inflection point (Definition 4.2) in the Gaussian perturbation setting as the smallest $n$ such that $2d\gamma^2 > \sigma_n^{\mathrm{pre}}$. As such, smaller or more quickly decaying features will induce a smaller inflection point.

To establish catastrophic overtraining, we now illustrate that degradation proceeds monotonically beyond the inflection point. That is, elongating the training budget beyond the inflection point will increasingly degrade the pre-training performance of the model.

**Theorem 4.4** (Informal version of Theorem A.3). *For some $\gamma > 0$, there exists an inflection point $r \in [1, d)$ such that $\widetilde{\mathcal{L}}_{\mathrm{pre}}(n)$ increases monotonically for $n \geq r$.*

Our results establish the inevitability of catastrophic over-

training with respect to the pre-training loss when the post-training modification consists of randomly perturbing the model parameters. In the next section, we study progressive sensitivity and catastrophic overtraining when fine-tuning on a family of canonical downstream tasks.

### 4.3. Fine-tuning

We now consider the case where the fine-tuning algorithm $\mathcal{A}$ corresponds to learning another linear feature map with a shared structure. We define the fine-tuning task as learning $\boldsymbol{y} = \boldsymbol{A}^{\mathrm{ft}}\boldsymbol{x}$, where $\boldsymbol{A}^{\mathrm{ft}} = \boldsymbol{U}\boldsymbol{\Sigma}^{\mathrm{ft}}\boldsymbol{V}^{\top}$. Sharing $\boldsymbol{U}$ and $\boldsymbol{V}$ with $\boldsymbol{A}^{\mathrm{pre}}$ permits transfer learning to occur, even though the spectrum of $\boldsymbol{A}^{\mathrm{ft}}$ is not the same as $\boldsymbol{A}^{\mathrm{pre}}$. We define the *fine-tuning features* $\sigma_1^{\mathrm{ft}} > \cdots > \sigma_d^{\mathrm{ft}}$ to be the singular values of $\boldsymbol{A}^{\mathrm{ft}}$.

Let $\mathcal{A}(\boldsymbol{\theta}(t)) = \boldsymbol{\theta}(t; k)$ denote a model pre-trained for time $t$ and then fine-tuned with a small but finite learning rate $\eta$ and a large batch size for $k \in [0, K]$ steps. The fine-tuning loss is similar to the pre-training loss with the new task $\boldsymbol{A}^{\mathrm{ft}}$, but we introduce a regularization term to limit the deviation from the pre-trained initialization. This regularization term is a standard design in meta learning literature (Chua et al., 2021; Denevi et al., 2018).

$$\mathcal{L}_{\mathrm{ft}}(\boldsymbol{\theta}(t; k); \lambda) = \mathbb{E}\big[\|\boldsymbol{\theta}(t; k) - \boldsymbol{A}^{\mathrm{ft}}\|_F^2 \\ + \lambda\|\boldsymbol{\theta}(t; k) - \boldsymbol{\theta}(t)\|_F^2\big]. \quad (3)$$

Analogous to the pre-training setting, our analysis proceeds by tracking the vector of the diagonal elements $\sigma^{\mathrm{ft}}(t; k)$ of $\boldsymbol{\Sigma}(t; k) = \boldsymbol{U}^{\top}\boldsymbol{\theta}(t; k)\boldsymbol{V}$. We define $\Delta_{\mathrm{pre}}(t_n) = \mathcal{L}_{\mathrm{pre}}(\boldsymbol{\theta}(t_n; K)) - \mathcal{L}_{\mathrm{pre}}(\boldsymbol{\theta}(t_n; 0))$ as the change in the pre-training performance over the course of fine-tuning, and we characterize how $\Delta_{\mathrm{pre}}(t_n)$ changes as the pre-training time $t_n$ increases. In particular, if $\Delta_{\mathrm{pre}}(t_n)$ is monotonically increasing, then we can conclude that *progressive sensitivity* is present.

To begin, we formalize the misalignment between the pre-training and downstream tasks in terms of their features.

**Definition 4.5.** The pre-training task $\boldsymbol{A}^{\mathrm{pre}}$ and the fine-tuning task $\boldsymbol{A}^{\mathrm{ft}}$ are $(\alpha, r)$-*misaligned* when $\sigma_i^{\mathrm{ft}} > \alpha\sigma_i^{\mathrm{pre}}$ for all $i > r$.

Our first result establishes that our setting exhibits progressive sensitivity when the fine-tuning task is different from the pre-training one.

**Theorem 4.6** (Progressive sensitivity; informal version of Theorem A.24). *Assume that $\boldsymbol{A}^{\mathrm{pre}}$ and $\boldsymbol{A}^{\mathrm{ft}}$ are $(\alpha, 1)$-misaligned with $\alpha > 1$. Then, $\Delta_{\mathrm{pre}}(t_n) \geq 0$ and $\Delta_{\mathrm{pre}}(t_n)$ is monotonically increasing with the number of learned pre-training features $n$.*

Having established the prevalence of progressive sensitivity, we now turn our attention to understanding how and when

we observe catastrophic overtraining with respect to the pre-training loss. We first show that when regularization is not present and the downstream task is sufficiently distinct from the pre-trained task, then elongating pre-training will pre-training performance. Furthermore, we demonstrate that regularization can delay the inflection point at which pre-training performance starts to degrade (Definition 4.2), albeit at a cost to the downstream performance.

**Theorem 4.7** (Catastrophic overtraining; informal version of Theorem A.25). *The following are true with high probability:*

1. ***Catastrophic overtraining is inevitable without regularization.*** *Let $\lambda = 0$. There exists an $\alpha_0 > 0$ such that if $A^{\mathrm{pre}}$ and $A^{\mathrm{ft}}$ are $(\alpha, r)$-misaligned, for $\alpha > \alpha_0$, then the pre-training loss after fine-tuning $\mathcal{L}_{\mathrm{pre}}(\boldsymbol{\theta}(t_n; K))$ monotonically increases for $n \geq r$.*

2. ***Regularization can delay the degradation of pre-training performance at the cost of downstream performance.*** *For any $n$, the inflection point $r(\lambda)$ and the unregularized fine-tuning loss $\|\boldsymbol{\theta}^n(K) - A^{\mathrm{ft}}\|_F^2$ increase monotonically with $\lambda$.*

Our results in this section demonstrate that progressive sensitivity and catastrophic overtraining can arise in the relatively simple setting of training linear networks, which learn task-related features incrementally. We characterize the inflection point (Definition 4.2) under various post-training modifications, including applying Gaussian perturbations and fine-tuning on a canonical task. Our main results demonstrate that elongating the pre-training period will inevitably result in progressive sensitivity and catastrophic overtraining, and although appropriate regularization can delay the onset of these phenomena, this may come at the cost of the downstream task performance (Theorems 4.4, 4.6 and 4.7).

## 5. Related Work

**Loss of plasticity.** The idea that more training can degrade a model's adaptability to new tasks, termed *loss of plasticity*, has been primarily studied in small-model continual learning (Ash & Adams, 2020; Dohare et al., 2021) and reinforcement learning (Kumar et al., 2020; Lyle et al., 2022; 2023; Ma et al., 2023; Abbas et al., 2023). Loss of plasticity has been attributed to loss curvature (Lyle et al., 2023; Lewandowski et al., 2023), increased weight norm (Nikishin et al., 2022), feature rank (Kumar et al., 2020; Gulcehre et al., 2022), and feature inactivity (Lyle et al., 2022; Dohare et al., 2021). Multiple remedies have been proposed, including architecture changes (Lyle et al., 2023), parameter resets (Nikishin et al., 2024; D'Oro et al., 2022), and regularization (Kumar et al., 2023; Ash & Adams, 2020).

**Catastrophic forgetting.** The phenomenon of *catastrophic forgetting*—where neural networks trained sequentially on tasks tend to forget prior tasks–has also been well-documented in the literature (Kirkpatrick et al., 2017; French, 1999; Goodfellow et al., 2013; Kemker et al., 2018; Kotha et al., 2023). There have been several proposed mitigation strategies, for example, Ahn et al. (2019); Hou et al. (2018); Chaudhry et al. (2019a) propose using regularization to mitigate catastrophic forgetting. Other fixes include generative replay of examples from previous tasks (Shin et al., 2017) or maintaining a memory buffer of previous tasks (Chaudhry et al., 2019b; de Masson d'Autume et al., 2019). In this work, we show that catastrophic forgetting can become more severe with overtraining.

**Scaling laws for optimal pre-training.** In our work, we argue that training for fewer tokens can be beneficial for downstream performance after fine-tuning. Related to our work, Isik et al. (2024) proposes scaling laws for certain downstream translation tasks after fine-tuning, but does not observe degradation with overtraining. In addition, optimal token budgets have been identified for fixed compute (Kaplan et al., 2020; Hoffmann et al., 2022), extended to various contexts (Hernandez et al., 2021; Cherti et al., 2023; Muennighoff et al., 2023; Goyal et al., 2024; Liu et al., 2025; Bhagia et al., 2024). Existing laws sometimes inaccurately predict downstream performance (Diaz & Madaio, 2024), and U-shaped scaling trends have been observed (Caballero et al., 2022; Wei et al., 2022; McKenzie et al., 2022a). Practitioners often overtrain small models beyond optimal tokens to reduce inference cost (Sardana et al., 2024; Gadre et al., 2024).

**Transfer learning theory.** Our theoretical analysis uses classical deep linear transfer learning setups (Gidel et al., 2019; Saxe et al., 2018). Such setups have been applied to knowledge acquisition (Wei et al., 2024; Arora et al., 2018), downstream feature benefits (Saunshi et al., 2021; Wei et al., 2021; Shachaf et al., 2021; Chua et al., 2021; Wu et al., 2020; Tripuraneni et al., 2020), and fine-tuning-induced performance degradation (Kumar et al., 2022).

We discuss additional related work in Appendix B.

## 6. Discussion

In this work, we uncovered a surprising trend: contrary to common belief, longer pre-training does not always lead to better post-trained models. We have shown that this is a consequence of a broader underlying phenomenon where models become more sensitive to perturbations as they are pre-trained on more tokens. Our theoretical analysis implies that this degradation of adaptability is especially catastrophic when the pre-training and fine-tuning tasks are misaligned, and in such a case catastrophic overtraining may be inevitable, even if the fine-tuning process is regularized.

## Acknowledgments

This material is based upon work supported by the National Science Foundation Graduate Research Fellowship under Grant No. DGE2140739. Any opinion, findings, and conclusions or recommendations expressed in this material are those of the authors(s) and do not necessarily reflect the views of the National Science Foundation.

We gratefully acknowledge support from Apple, NSF and the AI2050 program at Schmidt Sciences (Grant #G2264481).

Xiang Yue was supported in part by a Carnegie Bosch Institute Fellowship.

The authors would like to thank the following individuals for their helpful feedback and discussions: Christina Baek, Tianyu Gao, Gaurav Ghosal, Suhas Kotha, Vaishnavh Nagarajan, Chen Wu, and Ziqian Zhong.

## Impact Statement

This paper presents work whose goal is to advance the field of Machine Learning. There are many potential societal consequences of our work, none which we feel must be specifically highlighted here.

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

# A. Omitted Proofs from Section 4

## A.1. Formal Definitions and Assumptions

We provide formal definitions and assumptions underlying the theoretical analysis in Section 4. Throughout the text, we will use a constant $\delta$ to express a small probability.

**Model Architecture** The model consists of a two-layer linear network parameterized by $\boldsymbol{\theta} = \boldsymbol{W}_1 \boldsymbol{W}_2$, where $\boldsymbol{W}_1, \boldsymbol{W}_2 \in \mathbb{R}^{d \times d}$. The network maps input $\boldsymbol{x} \in \mathbb{R}^d$ to output $\boldsymbol{y} = \boldsymbol{W}_1 \boldsymbol{W}_2 \boldsymbol{x} \in \mathbb{R}^d$.

**Pretraining Task** The pretraining data follows $\boldsymbol{y} = \boldsymbol{A}^{\mathrm{pre}} \boldsymbol{x}$ where $\boldsymbol{A}^{\mathrm{pre}} \in \mathbb{R}^{d \times d}$ is a matrix with singular value decomposition (SVD) $\boldsymbol{A}^{\mathrm{pre}} = \boldsymbol{U} \boldsymbol{\Sigma}^{\mathrm{pre}} \boldsymbol{V}^\top$. Here, $\boldsymbol{U}, \boldsymbol{V} \in \mathbb{R}^{d \times d}$ are orthogonal matrices, and $\boldsymbol{\Sigma}^{\mathrm{pre}} \in \mathbb{R}^{d \times d}$ is diagonal with positive entries $\{\boldsymbol{\Sigma}_i^{\mathrm{pre}}\}_{i=1}^d$ arranged in decreasing order. Inputs $\boldsymbol{x} \sim \mathcal{N}(\mathbf{0}, \boldsymbol{I}_d)$ are standard Gaussian.

**Pretraining Process** The model is trained via gradient flow on the population loss:

$$\mathcal{L}_{\mathrm{pre}}(\boldsymbol{\theta}) = \mathbb{E}_{\boldsymbol{x}} \left[ \|\boldsymbol{\theta}\boldsymbol{x} - \boldsymbol{A}^{\mathrm{pre}}\boldsymbol{x}\|_2^2 \right] = \|\boldsymbol{\theta} - \boldsymbol{A}^{\mathrm{pre}}\|_F^2, \tag{4}$$

with parameters initialized as $\boldsymbol{W}_1(0) = \boldsymbol{W}_2(0) = \exp(-\tau)\mathrm{I}$ with a large $\tau > 0$. The gradient flow dynamics follow:

$$\dot{\boldsymbol{W}}_1(t) = -2(\boldsymbol{\theta}(t) - \boldsymbol{A}^{\mathrm{pre}})\boldsymbol{W}_2(t)^\top \tag{5}$$

$$\dot{\boldsymbol{W}}_2(t) = -2\boldsymbol{W}_1(t)^\top(\boldsymbol{\theta}(t) - \boldsymbol{A}^{\mathrm{pre}}) \tag{6}$$

where $\boldsymbol{\theta}(t) = \boldsymbol{W}_1(t)\boldsymbol{W}_2(t)$.

This setup is inherited from Gidel et al. (2019), where the authors consider a more general setup with a rank-$R$ matrix $\boldsymbol{A}^{\mathrm{pre}}$ and show that the gradient flow dynamics converge to the optimal rank-$r$ approximation of $\boldsymbol{A}^{\mathrm{pre}}$ sequentially for $r = 1, \ldots, R$.

**Theorem A.1** (Theorem 1 of Gidel et al. (2019)). *Suppose $\boldsymbol{A}^{\mathrm{pre}}$ has rank $R$. There exists $t_1, \ldots t_R$ and constant $C > 0$ depending on $\boldsymbol{A}^{\mathrm{pre}}$, such that for $\boldsymbol{\theta}(t)$ following Equations (5) and (6),*

$$\|W_1(t_i) - U\left(\Sigma^{\mathrm{pre},i}\right)^{1/2}\|_{\mathrm{F}} \le \exp(-C\tau);$$

$$\|W_2(t_i) - \left(\Sigma^{\mathrm{pre},i}\right)^{1/2} V^T\|_{\mathrm{F}} \le \exp(-C\tau).$$

*where $\Sigma^{\mathrm{pre},i}$ shares the first $i$ diagonal elements as $\Sigma^{\mathrm{pre}}$ and the rest diagonal elements are 0.*

**Finetuning Task** The finetuning task follows $\boldsymbol{y} = \boldsymbol{A}^{\mathrm{ft}}\boldsymbol{x}$ where $\boldsymbol{A}^{\mathrm{ft}} = \boldsymbol{U}\boldsymbol{\Sigma}^{\mathrm{ft}}\boldsymbol{V}^\top$ shares the singular vectors of $\boldsymbol{A}^{\mathrm{pre}}$ but has a spectrum $\boldsymbol{\Sigma}^{\mathrm{ft}}$. The input distribution remains $\boldsymbol{x} \sim \mathcal{N}(\mathbf{0}, \boldsymbol{I}_d)$.

**Finetuning Process** Starting from $\boldsymbol{\theta}^n(0) = \boldsymbol{\theta}(t_n)$ in Theorem A.1, the model is fine-tuned using gradient descent with learning rate $\eta$, batch size $m$, and $K$ iterations. We will call $\boldsymbol{\theta}^n(0)$ the real initialization and denote the following initialization $\bar{\boldsymbol{\theta}}^n(0)$ as the ideal initialization,

$$\bar{\boldsymbol{W}}_1^n(0) = U\left(\Sigma^{\mathrm{pre},n}\right)^{1/2} \tag{7}$$

$$\bar{\boldsymbol{W}}_2^n(0) = \left(\Sigma^{\mathrm{pre},n}\right)^{1/2} V^T \tag{8}$$

The population loss is:

$$\mathcal{L}_{\mathrm{ft}}(\boldsymbol{\theta}) = \mathbb{E}_{\boldsymbol{x}} \left[ \|\boldsymbol{\theta}\boldsymbol{x} - \boldsymbol{A}^{\mathrm{ft}}\boldsymbol{x}\|_2^2 \right] + \lambda\|\boldsymbol{\theta} - \boldsymbol{\theta}^n(0)\|_F^2 = \|\boldsymbol{\theta} - \boldsymbol{A}^{\mathrm{ft}}\|_F^2 + \lambda\|\boldsymbol{\theta} - \boldsymbol{\theta}^n(0)\|_F^2 \tag{9}$$

We will estimate $\mathcal{L}_{\mathrm{ft}}$ using a batch of samples $B_k$ with size $m$ on every step,

$$\mathcal{L}_{\mathrm{ft}}(\boldsymbol{\theta}; B_k) = \frac{1}{m} \sum_{x \in B_k} \left[ \|\boldsymbol{\theta}\boldsymbol{x} - \boldsymbol{A}^{\mathrm{ft}}\boldsymbol{x}\|_2^2 \right] + +\lambda\|\boldsymbol{\theta} - \boldsymbol{\theta}^n(0)\|_F^2$$

Denote the covariance of $x$ in batch $B_k$ as

$$\Sigma_k^{(x)} = \frac{1}{m} \sum_{x \in B_k} xx^T.$$

Then,

$$\mathcal{L}_{\text{ft}}(\boldsymbol{\theta}; B_k) = \text{Tr}\left( (\boldsymbol{\theta} - \boldsymbol{A}^{\text{ft}})^T (\boldsymbol{\theta} - \boldsymbol{A}^{\text{ft}}) \Sigma_k^{(x)} \right) + \lambda \|\boldsymbol{\theta} - \boldsymbol{\theta}^n(0)\|_F^2.$$

The parameter update rule at step $k$ is:

$$\boldsymbol{W}_1^n(k+1) = \boldsymbol{W}_1^n(k) - 2\eta(\boldsymbol{\theta}^n(k) - \boldsymbol{A}^{\text{ft}})\Sigma_k^{(x)}(\boldsymbol{W}_2^n(k))^\top - 2\eta\lambda(\boldsymbol{\theta}^n(k) - \boldsymbol{\theta}^n(0))(\boldsymbol{W}_2^n(k))^\top \tag{10}$$

$$\boldsymbol{W}_2^n(k+1) = \boldsymbol{W}_2^n(k) - 2\eta(\boldsymbol{W}_1^n(k))^\top \Sigma_k^{(x)}(\boldsymbol{\theta}^n(k) - \boldsymbol{A}^{\text{ft}}) - 2\eta\lambda(\boldsymbol{W}_1^n(k))^\top(\boldsymbol{\theta}^n(k) - \boldsymbol{\theta}^n(0)) \tag{11}$$

where $\boldsymbol{\theta}^n(k) = \boldsymbol{W}_1^n(k)\boldsymbol{W}_2^n(k)$.

We will denote the final finetuned loss as $\mathcal{L}_{\text{ft}}(n) = \mathcal{L}_{\text{ft}}(\boldsymbol{\theta}^n(K))$.

We will use $\Gamma$ to denote the upper bound of $\boldsymbol{\Sigma}^{\text{pre}}$ and $\boldsymbol{\Sigma}^{\text{ft}}$ as,

$$\Gamma = \max\left\{ \boldsymbol{\Sigma}_{1,1}^{\text{pre}}, \max_{i \leq d} \boldsymbol{\Sigma}_{i,i}^{\text{ft}} \right\}. \tag{12}$$

### A.2. Formal Statement and Proof of Theorem 4.4

In this section, we consider perturbations of the weights with isotropic Gaussian noise. For a parameter $\boldsymbol{\theta} = \boldsymbol{W}_1\boldsymbol{W}_2$, we will consider perturbations of the form $(\boldsymbol{W}_1 + \alpha)(\boldsymbol{W}_2 + \beta)$ where $\alpha, \beta \in \mathbb{R}^{d \times d}$ are independent isotropic Gaussian noise matrices with $\alpha_{ij}, \beta_{ij} \sim \mathcal{N}(0, \gamma^2)$ for some $\gamma > 0$. We will define the perturbed pretraining loss as,

$$\tilde{\mathcal{L}_{\text{pre}}}(\boldsymbol{\theta}) = \mathbb{E}_{\alpha, \beta \sim \mathcal{N}(0, \gamma^2)} \left[ \|(\boldsymbol{W}_1 + \alpha)(\boldsymbol{W}_2 + \beta) - \boldsymbol{A}^{\text{pre}}\|_F^2 \right] \tag{13}$$

Under this definition, assuming pretraining initialization is sufficiently small, we have that the loss under a Gaussian perturbation is monotonically increasing.

**Assumption A.2** (Small Pretraining Initialization). $\tau$ satisfies that, for $C$ in Theorem A.1,

$$\exp(-C\tau) \leq \min\left\{ \boldsymbol{\Sigma}_{1,1}^{\text{pre}}/2, 1/4, \frac{(\boldsymbol{\Sigma}_{d,d}^{\text{pre}})^2}{16d\boldsymbol{\Sigma}_{1,1}^{\text{pre}}\left(2\boldsymbol{\Sigma}_{1,1}^{\text{pre}} + \gamma^2\right)} \right\}.$$

**Theorem A.3.** *Under Assumption A.2, if $\gamma^2 > \boldsymbol{\Sigma}_{d,d}^{\text{pre}}/d$, there exists some $s \in \mathbb{N}$ and $s < d$ such that for all $n > s$, the loss under a Gaussian perturbation $\tilde{\mathcal{L}_{\text{pre}}}(\boldsymbol{\theta}^n(0))$ is monotonically increasing.*

*Proof.* Choose $s$ as the minimum number satisfying $\gamma^2 > \boldsymbol{\Sigma}_{s,s}^{\text{pre}}/d$, for $n > s$, then $s \leq d - 1$, by Lemma A.4,

$$\tilde{\mathcal{L}_{\text{pre}}}(\bar{\boldsymbol{\theta}}^n(0)) - \tilde{\mathcal{L}_{\text{pre}}}(\bar{\boldsymbol{\theta}}^{n-1}(0)) > \left(\boldsymbol{\Sigma}_{n,n}^{\text{pre}}\right)^2.$$

By Lemma A.6,

$$\tilde{\mathcal{L}_{\text{pre}}}(\boldsymbol{\theta}^n(0)) - \tilde{\mathcal{L}_{\text{pre}}}(\bar{\boldsymbol{\theta}}^n(0)) > - \left(\boldsymbol{\Sigma}_{n,n}^{\text{pre}}\right)^2/2.$$

$$\tilde{\mathcal{L}_{\text{pre}}}(\bar{\boldsymbol{\theta}}^{n-1}(0)) - \tilde{\mathcal{L}_{\text{pre}}}(\boldsymbol{\theta}^{n-1}(0)) > - \left(\boldsymbol{\Sigma}_{n,n}^{\text{pre}}\right)^2/2.$$

Combining the above,

$$\tilde{\mathcal{L}_{\text{pre}}}(\boldsymbol{\theta}^n(0)) - \tilde{\mathcal{L}_{\text{pre}}}(\boldsymbol{\theta}^{n-1}(0)) > 0.$$

The proof is complete. □

**Lemma A.4.** *The following inequality holds for any $n > 1$:*

$$\tilde{\mathcal{L}}_{\mathrm{pre}}(\bar{\boldsymbol{\theta}}^n(0)) - \tilde{\mathcal{L}}_{\mathrm{pre}}(\bar{\boldsymbol{\theta}}^{n-1}(0)) \geq (2d\gamma^2 - \boldsymbol{\Sigma}_{n,n}^{\mathrm{pre}})\boldsymbol{\Sigma}_{n,n}^{\mathrm{pre}} \tag{14}$$

*Proof.* We first expand the loss,

$$
\begin{aligned}
\tilde{\mathcal{L}}_{\mathrm{pre}}(\bar{\boldsymbol{\theta}}^n(0)) &= \mathbb{E}\left[\left\|(\bar{\boldsymbol{W}}_1^n + \alpha)(\bar{\boldsymbol{W}}_2^n + \beta) - \boldsymbol{A}^{\mathrm{pre}}\right\|_F^2\right] \\
&= \mathbb{E}\left[\left\|\left(U\left(\Sigma^{\mathrm{pre},n}\right)^{1/2} + \alpha\right)\left((\Sigma^{\mathrm{pre},n})^{1/2}V^T + \beta\right) - U\boldsymbol{\Sigma}^{\mathrm{pre}}V^\top\right\|_F^2\right] \\
&= \mathbb{E}\left[\left\|U\left((\Sigma^{\mathrm{pre},n})^{1/2} + \alpha\right)\left((\Sigma^{\mathrm{pre},n})^{1/2} + \beta\right)V^\top - U\boldsymbol{\Sigma}^{\mathrm{pre}}V^\top\right\|_F^2\right] \\
&= \mathbb{E}\left[\left\|\left((\Sigma^{\mathrm{pre},n})^{1/2} + \alpha\right)\left((\Sigma^{\mathrm{pre},n})^{1/2} + \beta\right) - \boldsymbol{\Sigma}^{\mathrm{pre}}\right\|_F^2\right] \\
&= \mathbb{E}\left[\left\|\left(\Sigma^{\mathrm{pre},n} + \alpha\left(\Sigma^{\mathrm{pre},n}\right)^{1/2} + (\Sigma^{\mathrm{pre},n})^{1/2}\beta + \alpha\beta - \boldsymbol{\Sigma}^{\mathrm{pre}}\right)\right\|_F^2\right] \\
&= \|\Sigma^{\mathrm{pre},n} - \boldsymbol{\Sigma}^{\mathrm{pre}}\|_F^2 + \mathbb{E}\left[\left\|\alpha\left(\Sigma^{\mathrm{pre},n}\right)^{1/2}\right\|_F^2\right] + \mathbb{E}\left[\left\|(\Sigma^{\mathrm{pre},n})^{1/2}\beta\right\|_F^2\right] + \mathbb{E}\left[\|\alpha\beta\|_F^2\right] \tag{15}
\end{aligned}
$$

where the fourth equality arises from the isotropy of the Gaussian noise, and the final equality comes from the independence and zero mean of the noise distributions.

**Lemma A.5.** *For Gaussian noise matrix $\alpha \in \mathbb{R}^{d \times d}$ where each entries has variance $\gamma^2$ and fixed matrix $M$, it holds that*

$$\mathbb{E}[\|\alpha M\|_F^2] = d\gamma^2 \|M\|_F^2.$$

*Proof.* It holds that

$$\mathbb{E}[\|\alpha M\|_F^2] = \mathbb{E}[\mathrm{Tr}(\alpha M M^T \alpha^T)] = \mathbb{E}[\mathrm{Tr}(\alpha\alpha^T)]\|M\|_F^2 = d\gamma^2\|M\|_F^2.$$

The proof is then completed. $\qquad\square$

By Lemma A.5 and Equation (15),

$$\tilde{\mathcal{L}}_{\mathrm{pre}}(\bar{\theta}^n) = \mathcal{L}_{\mathrm{pre}}(\bar{\theta}^n) + 2d\gamma^2\|(\Sigma^{\mathrm{pre},n})^{1/2}\|_F^2 + \mathbb{E}\left[\|\alpha\beta\|_F^2\right].$$

Taking difference with $\tilde{\mathcal{L}}_{\mathrm{pre}}(\bar{\theta}^{n-1})$

$$\tilde{\mathcal{L}}_{\mathrm{pre}}(\bar{\theta}^n) - \tilde{\mathcal{L}}_{\mathrm{pre}}(\bar{\theta}^{n-1}) = 2d\gamma^2\boldsymbol{\Sigma}_{n,n}^{\mathrm{pre}} - (\boldsymbol{\Sigma}_{n,n}^{\mathrm{pre}})^2.$$

$\qquad\square$

We then proceed to bound the difference between the perturbed loss of the ideal initialization and the perturbed loss of the real initialization when the pretraining initialization is sufficiently small.

**Lemma A.6.** *Under Assumption A.2, for any $n > 0$, it holds that*

$$\left|\tilde{\mathcal{L}}_{\mathrm{pre}}(\boldsymbol{\theta}^n(0)) - \tilde{\mathcal{L}}_{\mathrm{pre}}(\bar{\boldsymbol{\theta}}^n(0))\right| \leq (\boldsymbol{\Sigma}_{d,d}^{\mathrm{pre}})^2/2.$$

*Proof.* By the definition of $\tilde{\mathcal{L}}_{\mathrm{pre}}$,

$$
\begin{aligned}
\tilde{\mathcal{L}}_{\mathrm{pre}}(\boldsymbol{\theta}) &= \mathbb{E}_{\alpha,\beta\sim\mathcal{N}(0,\gamma^2)}\left[\|(\boldsymbol{W}_1 + \alpha)(\boldsymbol{W}_2 + \beta) - \boldsymbol{A}^{\mathrm{pre}}\|_F^2\right] \\
&= \mathbb{E}_{\alpha,\beta\sim\mathcal{N}(0,\gamma^2)}\left[\|(\boldsymbol{W}_1 + \alpha)(\boldsymbol{W}_2 + \beta) - \boldsymbol{A}^{\mathrm{pre}}\|_F^2\right] \\
&= \|\boldsymbol{W}_1\boldsymbol{W}_2 - \boldsymbol{A}^{\mathrm{pre}}\|_F^2 + \mathbb{E}[\|\alpha\beta\|_F^2] + \mathbb{E}[\|\boldsymbol{W}_1\beta\|_F^2] + \mathbb{E}[\|\alpha\boldsymbol{W}_2\|_F^2].
\end{aligned}
$$

By Lemma A.5,

$$\mathbb{E}[\|\boldsymbol{W}_1\boldsymbol{W}_2 - \boldsymbol{A}^{\mathrm{pre}}\|_F^2] = \|\bar{\boldsymbol{W}}_1\bar{\boldsymbol{W}}_2 - \boldsymbol{A}^{\mathrm{pre}}\|_F^2 + d\gamma^2\left(\|\boldsymbol{W}_1\|_F^2 + \|\boldsymbol{W}_2\|_F^2\right).$$

Taking the difference between $\tilde{\mathcal{L}_{\mathrm{pre}}}(\boldsymbol{\theta}^n(0))$ and $\tilde{\mathcal{L}_{\mathrm{pre}}}(\bar{\boldsymbol{\theta}}^n(0))$,

$$\begin{aligned}
\left|\tilde{\mathcal{L}_{\mathrm{pre}}}(\boldsymbol{\theta}^n(0)) - \tilde{\mathcal{L}_{\mathrm{pre}}}(\bar{\boldsymbol{\theta}}^n(0))\right| \leq &\left|\|\boldsymbol{W}_1\boldsymbol{W}_2 - \boldsymbol{A}^{\mathrm{pre}}\|_F^2 - \|\bar{\boldsymbol{W}}_1\bar{\boldsymbol{W}}_2 - \boldsymbol{A}^{\mathrm{pre}}\|_F^2\right| \\
&+ d\gamma^2\left|\|\boldsymbol{W}_1\|_F^2 - \|\bar{\boldsymbol{W}}_1\|_F^2\right| \\
&+ d\gamma^2\left|\|\boldsymbol{W}_2\|_F^2 - \|\bar{\boldsymbol{W}}_2\|_F^2\right|.
\end{aligned}$$

By Theorem A.1,

$$\begin{aligned}
\|\boldsymbol{W}_1 - \bar{\boldsymbol{W}}_1\|_F &\leq \exp(-C\tau); \\
\|\boldsymbol{W}_2 - \bar{\boldsymbol{W}}_2\|_F &\leq \exp(-C\tau).
\end{aligned}$$

Here $\exp(-C\tau) \leq \min\{\boldsymbol{\Sigma}_{1,1}^{\mathrm{pre}}/2, 1/4\}$.

Therefore,

$$\begin{aligned}
\left|\|\boldsymbol{W}_1\|_F^2 - \|\bar{\boldsymbol{W}}_1\|_F^2\right| &\leq \left|2\mathrm{Tr}((\bar{\boldsymbol{W}}_1)^T(\boldsymbol{W}_1 - \bar{\boldsymbol{W}}_1))\right| + \|\boldsymbol{W}_1 - \bar{\boldsymbol{W}}_1\|_F^2 \\
&\leq 2\exp(-C\tau)\boldsymbol{\Sigma}_{1,1}^{\mathrm{pre}} + \exp(-2C\tau) \leq 4\exp(-C\tau)\boldsymbol{\Sigma}_{1,1}^{\mathrm{pre}}.
\end{aligned}$$

Similarly,

$$\begin{aligned}
\left|\|\boldsymbol{W}_2\|_F^2 - \|\bar{\boldsymbol{W}}_2\|_F^2\right| &\leq \left|2\mathrm{Tr}((\bar{\boldsymbol{W}}_2)^T(\boldsymbol{W}_2 - \bar{\boldsymbol{W}}_2))\right| + \|\boldsymbol{W}_2 - \bar{\boldsymbol{W}}_2\|_F^2 \\
&\leq 2\exp(-C\tau)\boldsymbol{\Sigma}_{1,1}^{\mathrm{pre}} + \exp(-2C\tau) \leq 4\exp(-C\tau)\boldsymbol{\Sigma}_{1,1}^{\mathrm{pre}}.
\end{aligned}$$

Finally,

$$\begin{aligned}
&\left|\|\boldsymbol{W}_1\boldsymbol{W}_2 - \boldsymbol{A}^{\mathrm{pre}}\|_F^2 - \|\bar{\boldsymbol{W}}_1\bar{\boldsymbol{W}}_2 - \boldsymbol{A}^{\mathrm{pre}}\|_F^2\right| \\
&\leq \|(\boldsymbol{W}_1\boldsymbol{W}_2 - \bar{\boldsymbol{W}}_1\bar{\boldsymbol{W}}_2)\|_F\|\boldsymbol{W}_1\boldsymbol{W}_2 + \bar{\boldsymbol{W}}_1\bar{\boldsymbol{W}}_2 - 2\boldsymbol{A}^{\mathrm{pre}}\|_F.
\end{aligned}$$

Here

$$\begin{aligned}
\|(\boldsymbol{W}_1\boldsymbol{W}_2 - \bar{\boldsymbol{W}}_1\bar{\boldsymbol{W}}_2)\|_F &\leq \|\boldsymbol{W}_1 - \bar{\boldsymbol{W}}_1\|_F\|\boldsymbol{W}_2\|_F + \|\boldsymbol{W}_1\|_F\|\boldsymbol{W}_2 - \bar{\boldsymbol{W}}_2\|_F + \|\boldsymbol{W}_1 - \bar{\boldsymbol{W}}_1\|_F\|\boldsymbol{W}_2 - \bar{\boldsymbol{W}}_2\|_F \\
&\leq 2\sqrt{d}\exp(-C\tau)\boldsymbol{\Sigma}_{1,1}^{\mathrm{pre}} + \exp(-2C\tau) \leq 4\sqrt{d}\boldsymbol{\Sigma}_{1,1}^{\mathrm{pre}}\exp(-C\tau)
\end{aligned}$$

And

$$\begin{aligned}
\|\boldsymbol{W}_1\boldsymbol{W}_2 + \bar{\boldsymbol{W}}_1\bar{\boldsymbol{W}}_2 - 2\boldsymbol{A}^{\mathrm{pre}}\|_F &\leq \|\boldsymbol{W}_1\boldsymbol{W}_2 - \bar{\boldsymbol{W}}_1\bar{\boldsymbol{W}}_2)\|_F + 2\|\bar{\boldsymbol{W}}_1\bar{\boldsymbol{W}}_2 - \boldsymbol{A}^{\mathrm{pre}}\|_F \\
&\leq 2\sqrt{d}\exp(-C\tau)\boldsymbol{\Sigma}_{1,1}^{\mathrm{pre}} + 2\exp(-2C\tau) + 2\sqrt{d}\boldsymbol{\Sigma}_{1,1}^{\mathrm{pre}} \leq 4\sqrt{d}\boldsymbol{\Sigma}_{1,1}^{\mathrm{pre}}.
\end{aligned}$$

Combining the above,

$$\left|\|\boldsymbol{W}_1\boldsymbol{W}_2 - \boldsymbol{A}^{\mathrm{pre}}\|_F^2 - \|\bar{\boldsymbol{W}}_1\bar{\boldsymbol{W}}_2 - \boldsymbol{A}^{\mathrm{pre}}\|_F^2\right| \leq 16d\left(\boldsymbol{\Sigma}_{1,1}^{\mathrm{pre}}\right)^2\exp(-C\tau).$$

Combining all the above, we have

$$\left|\tilde{\mathcal{L}_{\mathrm{pre}}}(\boldsymbol{\theta}^n(0)) - \tilde{\mathcal{L}_{\mathrm{pre}}}(\bar{\boldsymbol{\theta}}^n(0))\right| \leq \exp(-C\tau)8d\boldsymbol{\Sigma}_{1,1}^{\mathrm{pre}}\left(2\boldsymbol{\Sigma}_{1,1}^{\mathrm{pre}} + \gamma^2\right) \leq (\boldsymbol{\Sigma}_{d,d}^{\mathrm{pre}})^2/2.$$

The final inequality follows from Assumption A.2. $\qquad\square$

## A.3. Dynamic Analysis of Finetuning Process

Before we proceed to the main result of finetuning, we will first analyze the dynamic of the finetuning process in this section.

We will introduce two auxiliary dynamics to help us track the evolution of the finetuning process.

The first auxiliary dynamic $\bar{\boldsymbol{\theta}}^n(t)$ is named as *Ideal initialization dynamic*, which is defined as the dynamic starting from the ideal initialization $\bar{\boldsymbol{\theta}}^n(0)$ in Equations (7) and (8) with the same update rule Equations (10) and (11) and data order as the finetuning process.

The second auxiliary dynamic $\hat{\boldsymbol{\theta}}^n(t)$ is named as *Ideal initialization with infinite batch size*, which is defined as the dynamic starting from the ideal initialization $\bar{\boldsymbol{\theta}}^n(0)$ in Equations (7) and (8) with the update rule Equations (16) and (17), which corresponds to the case when the batch size is infinite and $\Sigma_k^{(x)}$ converges to the identity matrix.

$$\hat{\boldsymbol{W}}_1^n(k+1) = \hat{\boldsymbol{W}}_1^n(k) - 2\eta(\hat{\boldsymbol{\theta}}^n(k) - \boldsymbol{A}^{\mathrm{ft}})(\hat{\boldsymbol{W}}_2^n(k))^\top - 2\eta\lambda(\hat{\boldsymbol{\theta}}^n(k) - \boldsymbol{\theta}^n(0))(\hat{\boldsymbol{W}}_2^n(k))^\top \tag{16}$$

$$\hat{\boldsymbol{W}}_2^n(k+1) = \hat{\boldsymbol{W}}_2^n(k) - 2\eta(\hat{\boldsymbol{W}}_1^n(k))^\top(\hat{\boldsymbol{\theta}}^n(k) - \boldsymbol{A}^{\mathrm{ft}}) - 2\eta\lambda(\hat{\boldsymbol{W}}_1^n(k))^\top(\hat{\boldsymbol{\theta}}^n(k) - \boldsymbol{\theta}^n(0)) \tag{17}$$

We will show the following results about these three dynamics:

1. Lemma A.7 provides analytical expression for the ideal initialization dynamic with infinite batch size.

2. Lemma A.17 shows that the ideal initialization dynamic with finite batch size is close to the ideal initialization dynamic with infinite batch size, with error bound depending on the batch size.

3. Lemma A.19 shows that the real initialization dynamic is close to the ideal initialization dynamic, with error bound depending on the scale of pretraining initialization (which then controls the distance between the real initialization and the ideal initialization by Theorem A.1).

4. We conclude our analysis by providing our assumption for the main result of the paper Assumption A.21 and show that the finetuning process tracks the ideal initialization dynamic with infinite batch size closely and eventually approximately converges to the minimum (Lemmas A.22 and A.23).

Throughout this subsection, we will call $\boldsymbol{W}_1$ and $\boldsymbol{W}_2$ as well conditioned if $\|\boldsymbol{W}_1\|_{\mathrm{op}} \leq 2\sqrt{\Gamma}$ and $\|\boldsymbol{W}_2\|_{\mathrm{op}} \leq 2\sqrt{\Gamma}$.

### A.3.1. Analytical Expression for the Ideal Initialization Dynamic with Infinite Batch Size

We will introduce the following function to better track the evolution of weight in the ideal initialization dynamic with infinite batch size.

$$f(x; \eta, \lambda, \sigma, \sigma_0) = x + 2\eta x(\sigma^2 - x^2) + 2\eta\lambda(\sigma_0^2 - x^2). \tag{18}$$

**Lemma A.7.** *For the ideal initialization dynamic with infinite batch size in Equations* (16) *and* (17)*, we have*

$$\hat{\boldsymbol{W}}_1^n(k) = U(\Sigma^n(k))^{1/2}$$
$$\hat{\boldsymbol{W}}_2^n(k) = (\Sigma^n(k))^{1/2}V$$

*where*

$$(\Sigma^n(k))_{i,i}^{1/2} = 1(i \leq n)f^{(k)}((\boldsymbol{\Sigma}_{i,i}^{\mathrm{pre}})^{1/2}; \eta, \lambda, (\boldsymbol{\Sigma}_{i,i}^{\mathrm{ft}})^{1/2}, (\boldsymbol{\Sigma}_{i,i}^{\mathrm{pre}})^{1/2}).$$

*Proof.* Consider

$$\Sigma_1^n(k) = U^T W_1^n(k)$$
$$\Sigma_2^n(k) = W_2^n(k)V$$

We then have

$$\Sigma_1^n(k+1) = \Sigma_1^n(k) - 2\eta\left(\Sigma_1^n(k)\Sigma_2^n(k) - \boldsymbol{\Sigma}^{\mathrm{ft}}\right)\Sigma_2^n(k)^T - 2\eta\lambda\left(\Sigma_1^n(k)\Sigma_2^n(k) - \Sigma_1^n(0)\Sigma_2^n(0)\right)\Sigma_2^n(k)^T$$

$$\Sigma_2^n(k+1) = \Sigma_2^n(k) - 2\eta\Sigma_1^n(k)^T\left(\Sigma_1^n(k)\Sigma_2^n(k) - \boldsymbol{\Sigma}^{\mathrm{ft}}\right) - 2\eta\lambda\Sigma_1^n(k)^T\left(\Sigma_1^n(k)\Sigma_2^n(k) - \Sigma_1^n(0)\Sigma_2^n(0)\right).$$

Through induction, we can prove that $\Sigma_1^n(k) = \Sigma_2^n(k)$ are diagonal for all $k$. This then follows from the definition of $f$. $\quad\square$

This suggests that $\hat{\boldsymbol{W}}_1^n(k)$ and $\hat{\boldsymbol{W}}_2^n(k)$ is always well bounded by $\Gamma$.

**Assumption A.8.** We have that learning rate $\eta$ and regularization parameter $\lambda$ are upper bounded,

$$4\eta(\lambda + 2)\Gamma < 1.$$

**Lemma A.9.** *Under Assumption A.8, for the ideal initialization dynamic with infinite batch size in Equations* (16) *and* (17), *we have that*

$$\|\hat{\boldsymbol{W}}_1^n(k)\|_{\text{op}} \leq \sqrt{\Gamma}$$
$$\|\hat{\boldsymbol{W}}_2^n(k)\|_{\text{op}} \leq \sqrt{\Gamma}$$

*with $\Gamma$ being the upper bound of $\boldsymbol{\Sigma}^{\text{pre}}$ and $\boldsymbol{\Sigma}^{\text{ft}}$ as defined in Equation* (12).

*Proof.* This is a direct consequence of Lemmas A.7 and A.28. $\qquad\square$

Next, we will show that $(U^T \hat{\boldsymbol{\theta}}^n(K)V)_{i,i}$ will converge to a weighted combination of $\boldsymbol{\Sigma}_{i,i}^{\text{pre}}$ and $\boldsymbol{\Sigma}_{i,i}^{\text{ft}}$ for finites steps $K$.

**Assumption A.10** (Large Enough but Finite Steps). We have that the step size $K \geq \frac{1}{\eta \min\{\boldsymbol{\Sigma}_{i,i}^{\text{pre}}, \boldsymbol{\Sigma}_{i,i}^{\text{ft}}\}} \log \frac{100\Gamma}{\epsilon}$ for some constant $\epsilon > 0$.

**Lemma A.11.** *Under Assumption A.8 and Assumption A.10, for the ideal initialization dynamic with infinite batch size in Equations* (16) *and* (17), *we have that for any $i \leq n$,*

$$\left\| (U^T \hat{\boldsymbol{\theta}}^n(K)V)_{i,i} - \frac{\boldsymbol{\Sigma}_{i,i}^{\text{pre}} + \lambda \boldsymbol{\Sigma}_{i,i}^{\text{ft}}}{1 + \lambda} \right\|_{\text{op}} \leq \epsilon.$$

*Proof.* By Lemmas A.7 and A.28, we have that

$$\left| (\boldsymbol{W}_1^n(K))_{i,i} - \frac{\boldsymbol{\Sigma}_{i,i}^{\text{pre}} + \lambda \boldsymbol{\Sigma}_{i,i}^{\text{ft}}}{1 + \lambda} \right| \leq (1 - 2\eta \min\{\boldsymbol{\Sigma}_{i,i}^{\text{pre}}, \boldsymbol{\Sigma}_{i,i}^{\text{ft}}\})^K \left| \boldsymbol{\Sigma}_{i,i}^{\text{pre}} - \frac{\boldsymbol{\Sigma}_{i,i}^{\text{pre}} + \lambda \boldsymbol{\Sigma}_{i,i}^{\text{ft}}}{1 + \lambda} \right|$$

This then suggests that once

$$K \geq \frac{1}{2\eta \min\{\boldsymbol{\Sigma}_{i,i}^{\text{pre}}, \boldsymbol{\Sigma}_{i,i}^{\text{ft}}\}} \log \frac{100\Gamma^{1/2}|\boldsymbol{\Sigma}_{i,i}^{\text{pre}} - \boldsymbol{\Sigma}_{i,i}^{\text{ft}}|}{\epsilon},$$

It then follows that

$$\left| (\boldsymbol{W}_1^n(K))_{i,i} - \frac{\boldsymbol{\Sigma}_{i,i}^{\text{pre}} + \lambda \boldsymbol{\Sigma}_{i,i}^{\text{ft}}}{1 + \lambda} \right| \leq \frac{\epsilon}{100\Gamma^{1/2}}.$$

Similarly, we have the bound for $(\boldsymbol{W}_2^n(K))_{i,i}$. Combining the two bounds, the proof is complete. $\qquad\square$

### A.3.2. CORRESPONDENCE BETWEEN IDEAL INITIALIZATION DYNAMIC WITH INFINITE BATCH SIZE AND FINITE BATCH SIZE

We then proceed to bound the difference between the ideal initialization dynamic with infinite batch size and the ideal initialization dynamic with finite batch size.

**Lemma A.12** (4.7.3 of (Vershynin, 2018)). *For a fixed $k$, there exists a constant $C_1$, with probability $1 - \delta$, we have that when batch size $m \geq d + \log(1/\delta)$,*

$$\|\Sigma_k^{(x)} - \boldsymbol{I}_d\|_{\text{op}} \leq C_1 \sqrt{\frac{d + \log(1/\delta)}{m}}$$

**Assumption A.13** (Large Batch Size). We have that for constant $C_1$ defined in Lemma A.12 and $\epsilon > 0$, $m \geq C_1^2(d - \log(10K\delta))/\epsilon^2$.

**Lemma A.14.** *Under Assumption A.13, for the ideal initialization dynamic with infinite batch size in Equations (16) and (17), we have that*

$$\forall k \leq K, \quad \|\Sigma_k^{(x)} - \boldsymbol{I}_d\|_{\mathrm{op}} \leq \epsilon$$

*with probability $1 - \delta$.*

*Proof.* This is a direct consequence of Lemma A.12 and Assumption A.13. □

**Lemma A.15.** *When the event defined in Assumption A.13 happens, for any $k \leq K$, for the same well-conditioned parameter $\boldsymbol{\theta}(k)$ and $\boldsymbol{\theta}(0)$, if applying the update rule Equations (16) and (17) yield $\hat{\boldsymbol{\theta}}(k+1)$ and applying the update rule Equations (10) and (11) yield $\bar{\boldsymbol{\theta}}(k+1)$, then the difference between $\hat{\boldsymbol{\theta}}(k+1)$ and $\bar{\boldsymbol{\theta}}(k+1)$ is bounded by*

$$\|\hat{\boldsymbol{W}}_1(k+1) - \bar{\boldsymbol{W}}_1(k+1)\|_{\mathrm{op}} \leq 32\eta\epsilon\Gamma^{3/2}$$
$$\|\hat{\boldsymbol{W}}_2(k+1) - \bar{\boldsymbol{W}}_2(k+1)\|_{\mathrm{op}} \leq 32\eta\epsilon\Gamma^{3/2}$$

*Proof.* Taking the difference between the two update rules, we have that

$$\|\hat{\boldsymbol{W}}_1(k+1) - \bar{\boldsymbol{W}}_1(k+1)\|_{\mathrm{op}} = 2\eta\|(\boldsymbol{\theta}(k) - \boldsymbol{A}^{\mathrm{ft}})\left(\Sigma_k^{(x)} - \boldsymbol{I}_d\right)\boldsymbol{W}_2(k)^\top\|_{\mathrm{op}}$$
$$\leq 2\eta\|\boldsymbol{\theta}(k) - \boldsymbol{A}^{\mathrm{ft}}\|_{\mathrm{op}}\|\Sigma_k^{(x)} - \boldsymbol{I}_d\|_{\mathrm{op}}\|\boldsymbol{W}_2(k)\|_{\mathrm{op}}$$
$$\leq 2\eta\|\boldsymbol{\theta}(k) - \boldsymbol{A}^{\mathrm{ft}}\|_{\mathrm{op}}\epsilon\|\boldsymbol{W}_2(k)\|_{\mathrm{op}}$$
$$\leq 32\eta\epsilon\Gamma^{3/2}.$$

Similarly we can have the bound for $\|\hat{\boldsymbol{W}}_2(k+1) - \bar{\boldsymbol{W}}_2(k+1)\|_{\mathrm{op}}$. □

**Lemma A.16.** *When the event defined in Assumption A.13 happens, for the ideal initialization dynamic with infinite batch size in Equations (16) and (17), consider two different well-conditioned parameters $\boldsymbol{\theta}(k)$ and $\boldsymbol{\theta}'(k)$ with the same initialization $\boldsymbol{\theta}(0)$, denote $\epsilon_k = \max\{\|\boldsymbol{W}_1(k) - \boldsymbol{W}_1'(k)\|_{\mathrm{op}}, \|\boldsymbol{W}_2(k) - \boldsymbol{W}_2'(k)\|_{\mathrm{op}}\}$. we have that*

$$\epsilon_{k+1} \leq (1 + 16\eta\Gamma)\epsilon_k.$$

*Proof.* Define $\boldsymbol{A}^{\mathrm{target}} = \frac{\lambda\boldsymbol{A}^{\mathrm{pre}} + \boldsymbol{A}^{\mathrm{ft}}}{1+\lambda}$.

Given the update rule, we have that

$$\boldsymbol{W}_1(k+1) - \boldsymbol{W}_1'(k+1) = \underbrace{(\boldsymbol{W}_1(k) - \boldsymbol{W}_1'(k))}_{\text{prev error}} - 2\eta\left[(\boldsymbol{\theta}(k) - \boldsymbol{A}^{\mathrm{target}})\boldsymbol{W}_2(k)^\top - (\boldsymbol{\theta}'(k) - \boldsymbol{A}^{\mathrm{target}})\boldsymbol{W}_2'(k)^\top\right].$$

We only need to properly bound the second term,

$$\|\left[(\boldsymbol{\theta}(k) - \boldsymbol{A}^{\mathrm{target}})\boldsymbol{W}_2(k)^\top - (\boldsymbol{\theta}'(k) - \boldsymbol{A}^{\mathrm{target}})\boldsymbol{W}_2'(k)^\top\right]\|_{\mathrm{op}}$$
$$\leq \|\boldsymbol{\theta}(k) - \boldsymbol{\theta}'(k)\|_{\mathrm{op}}\|\boldsymbol{W}_2(k)\|_{\mathrm{op}} + \|\boldsymbol{\theta}(k) - \boldsymbol{A}^{\mathrm{target}}\|_{\mathrm{op}}\|\boldsymbol{W}_2(k) - \boldsymbol{W}_2'(k)\|_{\mathrm{op}}$$

The difference between $\boldsymbol{\theta}(k)$ and $\boldsymbol{\theta}'(k)$ is bounded by

$$\|\boldsymbol{\theta}(k) - \boldsymbol{\theta}'(k)\|_{\mathrm{op}} \leq \|\boldsymbol{W}_1(k) - \boldsymbol{W}_1'(k)\|_{\mathrm{op}}\|\boldsymbol{W}_2(k)\|_{\mathrm{op}} + \|\boldsymbol{W}_1'(k)\|_{\mathrm{op}}\|\boldsymbol{W}_2(k) - \boldsymbol{W}_2'(k)\|_{\mathrm{op}} \leq 4\sqrt{\Gamma}\epsilon_k.$$

Therefore, we have that

$$\|\left[(\boldsymbol{\theta}(k) - \boldsymbol{A}^{\mathrm{target}})\boldsymbol{W}_2(k)^\top - (\boldsymbol{\theta}'(k) - \boldsymbol{A}^{\mathrm{target}})\boldsymbol{W}_2'(k)^\top\right]\|_{\mathrm{op}} \leq 16\Gamma\epsilon_k.$$

We then concludes that

$$\epsilon_{k+1} \leq (1 + 16\eta\Gamma)\epsilon_k.$$

This then concludes the proof. □

**Lemma A.17.** *When the event defined in Lemma A.14 happens for $\epsilon < \frac{1}{4(1+16\eta\Gamma)^K}$, define the error between the ideal initialization dynamic with infinite batch size and the ideal initialization dynamic with finite batch size as $\varepsilon_k = \max\{\|\hat{\boldsymbol{W}}_1(k) - \bar{\boldsymbol{W}}_1(k)\|_{\mathrm{op}}, \|\hat{\boldsymbol{W}}_2(k) - \bar{\boldsymbol{W}}_2(k)\|_{\mathrm{op}}\}$, then we have that*

$$\varepsilon_k \leq 2(1 + 16\eta\Gamma)^k \epsilon \Gamma^{1/2} < \Gamma^{1/2}/2.$$

*Proof.* From Lemma A.9, we have that $\hat{\boldsymbol{\theta}}$ is well-conditioned, if $\bar{\boldsymbol{\theta}}$ is well-conditioned, combining Lemmas A.15 and A.16, we have that

$$\varepsilon_{k+1} \leq (1 + 16\eta\Gamma)\varepsilon_k + 32\eta\epsilon\Gamma^{3/2}.$$

Now we can inductively prove that for $k \in [0, K]$,

$$\varepsilon_k \leq \left((1 + 16\eta\Gamma)^k - 1\right) 2\epsilon\Gamma^{1/2}.$$

Given that $\epsilon < \frac{1}{2(1+16\eta\Gamma)^K}$, we have that

$$\varepsilon_K < \Gamma^{1/2}/4.$$

This then concludes the proof. $\square$

### A.3.3. ERROR INCURS BY DIFFERENT INITIALIZATION

Finally, we will show that the real initialization dynamic is close to the ideal initialization dynamic, with error bound depending on the scale of pretraining initialization (which then controls the distance between the real initialization and the ideal initialization by Theorem A.1).

**Lemma A.18.** *When the event defined in Lemma A.14 happens for $\epsilon < \frac{1}{4(1+16\eta\Gamma)^K}$, for the ideal initialization dynamic with finite batch size in Equations (10) and (11), consider two different well-conditioned parameters $\boldsymbol{\theta}(k)$ and $\boldsymbol{\theta}'(k)$ with the same initialization $\boldsymbol{\theta}(0)$, denote $\epsilon_k = \max\{\|\boldsymbol{W}_1(k) - \boldsymbol{W}_1'(k)\|_{\mathrm{op}}, \|\boldsymbol{W}_2(k) - \boldsymbol{W}_2'(k)\|_{\mathrm{op}}\}$. we have that*

$$\epsilon_{k+1} \leq (1 + 32\eta\Gamma)\epsilon_k.$$

*Proof.* The proof is similar to Lemma A.16 and is omitted here. $\square$

**Lemma A.19.** *When the event defined in Lemma A.14 happens for $\epsilon < \frac{1}{4(1+32\eta\Gamma)^K}$, consider two finetuning processes, with $\boldsymbol{\theta}^n(t)$ starts from the real initialization $\boldsymbol{\theta}(n)$ in Theorem A.1 and $\bar{\boldsymbol{\theta}}^n(t)$ starts from the ideal initialization $\bar{\boldsymbol{\theta}}(n)$ in Equations (7) and (8). Then the two processes are close to each other for all $k \leq K$,*

$$\|\boldsymbol{W}_1^n(k) - \bar{\boldsymbol{W}}_1^n(k)\|_{\mathrm{op}} \leq (1 + 32\eta\Gamma)^k \exp(-C\tau).$$
$$\|\boldsymbol{W}_2^n(k) - \bar{\boldsymbol{W}}_2^n(k)\|_{\mathrm{op}} \leq (1 + 32\eta\Gamma)^k \exp(-C\tau).$$

*Proof.* Define $\tilde{\varepsilon}_k = \max\{\|\boldsymbol{W}_1^n(k) - \bar{\boldsymbol{W}}_1^n(k)\|_F, \|\boldsymbol{W}_2^n(k) - \bar{\boldsymbol{W}}_2^n(k)\|_F\}$. By Lemma A.17, $\bar{\boldsymbol{\theta}}$ is well-conditioned, if $\boldsymbol{\theta}$ is well-conditioned, combining Lemma A.18, we have that

$$\tilde{\varepsilon}_{k+1} \leq (1 + 32\eta\Gamma)\tilde{\varepsilon}_k.$$

This then suggests that

$$\tilde{\varepsilon}_k \leq (1 + 32\eta\Gamma)^k \exp(-C\tau).$$

This then concludes the proof. $\square$

A.3.4. COMBING TWO APPROXIMATIONS

**Lemma A.20.** *Under Assumption A.8 and Assumption A.13, for $\epsilon < \frac{1}{4(1+16\eta\Gamma)^K}$, with probability $1 - \delta$, we have that both $W_1^n(k)$ and $W_2^n(k)$ are well-conditioned and*

$$\|W_1^n(k) - \hat{W}_1^n(k)\|_{\mathrm{op}} \leq (1 + 32\eta\Gamma)^k \exp(-C\tau) + 2(1 + 16\eta\Gamma)^k \Gamma^{1/2}\epsilon.$$
$$\|W_2^n(k) - \hat{W}_2^n(k)\|_{\mathrm{op}} \leq (1 + 32\eta\Gamma)^k \exp(-C\tau) + 2(1 + 16\eta\Gamma)^k \Gamma^{1/2}\epsilon.$$

*Proof.* This is a direct consequence of Lemmas A.14, A.17 and A.19. $\qquad\square$

Given this lemma, we now present our main assumption and corresponding bound under this assumption.

**Technical Assumptions.** We will make the following technical assumptions to simplify the analysis.

**Assumption A.21.** We will make the following assumption to control the regularity of training. For arbitrary constant $\lambda_0$, for

$$\epsilon < \frac{1}{4000d} \frac{\min_{n\leq d}\{|\Sigma_{n,n}^{\mathrm{pre}} - \Sigma_{n,n}^{\mathrm{ft}}|^2\}}{(\lambda_0 + 1)^2 \Gamma^2}$$

,

1. Finite regularization force: $0 \leq \lambda < \lambda_0$.

2. (Assumption A.8) Finetuning learning rate is bounded:

$$4\eta(\lambda_0 + 2)\Gamma < 1$$

3. (Assumption A.10) The finite number of step $K \geq \frac{1}{\min\{\Sigma_{i,i}^{\mathrm{pre}}, \Sigma_{i,i}^{\mathrm{ft}}\}} \log \frac{100\Gamma}{\epsilon}$.

4. (Assumption A.13) Large enough batch size $m$,

$$m \geq \frac{C_1^2(d - \log(10dK\delta))}{\epsilon^2}(1 + 32\eta\Gamma)^{2K}$$

   for $C_1$ defined in Lemma A.12.

5. Small enough initialization error $\exp(-C\tau) \leq \Gamma^{1/2}\epsilon/(1 + 32\eta\Gamma)^K$ for $C$ defined in Theorem A.1.

We will first show this important lemma that the distance between the real initialization and the ideal initialization is bounded under Assumption A.21.

**Lemma A.22.** *Under Assumption A.21, with probability $1 - \delta$, we have that for every $n \leq d$ and $k \leq K$,*

$$\|\theta^n(k) - \hat{\theta}^n(k)\|_F \leq \frac{\min_{i\leq n}\{|\Sigma_{i,i}^{\mathrm{pre}} - \Sigma_{i,i}^{\mathrm{ft}}|^2\}}{1000(\lambda_0 + 1)^2 \Gamma}.$$

*Proof.* This is a consequence of Lemma A.20. However, to go from the operator norm bound on $W_1^n(k)$ and $W_2^n(k)$ to the Frobenius norm bound on $\theta^n(k)$, we need the following two inequalities. The first one provides an operator norm bound on the difference between $\theta^n(k)$ and $\hat{\theta}^n(k)$,

$$\|\theta^n - \hat{\theta}^n\|_{\mathrm{op}} \leq \|W_1^n(k) - \hat{W}_1^n(k)\|_{\mathrm{op}}\|\hat{W}_2^n(k)\|_{\mathrm{op}} + \|W_2^n(k) - \hat{W}_2^n(k)\|_{\mathrm{op}}\|\hat{W}_1^n(k)\|_{\mathrm{op}}$$
$$\leq 4\Gamma^{1/2}(\|W_1^n(k) - \hat{W}_1^n(k)\|_{\mathrm{op}} + \|W_2^n(k) - \hat{W}_2^n(k)\|_{\mathrm{op}}).$$

The second one uses this operator norm bound to bound the Frobenius norm of the difference between $\theta^n(k)$ and $\hat{\theta}^n(k)$,

$$\|\theta^n(k) - \hat{\theta}^n(k)\|_F \leq d\|\theta^n(k) - \hat{\theta}^n(k)\|_{\mathrm{op}}.$$

Combining these two inequalities with Assumption A.21, we get the desired result. $\qquad\square$

We can continue to show that the finetunig process approximately converges to the minimum.

**Lemma A.23.** *Under Assumption A.21, with probability $1 - \delta$, we have that for every $n \leq d$,*

$$\|U^T \boldsymbol{\theta}^n(K)V - \frac{\boldsymbol{\Sigma}_{:n,:n}^{\text{ft}} + \lambda \boldsymbol{\Sigma}_{:n,:n}^{\text{pre}}}{1 + \lambda}\|_F \leq \frac{\min_{i \leq n}\{|\boldsymbol{\Sigma}_{i,i}^{\text{pre}} - \boldsymbol{\Sigma}_{i,i}^{\text{ft}}|^2\}}{500(\lambda_0 + 1)^2 \Gamma}.$$

*Proof.* This is a consequence of Lemmas A.11 and A.22. □

### A.4. Formal Statement and Proof of Theorem 4.6

**Theorem A.24.** *Under Assumption A.21, with probability $1 - \delta$, For $\Delta_{\text{pre}}(n) = \mathcal{L}_{\text{pre}}(\boldsymbol{\theta}^n(K)) - \mathcal{L}_{\text{pre}}(\boldsymbol{\theta}^n(0))$. $\Delta_{\text{pre}}(n) \geq 0$ and $\Delta_{\text{pre}}(n)$ does not decrease with $n$.*

*Proof.* We will first provide a tight bound for $\Delta_{\text{pre}}(n)$. By Lemma A.22, we have that

$$\|\boldsymbol{\theta}^n(0) - \hat{\boldsymbol{\theta}}^n(0)\|_F \leq \frac{\min_{i \leq d}\{|\boldsymbol{\Sigma}_{i,i}^{\text{pre}} - \boldsymbol{\Sigma}_{i,i}^{\text{ft}}|^2\}}{100(\lambda_0 + 1)^2 \Gamma}.$$

and by Lemma A.23, we have that

$$\|U^T \boldsymbol{\theta}^n(K)V - \frac{\boldsymbol{\Sigma}_{:n,:n}^{\text{ft}} + \lambda \boldsymbol{\Sigma}_{:n,:n}^{\text{pre}}}{1 + \lambda}\|_F \leq \frac{\min_{i \leq n}\{|\boldsymbol{\Sigma}_{i,i}^{\text{pre}} - \boldsymbol{\Sigma}_{i,i}^{\text{ft}}|^2\}}{50(\lambda_0 + 1)^2 \Gamma}.$$

This suggest that

$$\begin{aligned}
\left|\mathcal{L}_{\text{pre}}(\boldsymbol{\theta}^n(0)) - \mathcal{L}_{\text{pre}}(\hat{\boldsymbol{\theta}}^n(0))\right| &= \left|\|\boldsymbol{\theta}^n(0) - \boldsymbol{A}^{\text{pre}}\|_F^2 - \|\hat{\boldsymbol{\theta}}^n(0) - \boldsymbol{A}^{\text{pre}}\|_F^2\right| \\
&\leq \|\boldsymbol{\theta}^n(0) - \hat{\boldsymbol{\theta}}^n(0)\|_F \|\boldsymbol{\theta}^n(0) + \hat{\boldsymbol{\theta}}^n(0) - 2\boldsymbol{A}^{\text{pre}}\|_{\text{op}} \\
&\leq 32\Gamma \|\boldsymbol{\theta}^n(0) - \hat{\boldsymbol{\theta}}^n(0)\|_F \\
&\leq \frac{\min_{i \leq d}\{|\boldsymbol{\Sigma}_{i,i}^{\text{pre}} - \boldsymbol{\Sigma}_{i,i}^{\text{ft}}|^2\}}{10(\lambda_0 + 1)^2}
\end{aligned}$$

Similarly, we have that

$$\left|\mathcal{L}_{\text{pre}}(\boldsymbol{\theta}^n(K)) - \mathcal{L}_{\text{pre}}(U \frac{\boldsymbol{\Sigma}_{:n,:n}^{\text{ft}} + \lambda \boldsymbol{\Sigma}_{:n,:n}^{\text{pre}}}{1 + \lambda} V^T)\right| \leq \frac{\min_{i \leq n}\{|\boldsymbol{\Sigma}_{i,i}^{\text{pre}} - \boldsymbol{\Sigma}_{i,i}^{\text{ft}}|^2\}}{5(\lambda_0 + 1)^2}.$$

Combining these two inequalities, we have that

$$\left|\Delta_n - \left(\mathcal{L}_{\text{pre}}(U \frac{\boldsymbol{\Sigma}_{:n,:n}^{\text{ft}} + \lambda \boldsymbol{\Sigma}_{:n,:n}^{\text{pre}}}{1 + \lambda} V^T) - \mathcal{L}_{\text{pre}}(U \boldsymbol{\Sigma}_{:n,:n}^{\text{pre}} V^T)\right)\right| \leq \frac{3\min_{i \leq n}\{|\boldsymbol{\Sigma}_{i,i}^{\text{pre}} - \boldsymbol{\Sigma}_{i,i}^{\text{ft}}|^2\}}{10(\lambda_0 + 1)^2}.$$

Meanwhile, we have that

$$\begin{aligned}
\mathcal{L}_{\text{pre}}(U \frac{\boldsymbol{\Sigma}_{:n,:n}^{\text{ft}} + \lambda \boldsymbol{\Sigma}_{:n,:n}^{\text{pre}}}{1 + \lambda} V^T) - \mathcal{L}_{\text{pre}}(U \boldsymbol{\Sigma}_{:n,:n}^{\text{pre}} V^T) &= \sum_{i=1}^n \left(\frac{\boldsymbol{\Sigma}_{i,i}^{\text{ft}} + \lambda \boldsymbol{\Sigma}_{i,i}^{\text{pre}}}{1 + \lambda} - \boldsymbol{\Sigma}_{i,i}^{\text{pre}}\right)^2 \\
&= \sum_{i=1}^n \left(\frac{\boldsymbol{\Sigma}_{i,i}^{\text{ft}} - \boldsymbol{\Sigma}_{i,i}^{\text{pre}}}{1 + \lambda}\right)^2.
\end{aligned}$$

Therefore if we additionally define $\Delta_0 = 0$, we have that for $1 \leq n \leq d$,

$$\Delta_n - \Delta_{n-1} \geq \frac{(\boldsymbol{\Sigma}_{n,n}^{\text{pre}} - \boldsymbol{\Sigma}_{n,n}^{\text{ft}})^2}{(1 + \lambda)^2} - \frac{3\min_{i \leq n}\{|\boldsymbol{\Sigma}_{i,i}^{\text{pre}} - \boldsymbol{\Sigma}_{i,i}^{\text{ft}}|^2\}}{5(\lambda_0 + 1)^2} > 0.$$

This completes the proof. □

## A.5. Formal Statement and Proof of Theorem 4.7

**Theorem A.25.**    *1. Under Assumption A.21, when $\lambda = 0$, with probability $1 - \delta$, if $A^{\mathrm{pre}}$ and $A^{\mathrm{ft}}$ are $(4, r)$-misaligned, then $\mathcal{L}_{\mathrm{pre}}(\boldsymbol{\theta}^n(K)) - \mathcal{L}_{\mathrm{pre}}(\boldsymbol{\theta}^{n-1}(K)) > 0$ for $n \leq r$.*

2. *Define the* inflection point $r_\lambda$ *as the smallest value of $r$ for which the pre-training loss $\mathcal{L}_{\mathrm{pre}}(n)$ increases monotonically for every $n > r$. Assume that regularization strength $\lambda_1 > \lambda_2 > 0$ yields iterates $\boldsymbol{\theta}_1$ and $\boldsymbol{\theta}_2$, if Assumption A.21 holds for*

$$\epsilon < \frac{1}{4000d} \frac{\min_{n \leq d}\{|\boldsymbol{\Sigma}_{n,n}^{\mathrm{pre}} - \boldsymbol{\Sigma}_{n,n}^{\mathrm{ft}}|^2\}}{\Gamma^2} \min\left\{ \left(\frac{1}{(1 + \lambda_2)^2} - \frac{1}{(1 + \lambda_1)^2}\right), \left(\frac{\lambda_1^2}{(1 + \lambda_1)^2} - \frac{\lambda_2^2}{(1 + \lambda_2)^2}\right) \right\},$$

*then with probability $1 - \delta$, we have that $r_{\lambda_1} \leq r_{\lambda_2}$ and the unregularized finetuning loss $\|\boldsymbol{\theta}_1^n(K) - A^{\mathrm{ft}}\|_F^2 > \|\boldsymbol{\theta}_2^n(K) - A^{\mathrm{ft}}\|_F^2$ for every $n$.*

*Proof.* This is the combination of Lemmas A.26 and A.27.    □

**Lemma A.26.** *Under Assumption A.21, if $\Sigma_{n,n}^{\mathrm{ft}} > 4\Sigma_{n,n}^{\mathrm{pre}}$ and $\lambda = 0$, then $\mathcal{L}_{\mathrm{pre}}(\boldsymbol{\theta}^n(K)) - \mathcal{L}_{\mathrm{pre}}(\boldsymbol{\theta}^{n-1}(K)) > 0$.*

*Proof.* With the same argument as in Theorem A.25, we have that

$$|\mathcal{L}_{\mathrm{pre}}(\boldsymbol{\theta}^n(K)) - \mathcal{L}_{\mathrm{pre}}(U\boldsymbol{\Sigma}_{:n,:n}^{\mathrm{ft}}V^T)| \leq \frac{\min_{i \leq n}\{|\boldsymbol{\Sigma}_{i,i}^{\mathrm{pre}} - \boldsymbol{\Sigma}_{i,i}^{\mathrm{ft}}|^2\}}{5}.$$

Noted that

$$\mathcal{L}_{\mathrm{pre}}(U\boldsymbol{\Sigma}_{:n,:n}^{\mathrm{ft}}V^T) - \mathcal{L}_{\mathrm{pre}}(U\boldsymbol{\Sigma}_{:n-1,:n-1}^{\mathrm{ft}}V^T) = (\boldsymbol{\Sigma}_{n,n}^{\mathrm{ft}} - \boldsymbol{\Sigma}_{n,n}^{\mathrm{pre}})^2 - (\boldsymbol{\Sigma}_{n,n}^{\mathrm{pre}})^2$$

We further have that $\boldsymbol{\Sigma}_{n,n}^{\mathrm{ft}} - \boldsymbol{\Sigma}_{n,n}^{\mathrm{pre}} > 2\boldsymbol{\Sigma}_{n,n}^{\mathrm{pre}}$. Therefore,

$$\mathcal{L}_{\mathrm{pre}}(\boldsymbol{\theta}^n(K)) - \mathcal{L}_{\mathrm{pre}}(\boldsymbol{\theta}^{n-1}(K))$$
$$\geq \mathcal{L}_{\mathrm{pre}}(U\boldsymbol{\Sigma}_{:n,:n}^{\mathrm{ft}}V^T) - \mathcal{L}_{\mathrm{pre}}(U\boldsymbol{\Sigma}_{:n-1,:n-1}^{\mathrm{ft}}V^T) - \frac{2(\boldsymbol{\Sigma}_{n,n}^{\mathrm{pre}})^2}{5} > 0.$$

This completes the proof.    □

**Lemma A.27.** *Assume that regularization strength $\lambda_1 > \lambda_2 > 0$ yields iterates $\boldsymbol{\theta}_1$ and $\boldsymbol{\theta}_2$, if Assumption A.21 holds for*

$$\epsilon < \frac{1}{4000d} \frac{\min_{n \leq d}\{|\boldsymbol{\Sigma}_{n,n}^{\mathrm{pre}} - \boldsymbol{\Sigma}_{n,n}^{\mathrm{ft}}|^2\}}{\Gamma^2} \min\left\{ \left(\frac{1}{(1 + \lambda_2)^2} - \frac{1}{(1 + \lambda_1)^2}\right), \left(\frac{\lambda_1^2}{(1 + \lambda_1)^2} - \frac{\lambda_2^2}{(1 + \lambda_2)^2}\right) \right\},$$

*then with probability $1 - \delta$, we have that $r_{\lambda_1} \leq r_{\lambda_2}$ and the unregularized finetuning loss $\|\boldsymbol{\theta}_1^n(K) - A^{\mathrm{ft}}\|_F^2 > \|\boldsymbol{\theta}_2^n(K) - A^{\mathrm{ft}}\|_F^2$ for every $n$.*

*Proof.* Following similar proof as in Lemma A.23, we have that with probability $1 - \delta$,

$$\left\|\boldsymbol{\theta}_1^n(K) - U\frac{\boldsymbol{\Sigma}_{:n,:n}^{\mathrm{ft}} + \lambda_1 \boldsymbol{\Sigma}_{:n,:n}^{\mathrm{pre}}}{1 + \lambda_1}V^T\right\|_F \leq \frac{\min_{i \leq n}\{|\boldsymbol{\Sigma}_{i,i}^{\mathrm{pre}} - \boldsymbol{\Sigma}_{i,i}^{\mathrm{ft}}|^2\}}{500\Gamma} \left(\frac{1}{(1 + \lambda_2)^2} - \frac{1}{(1 + \lambda_1)^2}\right).$$

and

$$\left\|\boldsymbol{\theta}_2^n(K) - U\frac{\boldsymbol{\Sigma}_{:n,:n}^{\mathrm{ft}} + \lambda_2 \boldsymbol{\Sigma}_{:n,:n}^{\mathrm{pre}}}{1 + \lambda_2}V^T\right\|_F \leq \frac{\min_{i \leq n}\{|\boldsymbol{\Sigma}_{i,i}^{\mathrm{pre}} - \boldsymbol{\Sigma}_{i,i}^{\mathrm{ft}}|^2\}}{500\Gamma} \left(\frac{1}{(1 + \lambda_2)^2} - \frac{1}{(1 + \lambda_1)^2}\right).$$

This then implies that

$$\left| \|\boldsymbol{\theta}_1^n(K) - A^{\mathrm{pre}}\|_F^2 - \|\frac{\boldsymbol{\Sigma}_{:n,:n}^{\mathrm{ft}} + \lambda_1 \boldsymbol{\Sigma}_{:n,:n}^{\mathrm{pre}}}{1 + \lambda_1} - \boldsymbol{\Sigma}^{\mathrm{pre}}\|_F^2 \right| \leq \frac{\min_{i \leq n}\{|\boldsymbol{\Sigma}_{i,i}^{\mathrm{pre}} - \boldsymbol{\Sigma}_{i,i}^{\mathrm{ft}}|^2\}}{50} \left(\frac{1}{(1 + \lambda_2)^2} - \frac{1}{(1 + \lambda_1)^2}\right).$$

Similar bound holds for $\|\boldsymbol{\theta}_2^n(K) - \boldsymbol{A}^{\mathrm{ft}}\|_F^2$.

Combining these two inequalities, we have that

$$
\left(\|\boldsymbol{\theta}_2^n(K) - \boldsymbol{A}^{\mathrm{pre}}\|_F^2 - \|\boldsymbol{\theta}_2^{n-1}(K) - \boldsymbol{A}^{\mathrm{pre}}\|_F^2\right) - \left(\|\boldsymbol{\theta}_1^n(K) - \boldsymbol{A}^{\mathrm{pre}}\|_F^2 - \|\boldsymbol{\theta}_1^{n-1}(K) - \boldsymbol{A}^{\mathrm{pre}}\|_F^2\right)
$$

$$
\geq \left(\left|\frac{\boldsymbol{\Sigma}_{n,n}^{\mathrm{ft}} + \lambda_2 \boldsymbol{\Sigma}_{n,n}^{\mathrm{pre}}}{1 + \lambda_2} - \boldsymbol{\Sigma}^{\mathrm{pre}}\right|^2 - \left|\frac{\boldsymbol{\Sigma}_{n,n}^{\mathrm{ft}} + \lambda_1 \boldsymbol{\Sigma}_{n,n}^{\mathrm{pre}}}{1 + \lambda_1} - \boldsymbol{\Sigma}^{\mathrm{pre}}\right|^2\right) - \frac{\min_{i \leq n}\{|\boldsymbol{\Sigma}_{i,i}^{\mathrm{pre}} - \boldsymbol{\Sigma}_{i,i}^{\mathrm{ft}}|^2\}}{25}\left(\frac{1}{(1 + \lambda_2)^2} - \frac{1}{(1 + \lambda_1)^2}\right)
$$

$$
\geq \left(\frac{1}{(1 + \lambda_2)^2} - \frac{1}{(1 + \lambda_1)^2}\right)\left(\|\boldsymbol{\Sigma}_{n,n}^{\mathrm{pre}} - \boldsymbol{\Sigma}_{n,n}^{\mathrm{ft}}\|_F^2 - \frac{\min_{i \leq n}\{|\boldsymbol{\Sigma}_{i,i}^{\mathrm{pre}} - \boldsymbol{\Sigma}_{i,i}^{\mathrm{ft}}|^2\}}{25}\right) > 0.
$$

This then suggests that $\|\boldsymbol{\theta}_2^n(K) - \boldsymbol{A}^{\mathrm{pre}}\|_F^2 > \|\boldsymbol{\theta}_2^{n-1}(K) - \boldsymbol{A}^{\mathrm{pre}}\|_F^2$ when $\|\boldsymbol{\theta}_1^n(K) - \boldsymbol{A}^{\mathrm{pre}}\|_F^2 > \|\boldsymbol{\theta}_1^{n-1}(K) - \boldsymbol{A}^{\mathrm{pre}}\|_F^2$, showing that $r_{\lambda_1} \leq r_{\lambda_2}$. Using similar argument, we can show that the unregularized finetuning loss $\|\boldsymbol{\theta}_1^n(K) - \boldsymbol{A}^{\mathrm{ft}}\|_F^2 > \|\boldsymbol{\theta}_2^n(K) - \boldsymbol{A}^{\mathrm{ft}}\|_F^2$ for every $n$. $\qquad\square$

### A.6. Technical Lemmas

In this section, we will first prove some of the technical lemmas on function $f$ defined in Equation (18). Recall that $f$ is defined as,

$$
f(x; \eta, \lambda, \sigma, \sigma_0) = x + 2\eta x(\sigma^2 - x^2) + 2\eta\lambda x(\sigma_0^2 - x^2).
$$

**Lemma A.28.** $\forall \sigma > 0, k \in \mathbb{N}$, When $(\lambda + 2)\eta\left(2\max\{\sigma^2, \sigma_0^2\} + \lambda + \lambda\frac{\sigma_0}{\sigma}\right) < 1$, define $\sigma^* = \sqrt{\frac{\sigma_0^2 + \lambda\sigma^2}{1 + \lambda}}$, it holds that $f^{(k)}(\sigma_0; \eta, \lambda, \sigma, \sigma_0)$ in $[\min\{\sigma, \sigma_0\}, \max\{\sigma, \sigma_0\}]$, and

$$
|f^{(k)}(\sigma_0; \eta, \lambda, \sigma, \sigma_0) - \sigma^*| \leq (1 - 2\eta\min\{\sigma^2, \sigma_0^2\})^k|\sigma_0 - \sigma^*|
$$

*Proof.* Let $g(x; \sigma, \sigma_0, \lambda) = x(x^2 - \sigma^2) + \lambda x(x^2 - \sigma_0)$. Then $g(\sigma^*; \sigma, \sigma_0, \lambda) = 0$.

We have that

$$
f(x; \eta, \lambda, \sigma, \sigma_0) = x - 2\eta g(x; \sigma, \sigma_0, \lambda).
$$

For any $x \in [\min\{\sigma, \sigma_0\}, \max\{\sigma, \sigma_0\}]$. As

$$
g(x; \sigma, \sigma_0, \lambda) = x(x^2 - \sigma^2) + \lambda x(x^2 - \sigma_0^2) = x(x - \sigma^*)(x + (\lambda + 1)\sigma^*).
$$

$$
f(x; \eta, \lambda, \sigma, \sigma_0) - \sigma^* = x - \sigma^* - 2\eta g(x; \sigma, \sigma_0, \lambda) + 2\eta g(\sigma^*; \sigma, \sigma_0, \lambda)
$$
$$
= (x - \sigma^*)(1 - 2\eta x(x + (\lambda + 1)\sigma^*)).
$$

When $x \in [\min\{\sigma, \sigma_0\}, \max\{\sigma, \sigma_0\}]$, $x(x + (\lambda + 1)\sigma^*) \geq \min\{\sigma^2, \sigma_0^2\}$. On the other hand

$$
x(x + (\lambda + 1)\sigma^*) \leq (\lambda + 2)\max\{\sigma^2, \sigma_0^2\}.
$$

This suggest that

$$
1 - 2\eta x(x + (\lambda + 1)\sigma^*) > 0.
$$

Therefore,

$$
|f(x; \eta, \lambda, \sigma, \sigma_0) - \sigma^*| \leq |x - \sigma^*|(1 - 2\eta\min\{\sigma^2, \sigma_0^2\}).
$$

Also $f(x; \eta, \lambda, \sigma, \sigma_0) - \sigma^*$ has the same sign as $x - \sigma^*$. This concludes the proof. $\qquad\square$

# B. Extended Related Work

Here we present an expanded and extended discussion of the related work.

**Loss of plasticity.** The idea that more training can be harmful to performance has been studied before in other continual learning settings. Named *loss of plasticity*, this phenomenon refers to the degradation of the ability for a model to adapt to a new task. This has mainly been studied in the context of training on small models with small datasets (Ash & Adams, 2020; Dohare et al., 2021) or reinforcement learning (Kumar et al., 2020; Lyle et al., 2022; 2023; Ma et al., 2023; Abbas et al., 2023). Loss of plasticity has been attributed to the loss curvature (Lyle et al., 2023; Lewandowski et al., 2023), increased weight norm (Nikishin et al., 2022), feature rank (Kumar et al., 2020; Gulcehre et al., 2022), and feature inactivity (Lyle et al., 2022; Dohare et al., 2021). Multiple remedies have been proposed, including changes to the neural network architecture (Lyle et al., 2023), resetting model parameters (Nikishin et al., 2024; D'Oro et al., 2022), and regularization (Kumar et al., 2023; Ash & Adams, 2020).

While prior work focused on reinforcement learning or small-scale, synthetic setups, our work considers the large-scale autoregressive language modeling setting. Unlike prior work, where pre-training is often harmful for the downstream fine-tuning task, we show that overtraining on generic web data can also degrade fine-tuning performance despite being expected to help. Additionally, we highlight an increased sensitivity to degradation of the pre-training loss that arises with overtraining, an aspect largely overlooked in the literature.

**Catastrophic forgetting.** The phenomenon of *catastrophic forgetting*—where neural networks trained sequentially on tasks tend to forget prior tasks–has also been well-documented in the literature (Kirkpatrick et al., 2017; French, 1999; Goodfellow et al., 2013; Kemker et al., 2018; Kotha et al., 2023). There have been several proposed mitigation strategies, for example, Ahn et al. (2019); Hou et al. (2018); Chaudhry et al. (2019a) propose using regularization to mitigate catastrophic forgetting. Other fixes include generative replay of examples from previous tasks (Shin et al., 2017) or maintaining a memory buffer of previous tasks (Chaudhry et al., 2019b; de Masson d'Autume et al., 2019). In this work, we show that catastrophic forgetting can become more severe with overtraining.

**Relationship between pre-training loss and downstream performance.** In our work, we argue that the degradation of the pre-training loss and the downstream loss may be related. Several works have tried to study the relationship between the pre-training loss in language models and their downstream performance. Liu et al. (2022) analyze the effect of pre-training beyond convergence and suggest that overtrained models exhibit better transfer to downstream tasks. Our work considers web-scale pre-training, which rarely converges in practice, so these findings do not contradict ours. Similarly, Tay et al. (2022); Zhang et al. (2023) highlight the effect of architecture on downstream generalization, given the same pretraining loss.

**Scaling laws for optimal pre-training.** In our work, we argue that training for fewer tokens can be beneficial for downstream performance after fine-tuning. Related to our work, Isik et al. (2024) proposes scaling laws for certain downstream translation tasks after fine-tuning, but does not observe degradation with overtraining. In addition, the optimal pre-training token budget has also been studied in other contexts. Notably, Kaplan et al. (2020); Hoffmann et al. (2022) demonstrate that, given a fixed compute budget, there exists an optimal token budget for each model size. Subsequent works have extended scaling laws to broader contexts, including transfer learning, contrastive training, training under data constraints, and predicting performance from factors other than pre-training tokens (Hernandez et al., 2021; Cherti et al., 2023; Muennighoff et al., 2023; Goyal et al., 2024; Liu et al., 2025; Bhagia et al., 2024). However, scaling laws are not always optimal for predicting performance. Diaz & Madaio (2024) argue that existing scaling laws do not always predict downstream performance accurately. In addition, multiple works have observed U-shaped trends in performance as models scale (Caballero et al., 2022; Wei et al., 2022; McKenzie et al., 2022a).

To reduce inference cost, practitioners have turned to developing capable small models, which often requires overtraining beyond the compute-optimal token budget. In fact, Sardana et al. (2024) show that pre-training loss continues to decrease when trained for up to 10,000 tokens per parameter. Gadre et al. (2024) validated similar observations and propose scaling laws to predict the model performance in this overtraining regime.

**Transfer learning theory** Finally, our theoretical analysis of catastrophic overtraining adopts a classical transfer learning setup based on deep linear networks (Gidel et al., 2019; Saxe et al., 2018). Wei et al. (2024); Arora et al. (2018) use this setup to study how models learn and store knowledge. Another group of studies explain how transfer learning can improve performance after pre-training (Saunshi et al., 2021; Wei et al., 2021; Shachaf et al., 2021). Chua et al. (2021); Wu et al.

(2020); Tripuraneni et al. (2020) specifically adopt a similar deep linear network setting to study feature learning during pre-training, and how these learned features can benefit downstream tasks. Kumar et al. (2022) explores how fine-tuning can lead to degradation of out-of-distribution performance.

## C. Experimental Details from Section 2: Large Model Experiments

In this section, we present all of the omitted experimental details from Section 2 that are necessarily for replication.

### C.1. Pre-trained models.

For our pre-trained models, we use checkpoints from three base models: OLMo-1B (Groeneveld et al., 2024b), OLMo-2-7B (OLMo et al., 2024), and LLM360-Amber (Liu et al., 2023b). We choose checkpoints that have been released on each of the model's HuggingFace pages, given by Table 1.

| Model | HuggingFace ID | Revision | Step | Token Budget |
|---|---|---|---|---|
| OLMo-1B | `allenai/OLMo-1B-hf` | `step10000-tokens41B` | 10k | 0.04T |
| | | `step117850-tokens494B` | 118k | 0.5T |
| | | `step358000-tokens1501B` | 358k | 1.5T |
| | | `step447000-tokens1874B` | 447k | 1.9T |
| | | `step561250-tokens2353B` | 561k | 2.4T |
| | | `step738000-tokens3094B` | 738k | 3.1T |
| OLMo-2-7B | `allenai/OLMo-2-1124-7B` | `stage1-step19000-tokens80B` | 19k | 0.08T |
| | | `stage1-step120000-tokens504B` | 120k | 0.5T |
| | | `stage1-step441000-tokens1850B` | 441k | 1.9T |
| | | `stage1-step584000-tokens2450B` | 584k | 2.5T |
| | | `stage1-step727000-tokens3050B` | 727k | 3.1T |
| | | `stage1-step928646-tokens3896B` | 929k | 3.9T |
| LLM360-Amber (7B) | `LLM360/Amber` | `ckpt_040` | 40 | 0.12T |
| | | `ckpt_102` | 102 | 0.31T |
| | | `ckpt_244` | 244 | 0.75T |
| | | `ckpt_306` | 306 | 0.94T |
| | | `ckpt_358` | 358 | 1.1T |
| | | `ckpt_410` | 410 | 1.3T |

*Table 1.* Pre-trained models used in our experiments in Section 2.

### C.2. Fine-tuning setup.

We fine-tune with two different common post-training paradigms: instruction tuning and multimodal tuning. For instruction tuning, we use the following datasets.

**Anthropic-HH** (Bai et al., 2022). While Anthropic-HH is typically a dataset designed for preference tuning—the dataset includes both a "chosen" and a "rejected" response for each instruction—it can also be used as a standard instruction tuning dataset by treating the "chosen" response as the target. Anthropic-HH contains 180k instructions and responses.

**TULU** (Wang et al., 2023). We use the version 1.0 of the TULU SFT mixture, which contains 490k instructions and responses. However, for compute efficiency, we only use a randomly selected 200k subset.

**LLaVA** (Liu et al., 2023a). We use the LLaVA visual instruction tuning framework to train multimodel models. The LLaVA framework involves two stages: first, fine-tuning an adapter between a vision model and a pre-trained language model, and then fine-tuning the entire model to follow instructions in the presence of images.

When fine-tuning for instruction tuning, we use the standard SFT training algorithm with the following hyperparameters, as shown in Table 2. In this table, we also present the hyperparameters we use with the LLaVA framework, using the defaults for all non-specified hyperparameters.

| Dataset | Batch size | Learning rates | Learning rate schedule | Warmup steps | Optimizer | Weight decay |
|---|---|---|---|---|---|---|
| Anthropic-HH | 256 | `1e-6, 5e-6, 1e-5, 5e-5, 8e-5, 1e-4, 2e-4` | Cosine | 20 | AdamW | 0 |
| Alpaca | 256 | `1e-6, 5e-6, 1e-5, 5e-5, 8e-5, 1e-4, 2e-4` | Cosine | 20 | AdamW | 0 |
| TULU | 256 | `1e-6, 5e-6, 1e-5, 5e-5, 8e-5, 1e-4, 2e-4` | Cosine | 20 | AdamW | 0 |
| Visual (LLaVa) Stage 1 (Projector training) | 256 | `1e-3` | Cosine | 50 | AdamW | 0 |
| Visual (LLaVa) Stage 2 (Inst. tuning) | 256 | `8e-6, 1e-5, 2e-5, 4e-5, 1e-4` | Cosine | 40 | AdamW | 0 |

*Table 2.* Hyperparameters used for instruction tuning and LLaVA.

### C.3. Evaluations

We evaluate the fine-tuned models in two settings: downstream evaluations—tasks that is representative of the goal of fine-tuning—and generalist evaluations—tasks that are representative of the model's overall language understanding and inference capabilities. For downstream evaluations, we use the following datasets.

**AlpacaEval** (Li et al., 2023b). To evaluate the downstream performance of instruction-tuned models, we use AlpacaEval, a benchmark for evaluating the quality of a model's response to an instruction. The AlpacaEval benchmark contains 20k instructions, and measures the win-rate of the fine-tuned model against a reference model. By default, AlpacaEval reports win-rate vs GPT-4 responses. However, we evaluate models that are weak by comparison to GPT-4. If we compare against GPT-4, the win rate is so low that it is difficult to see the differences between models. Thus, we compare against a weaker model. In particular, for each of our models, we use a reference model of the same architecture that was also fine-tuned on the same dataset. More specifically, we use the model trained with seed 0 with learning rate $10^{-5}$. This means that the AlpacaEval scores are not comparable across different graphs, as the reference generations are different for each model and dataset. Additionally, the AlpacaEval score of the model trained with seed 0 and learning rate $10^{-5}$ is 50% by definition. Overall, we adopt these choices to ensure that the reference generations are comparable to each model output. We use LLaMA-3-70B-Instruct (Grattafiori et al., 2024) as an evaluator to determine the win rate.

**VLM Score.** To evaluate the downstream performance of our LLaVA models, we use an average of the following five standard vision-language benchmarks: MME (Fu et al., 2024), GQA (Hudson & Manning, 2019), AI2D (Kembhavi et al., 2016), POPE (Li et al., 2023c), and TextVQA (Singh et al., 2019). We report the average as the "VLM score".

**Generalist evaluations.** To evaluate each language model for generalist capabilities, we consider a suite of ten commonly used LLM evaluation benchmarks. These tasks assess performance beyond the fine-tuning task. These tasks cover reasoning (ARC_Challenge and ARC_Easy (Clark et al., 2018)), commonsense (PIQA (Bisk et al., 2020), Winogrande (Sakaguchi et al., 2021)), natural language inference (BoolQ (Clark et al., 2019), COPA, SCIQ) and sentence completion (HellaSwag). For all of our evaluations, we report 5-shot performance.

## D. Experimental Details from Section 3: Controlled Experiments

In this section, we provide additional experimental details for the controlled experiments presented in Section 3.

### D.1. Pre-training and fine-tuning setup.

For our controlled experiments, we pre-train models using the OLMo codebase (Groeneveld et al., 2024b). We use muP parameterization for all of our experiments (Yang et al., 2022).

**Pre-training.** We train three different model classes: OLMo-15M, OLMo-30M, and OLMo-90M with 15M, 30M and 90M non-embedding parameters, respectively. We use the following hyperparameters for pre-training, as shown in Table 3. For

each model, we train for tokens in the range 4B, 8B, 16B, 32B, 64B, 128B using the pre-tokenized C4 "high quality" web data distributed by OLMo (OLMo et al., 2024). We train with 8xA100 GPUs.

| Hyperparameters | OLMo-15M | OLMo-30M | OLMo-90M |
|---|---|---|---|
| Layers | 3 | 6 | 9 |
| Heads | 3 | 6 | 9 |
| Number of unique tokens | 50304 | 50304 | 50304 |
| Hidden dimensions | 192 | 384 | 576 |
| Inner MLP dimensions | 768 | 1536 | 2304 |
| Max context length | 1024 | 1024 | 1024 |
| Activation type | SwiGLU | SwiGLU | SwiGLU |
| Attention dropout | 0.1 | 0.1 | 0.1 |
| Residual dropout | 0.1 | 0.1 | 0.1 |
| Embedding dropout | 0.1 | 0.1 | 0.1 |
| Optimizer | AdamW | AdamW | AdamW |
| Learning rate | 0.0003 | 0.0003 | 0.0003 |
| Beta1 | 0.9 | 0.9 | 0.9 |
| Beta2 | 0.95 | 0.95 | 0.95 |
| Learning rate scheduler | Cosine | Cosine | Cosine |
| Warmup steps | 10% of training | 10% of training | 10% of training |
| Weight decay | 0.1 | 0.1 | 0.1 |
| Batch size | 256 | 256 | 256 |

*Table 3.* Pre-training hyperparameters used in our controlled experiments.

For each model, we anneal the learning rate to zero over the course of training, at the rate specified by the cosine learning rate scheduler.

**Fine-tuning.** For each of our controlled experiments, we fine-tune the pre-trained models on a series of downstream tasks of two types: classification and language modeling. These ten datasets are: classification—SUBJ (Pang & Lee, 2004), BoolQ (Clark et al., 2019), MR (Conneau & Kiela, 2018), CR (Conneau & Kiela, 2018), RTE (Dagan et al., 2005), TREC (Voorhees & Tice, 2000), English Tweet sentiment (Maggie et al., 2020), SIQA (Sap et al., 2019), and language modeling—GSM8k (Cobbe et al., 2021), Starcoder-Python (Li et al., 2023a). For Starcoder-Python, we use a 5k example subset. To avoid confusion, note that despite the fact that GSM8k is often evaluated as a math reasoning benchmark, we treat it as a language modeling task to evaluate how well the models can learn math-style text. We use the following hyperparameters for fine-tuning, as shown in Table 4.

| Hyperparameters | Values |
|---|---|
| Learning rate | 4e-6, 8e-6, 1e-5, 2e-5, 4e-5, 5e-5, 6e-5, 7e-5, 8e-5, 9e-5, 1e-4, 1.1e-4, 1.2e-4, 1.4e-4, 1.6e-4, 1.8e-4, 2e-4, 2.4e-4, 4e-4, 5e-4, 6e-4, 8e-4, 1e-3, 2e-3, 3e-3, 4e-3, 6e-3 |
| Batch size | 32, 64*, 256 |
| Learning rate scheduler | Cosine*, Constant |
| Optimizer | AdamW |
| Weight decay | 0.0 |
| Warmup steps | 10% of training |
| Epochs | 4 |

*Table 4.* Fine-tuning hyperparameters used in our controlled experiments. We tune over all specified learning rates. For the other hyperparameters, when multiple are specified, the asterisks (*) indicates the default value which is used unless a different hyperparameter is specified. We perform early stopping over the number of epochs.

**Evaluation.** For tuning, we use a heldout validation set from each dataset, but report scores on a separate heldout test set. In order to compute the perplexity for classification tasks, we compute a score for each class by measuring the length-normalized likelihood of the class, and then report the perplexity over the classes. For generative tasks, we use the standard language modeling loss. As a measure of generalist capability, we report the perplexity on a heldout C4 web data set.

**Appropriate learning rate ranges for Figure 5.** For visualization purposes, we choose to plot a subset of the learning rates which we evaluate in Figure 5. In particular, we plot learning rates where the maximum pre-training perplexity, over all

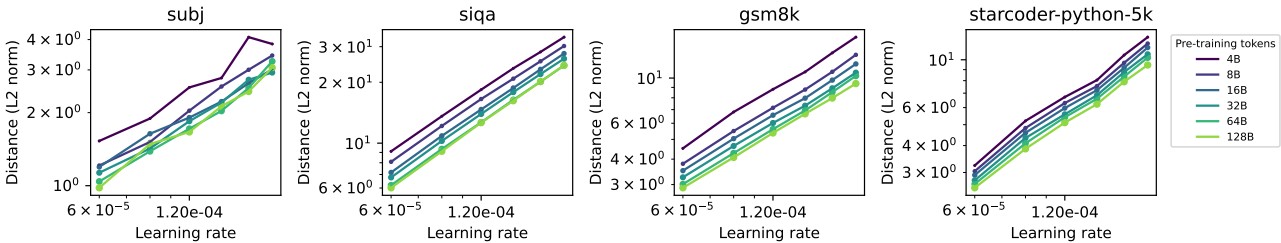

*Figure 8.* **Distance, as measured by L2 norm, between the pre-trained and fine-tuned model as a function of learning rate for OLMo-30M.** More specifically, if $\theta_{\text{pre}}$ and $\theta_{\text{ft}}$ are the parameters of the pre-trained and fine-tuned models, respectively, we plot $\|\theta_{\text{pre}} - \theta_{\text{ft}}\|_2$ as a function of the learning rate. We observe that the distance between the pre-trained and fine-tuned model is not exactly, but approximately, directly proportional to the learning rate and independent of the amount of pre-training.

token budgets, is less than 6. This ensures that the learning rates we plot are in a range where the model is still retaining pre-training capability, and has not degenerated to a high perplexity which may not represent the more general case.

**Using learning rate as a proxy for a fixed perturbation size.** We report the distance between the pre-trained and fine-tuned model as a function of the learning rate for different token budgets in Figure 8. Recall, from Section 3, that we specified that the learning rate is an approximate proxy for the size of the perturbation applied to the model. We observe that the distance between the pre-trained and fine-tuned model is not exactly, but approximately, directly proportional to the learning rate and independent of the amount of pre-training.

### D.2. Gaussian perturbations.

In this subsection, we outline the details concerning Gaussian perturbations applied during our experiments. In particular, we perturb each parameter by a random value sampled from a mean-zero Gaussian distribution and evaluate the degradation of pre-training perplexity in Section 3. Using an isotropic Gaussian perturbation, i.e., perturbing each parameter by the same amount, would discount differences in parameter magnitude across different layers. To account for this, we choose to scale the perturbation to each layer to be approximately proportional to the magnitude of the parameter in that layer—however, we want the magnitude to be constant for different pre-training token budgets. Thus, we choose to normalize the magnitude of each perturbation to the same magnitude as the layer at initialization prior to pre-training.

## E. Connection Between Progressive Sensitivity and Sharpness

In this section, we discuss the connection between our *progressive sensitivity* conjecture and the phenomenon known as *progressive sharpening* (Cohen et al., 2021) in greater detail.

**Progressive sharpening.** This phenomenon refers to the empirical observation that over training with a fixed learning rate, the spectral norm $\|\nabla^2 \mathcal{L}(\theta)\|_2$ of the Hessian of the loss function $\mathcal{L}$ at the parameters $\theta$ increases over time, at least early in training. In the case of of (full batch) gradient descent with a fixed learning rate $\eta$, $\|\nabla^2 \mathcal{L}(\theta)\|_2$ specifically increases until it reaches $2/\eta$, which is discussed in detail in Cohen et al. (2021). In addition to the spectral norm, other norms of the Hessian, such as the trace norm, also exhibit a similar behavior.

**Relationship between progressive sensitivity and progressive sharpening when loss is quadratic.** As it turns out, progressive sensitivity and progressive sharpening are closely related specifically in the quadratic setting. In particular, consider a quadratic loss function $\mathcal{L}(\theta) = \frac{1}{2}\theta^\top H \theta + g^\top \theta + c$, where $\theta \in \mathbb{R}^d$, $H \in \mathbb{R}^{d \times d}$ is a symmetric matrix, $g \in \mathbb{R}^d$, and $c \in \mathbb{R}$. We will look specifically at the sensitivity to a Gaussian perturbation $\Delta(\theta, \lambda) = \mathbb{E}\left[\mathcal{L}(\theta + \lambda \varepsilon) - \mathcal{L}(\theta)\right]$, where $\varepsilon \sim \mathcal{N}(0, I)$ is a unit Gaussian vector.

**Proposition E.1.** *The sensitivity of $\mathcal{L}$ to a Gaussian perturbation is given by $\Delta(\theta, \lambda) = \frac{1}{2}\lambda^2 \operatorname{Tr} H$.*

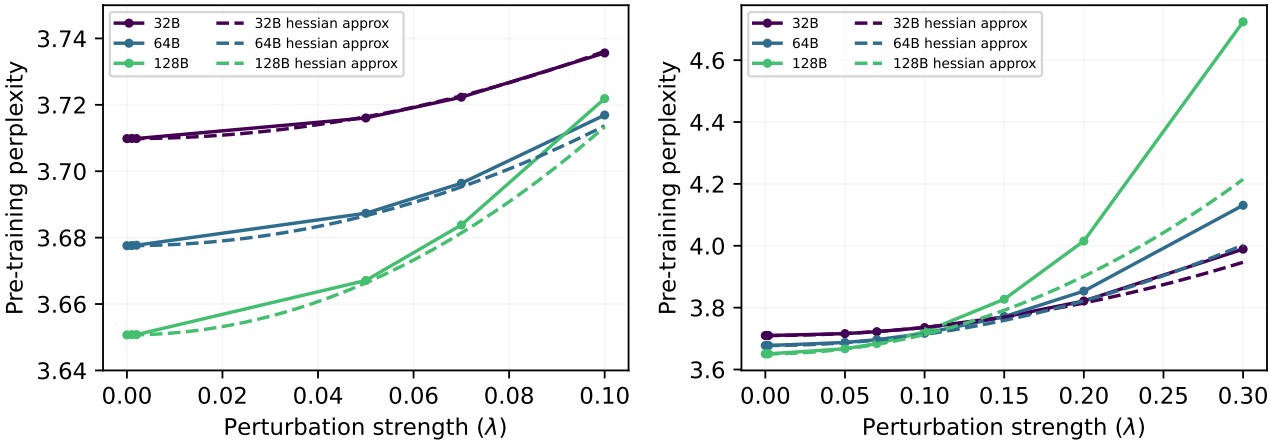

*Figure 9.* **Hessian approximation of the pre-training loss under a single interpolated Gaussian parameter perturbation.** We randomly draw a Gaussian perturbation $\varepsilon$, and then compute the loss $L(\theta + \lambda\varepsilon)$, where $\lambda$ is the scaling factor, for many different $\lambda$ (extremely close to zero on the left, and with a wider range on the right). We then compute Hessian, and use it to render the quadratic approximation of the loss.

*Proof.* We have,

$$\mathbb{E}\left[\mathcal{L}(\theta) - \mathcal{L}(\theta + \lambda\varepsilon)\right] = \mathbb{E}\left[\frac{1}{2}\left((\theta + \lambda\varepsilon)^\top H(\theta + \lambda\varepsilon) + g^\top(\theta + \lambda\varepsilon) + c\right) - \frac{1}{2}(\theta^\top H\theta + g^\top\theta + c)\right] \tag{19}$$

$$= \mathbb{E}\left[\frac{1}{2}\lambda^2\varepsilon^\top H\varepsilon\right] = \frac{1}{2}\lambda^2\operatorname{Tr}H, \tag{20}$$

where the second equality follows from the linearity of expectation and the fact that $\mathbb{E}[\varepsilon] = 0$. $\qquad\square$

This proposition establishes that the sensitivity under a Gaussian perturbation is exactly related to the Hessian when the loss function is quadratic. This connection will hold, in general, when the loss function is well-approximated by its second-order Taylor expansion, such as when $\lambda$ is small. In this instance, progressive sharpening and progressive sensitivity are closely related.

**Progressive sharpness is not sufficient to explain degradation when $\lambda$ is large.** We plot the empirical loss of three different OLMo-30B models (trained on 32B, 64B, and 128B tokens) under a Gaussian perturbation with perturbation strength $\lambda$, as well as the second-order Taylor approximation in Figure 9. In particular, we draw the perturbation $\varepsilon$ with the distribution described in Appendix D.2. We observe that while the loss is well-approximated by the Hessian when $\lambda$ is small (left), the approximation breaks down when $\lambda$ is large (right), and the actual loss is substantially higher than the quadratic approximation.

**Progressive sharpness is not a sufficient explanation for fine-tuning sensitivity.** Similar to the Gaussian case, we consider the loss of three OLMo-30B models as they are interpolated between the base model and the model fine-tuned on ag_news in Figure 10. In this example, a perturbation strength of $\lambda = 0$ corresponds to the base model, while a perturbation strength of $\lambda = 1$ corresponds to the fine-tuned model. Similar to the Gaussian case, we observe that the loss is not well-approximated by the Hessian when $\lambda$ is large, and the actual loss is substantially higher than the quadratic approximation (right).

**Progressive sensitivity as a generalization of progressive sharpness.** Our results highlight that in addition to progressive sharpness, which specifically refers to a progressive increase in the eigenvalues of the Hessian of the loss function with training, there is a more global phenomenon where the loss becomes even more sensitive to perturbations than the quadratic approximation predicts.

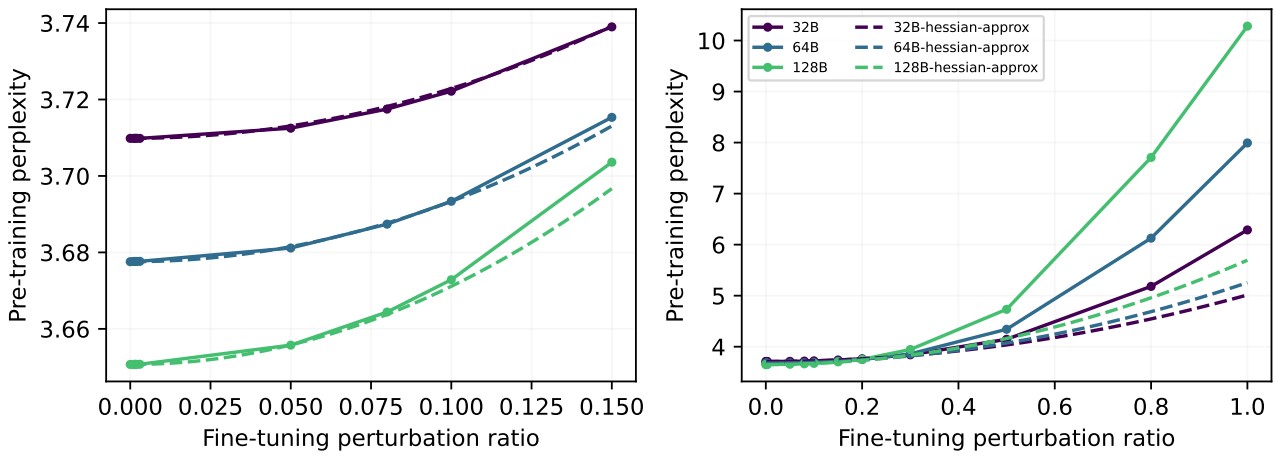

*Figure 10.* **Hessian approximation of the pre-training loss under an interpolated fine-tuning perturbation.** We fine-tune each model on ag_news yielding a fine-tuning perturbation $\varepsilon$, and then compute the loss $L(\theta + \lambda\varepsilon)$, where $\lambda$ is the scaling factor, for many different $\lambda$ (extremely close to zero on the left, and with a wider range on the right). We then compute Hessian, and use it to render the quadratic approximation of the loss.

## F. Omitted Figures from Section 2: Large Model Experiments

In this section, we provide the omitted figures from Section 2 that show the results of the extended experiments with large models.

The following Table 5 lists the table of contents for the omitted figures.

| Dataset (Variant) | OLMo-1B | OLMo-2-7B | LLM360-7B |
|---|---|---|---|
| Anthropic-HH (tuned learning rate) | Figure 11 | Figure 13 | Figure 15 |
| Anthropic-HH (all learning rates) | Figure 12 | Figure 14 | Figure 16 |
| TULU (tuned learning rate) | Figure 17 | Figure 19 | Figure 21 |
| TULU (all learning rates) | Figure 18 | Figure 20 | Figure 22 |
| VLM (tuned learning rate) | Figure 23 | Figure 25 | Figure 27 |
| VLM (all learning rates) | Figure 24 | Figure 26 | Figure 28 |

*Table 5.* Figure references for each dataset (Alpaca, Anthropic-HH, TULU, VLM) and model (OLMo-1B, OLMo-2-7B, LLM360-7B), separated by learning rate tuning variant.

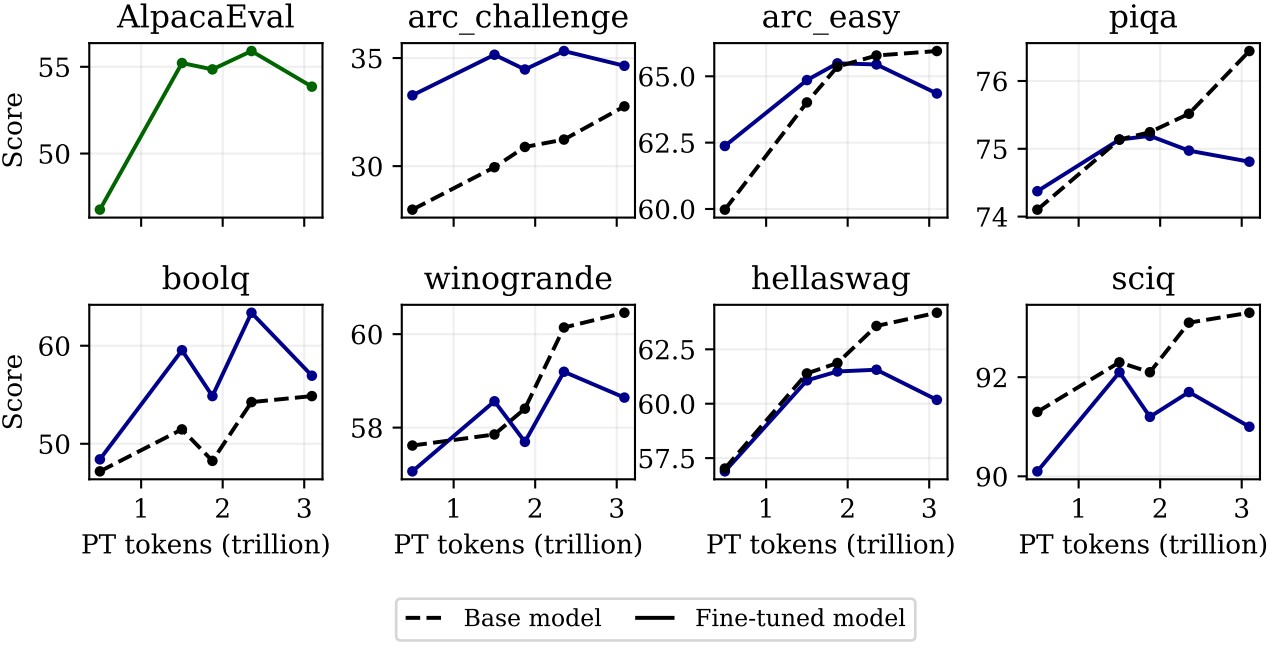

*Figure 11.* **Evaluation OLMo-1B post-trained on Anthropic-HH as a function of the number of pre-trained tokens, with tuned learning rates.** We report the scores on eight different datasets: AlpacaEval is considered to be the main evaluation of interest (corresponding with the downstream performance), and the other datasets are considered out-of-distribution (corresponding with the generalist performance). We use the intermediate checkpoints from Table 1 for the evaluation. We tune the learning rate for each checkpoint to maximize the main evaluation (AlpacaEval). This figure is analogous to Figure 2.

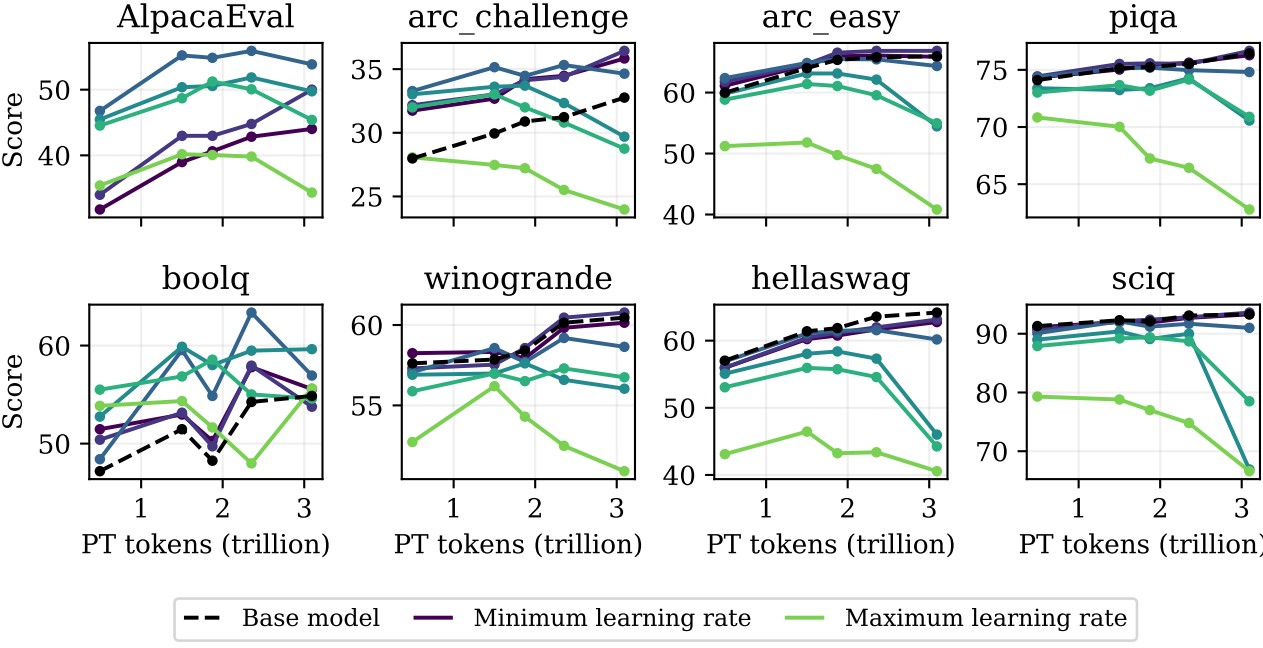

*Figure 12.* **Evaluation OLMo-1B post-trained on Anthropic-HH as a function of the number of pre-trained tokens, for all learning rates.** We report the scores on eight different datasets: AlpacaEval is considered to be the main evaluation of interest (corresponding with the downstream performance), and the other datasets are considered out-of-distribution (corresponding with the generalist performance). We use the intermediate checkpoints from Table 1 for the evaluation. We also compare to the base model (dashed line). This figure is similar to Figure 11, except we plot every learning rate, with a line representing a fixed learning rate.

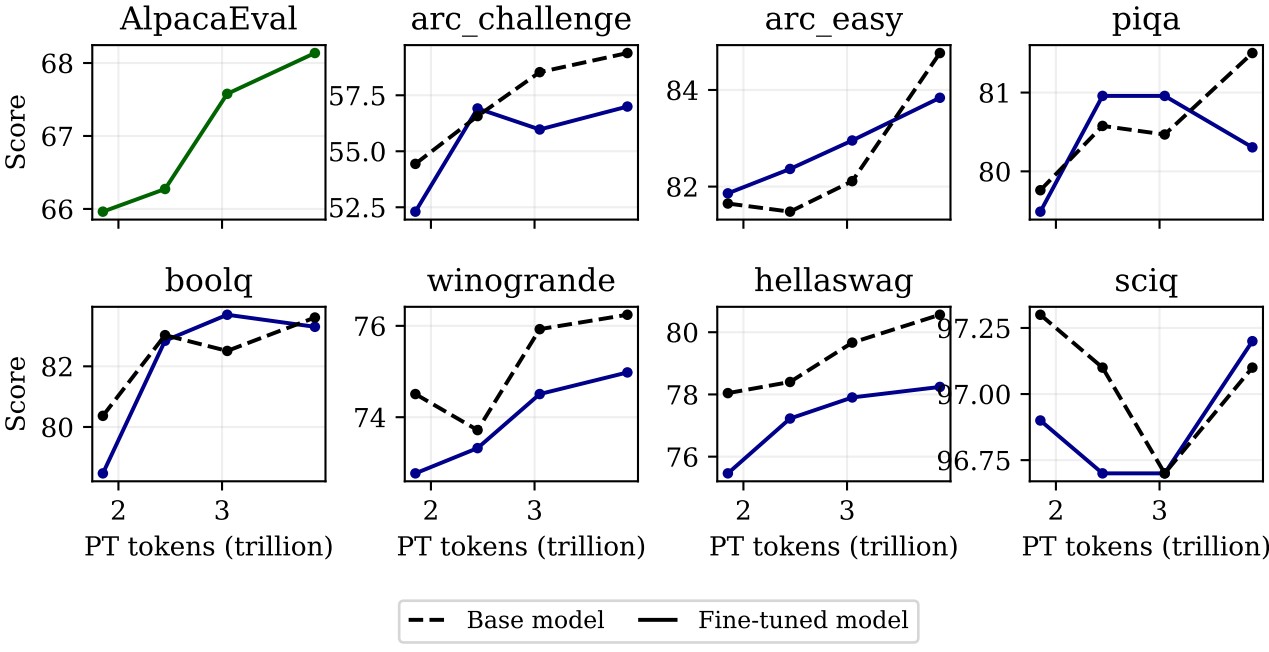

*Figure 13.* **Evaluation OLMo-2-7B post-trained on Anthropic-HH as a function of the number of pre-trained tokens, with tuned learning rates.** We report the scores on eight different datasets: AlpacaEval is considered to be the main evaluation of interest (corresponding with the downstream performance), and the other datasets are considered out-of-distribution (corresponding with the generalist performance). We use the intermediate checkpoints from Table 1 for the evaluation. We tune the learning rate for each checkpoint to maximize the main evaluation (AlpacaEval). This figure is analogous to Figure 2.

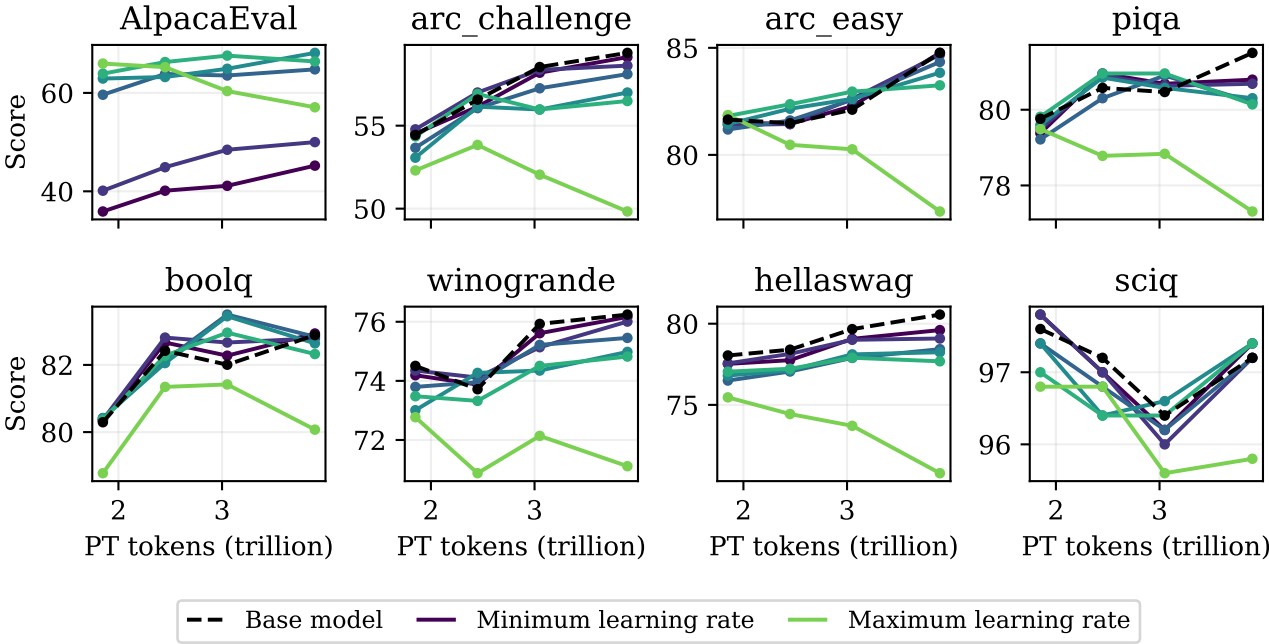

*Figure 14.* **Evaluation OLMo-2-7B post-trained on Anthropic-HH as a function of the number of pre-trained tokens, for all learning rates.** We report the scores on eight different datasets: AlpacaEval is considered to be the main evaluation of interest (corresponding with the downstream performance), and the other datasets are considered out-of-distribution (corresponding with the generalist performance). We use the intermediate checkpoints from Table 1 for the evaluation. We also compare to the base model (dashed line). This figure is similar to Figure 13, except we plot every learning rate, with a line representing a fixed learning rate.

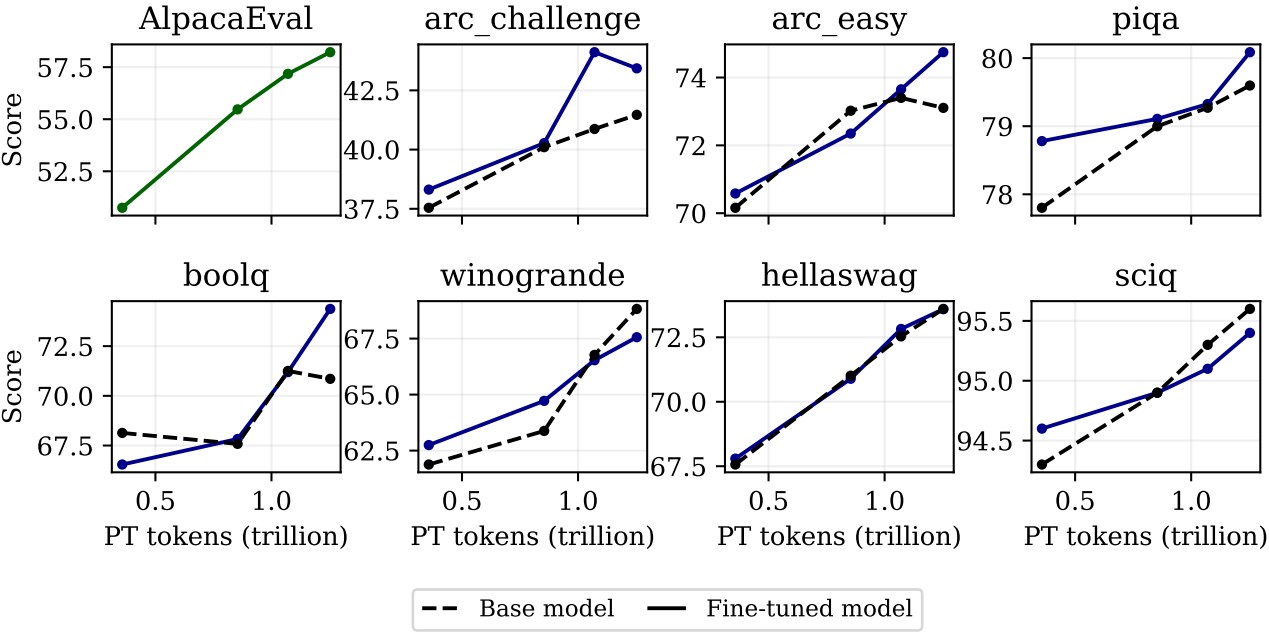

*Figure 15.* **Evaluation LLM360-7B post-trained on Anthropic-HH as a function of the number of pre-trained tokens, with tuned learning rates.** We report the scores on eight different datasets: AlpacaEval is considered to be the main evaluation of interest (corresponding with the downstream performance), and the other datasets are considered out-of-distribution (corresponding with the generalist performance). We use the intermediate checkpoints from Table 1 for the evaluation. We tune the learning rate for each checkpoint to maximize the main evaluation (AlpacaEval). This figure is analogous to Figure 2.

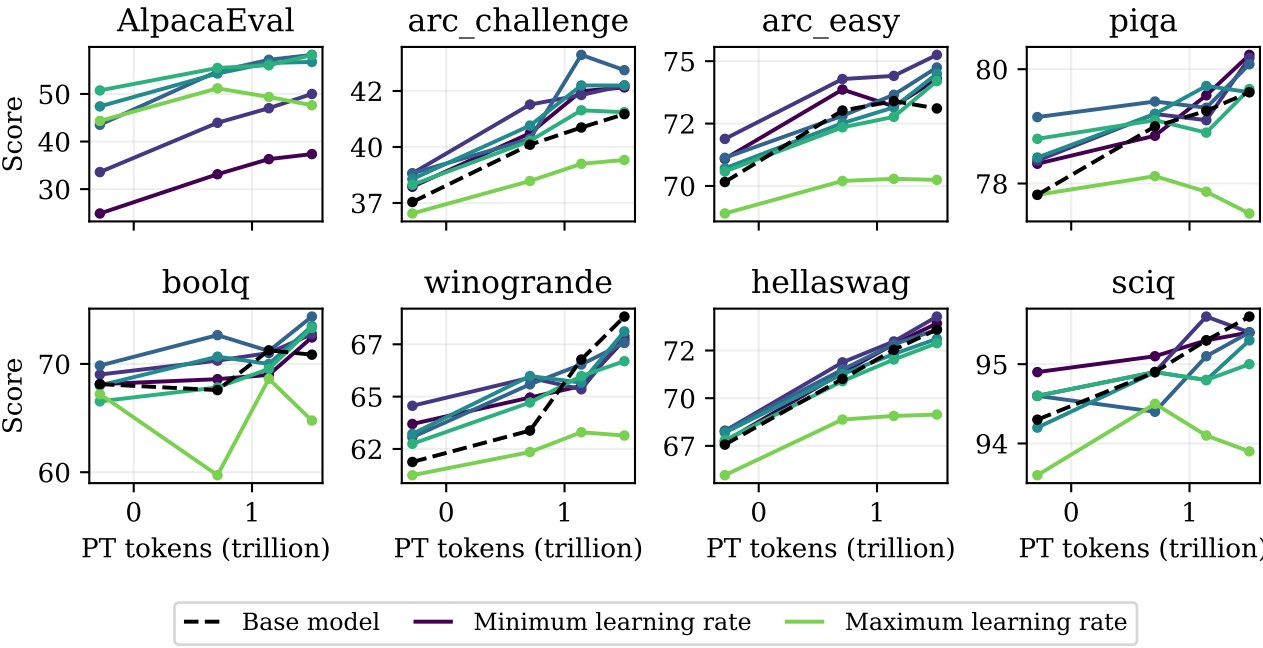

*Figure 16.* **Evaluation LLM360-7B post-trained on Anthropic-HH as a function of the number of pre-trained tokens, for all learning rates.** We report the scores on eight different datasets: AlpacaEval is considered to be the main evaluation of interest (corresponding with the downstream performance), and the other datasets are considered out-of-distribution (corresponding with the generalist performance). We use the intermediate checkpoints from Table 1 for the evaluation. We also compare to the base model (dashed line). This figure is similar to Figure 15, except we plot every learning rate, with a line representing a fixed learning rate.

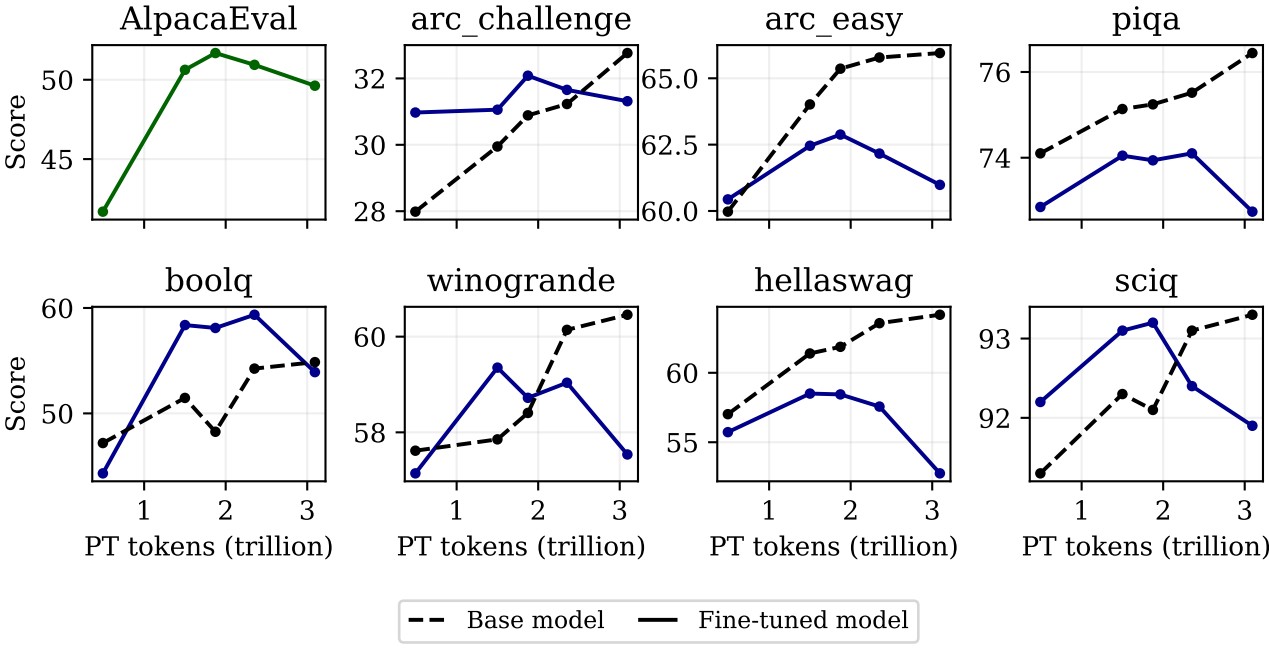

Figure 17. **Evaluation OLMo-1B post-trained on TULU as a function of the number of pre-trained tokens, with tuned learning rates.** We report the scores on eight different datasets: AlpacaEval is considered to be the main evaluation of interest (corresponding with the downstream performance), and the other datasets are considered out-of-distribution (corresponding with the generalist performance). We use the intermediate checkpoints from Table 1 for the evaluation. We tune the learning rate for each checkpoint to maximize the main evaluation (AlpacaEval). This figure is analogous to Figure 2.

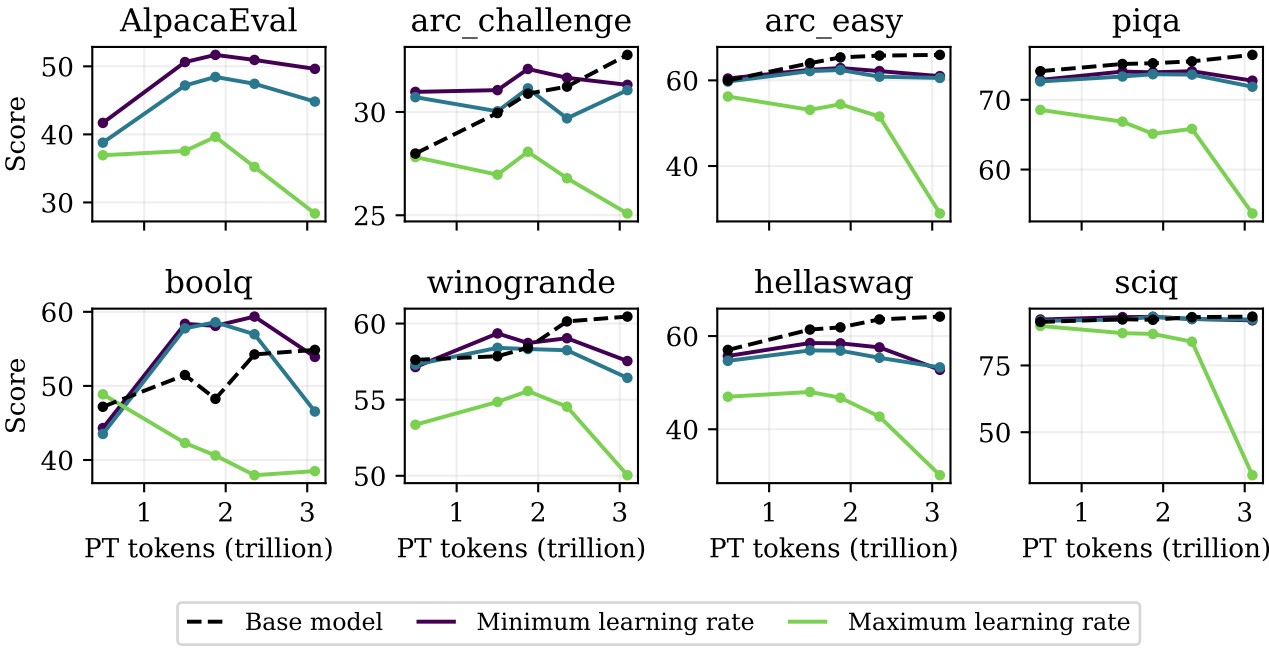

*Figure 18.* **Evaluation OLMo-1B post-trained on TULU as a function of the number of pre-trained tokens, for all learning rates.** We report the scores on eight different datasets: AlpacaEval is considered to be the main evaluation of interest (corresponding with the downstream performance), and the other datasets are considered out-of-distribution (corresponding with the generalist performance). We use the intermediate checkpoints from Table 1 for the evaluation. We also compare to the base model (dashed line). This figure is similar to Figure 17, except we plot every learning rate, with a line representing a fixed learning rate.

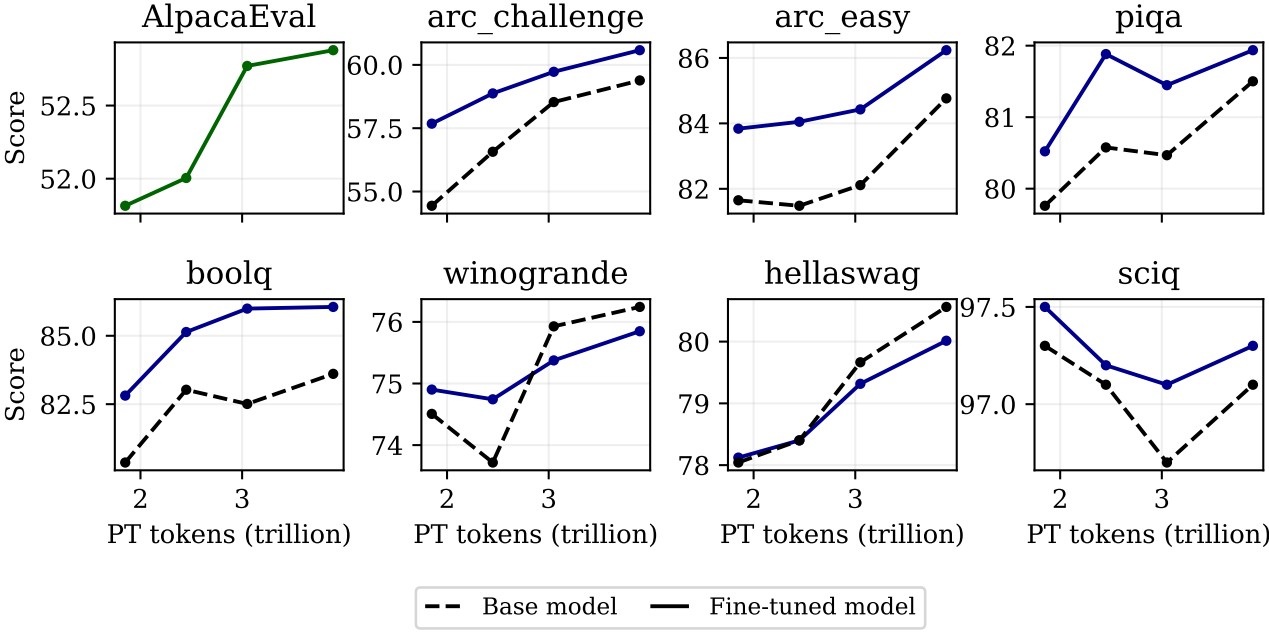

*Figure 19.* **Evaluation OLMo-2-7B post-trained on TULU as a function of the number of pre-trained tokens, with tuned learning rates.** We report the scores on eight different datasets: AlpacaEval is considered to be the main evaluation of interest (corresponding with the downstream performance), and the other datasets are considered out-of-distribution (corresponding with the generalist performance). We use the intermediate checkpoints from Table 1 for the evaluation. We tune the learning rate for each checkpoint to maximize the main evaluation (AlpacaEval). This figure is analogous to Figure 2.

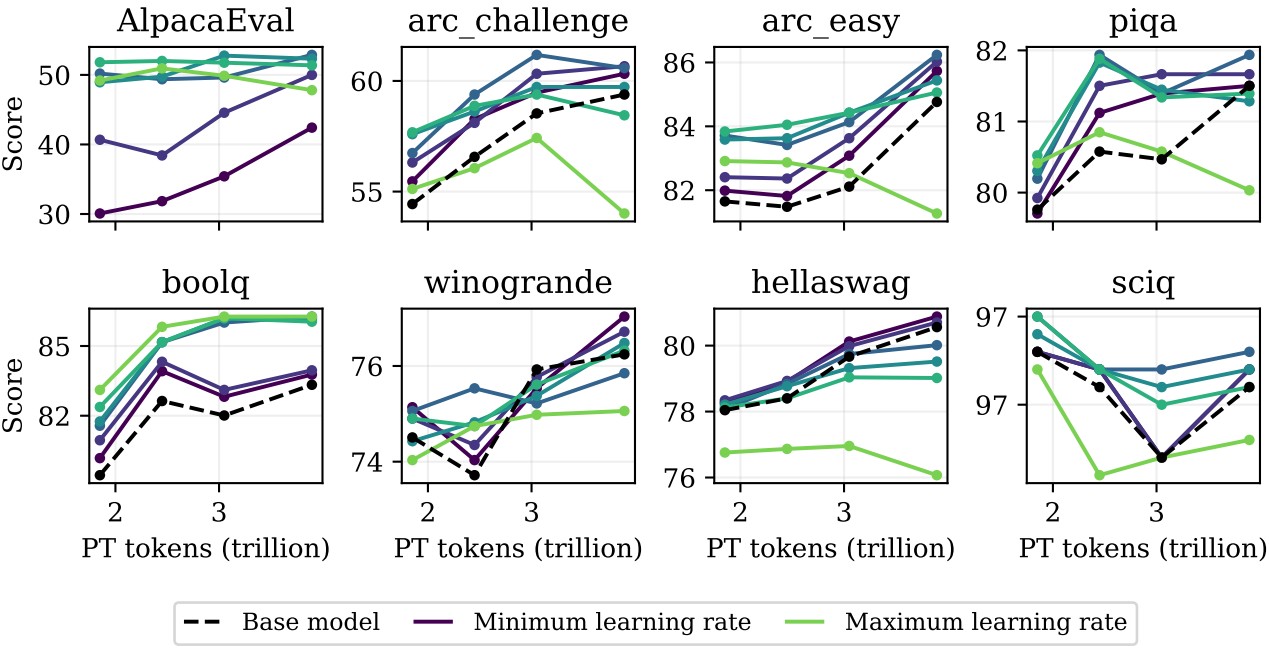

*Figure 20.* **Evaluation OLMo-2-7B post-trained on TULU as a function of the number of pre-trained tokens, for all learning rates.** We report the scores on eight different datasets: AlpacaEval is considered to be the main evaluation of interest (corresponding with the downstream performance), and the other datasets are considered out-of-distribution (corresponding with the generalist performance). We use the intermediate checkpoints from Table 1 for the evaluation. We also compare to the base model (dashed line). This figure is similar to Figure 19, except we plot every learning rate, with a line representing a fixed learning rate.

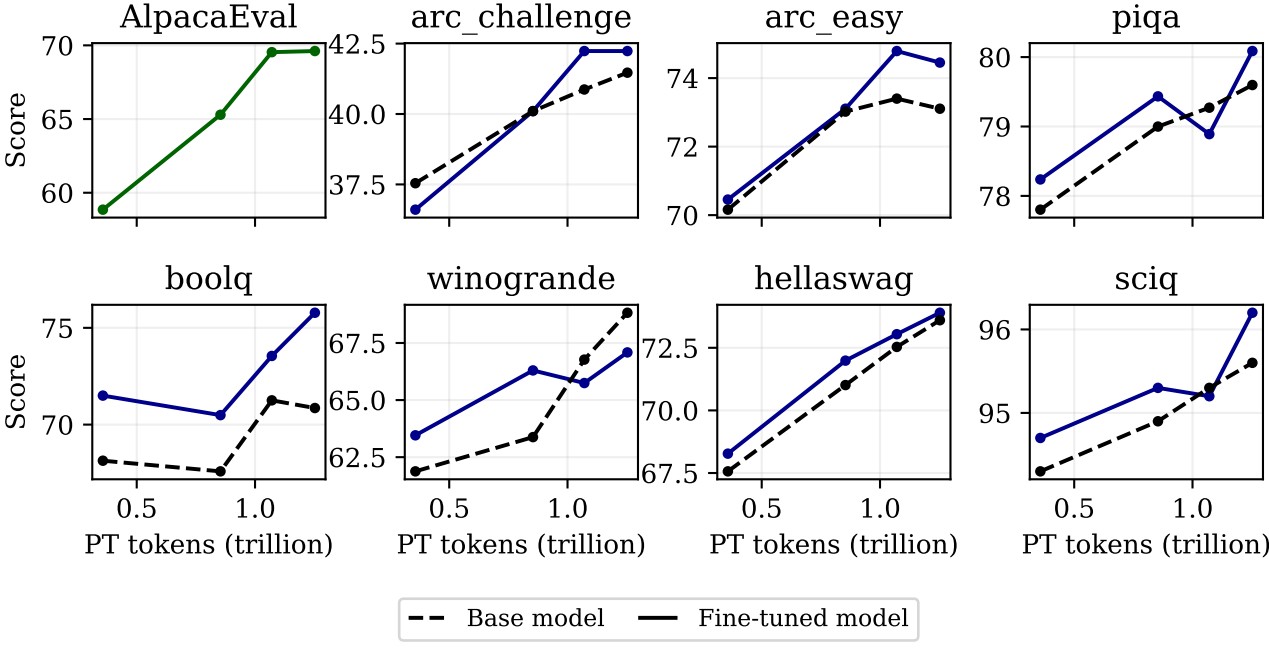

*Figure 21.* **Evaluation LLM360-7B post-trained on TULU as a function of the number of pre-trained tokens, with tuned learning rates.** We report the scores on eight different datasets: AlpacaEval is considered to be the main evaluation of interest (corresponding with the downstream performance), and the other datasets are considered out-of-distribution (corresponding with the generalist performance). We use the intermediate checkpoints from Table 1 for the evaluation. We tune the learning rate for each checkpoint to maximize the main evaluation (AlpacaEval). This figure is analogous to Figure 2.

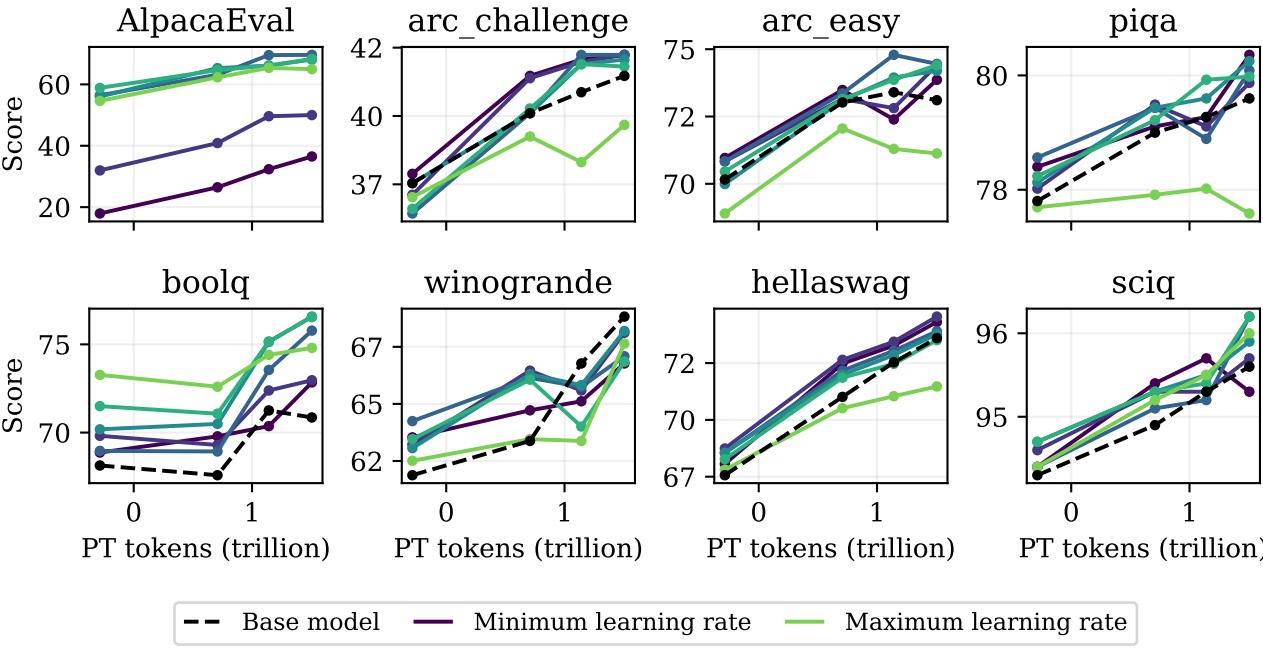

*Figure 22.* **Evaluation LLM360-7B post-trained on TULU as a function of the number of pre-trained tokens, for all learning rates.**
We report the scores on eight different datasets: AlpacaEval is considered to be the main evaluation of interest (corresponding with the
downstream performance), and the other datasets are considered out-of-distribution (corresponding with the generalist performance). We
use the intermediate checkpoints from Table 1 for the evaluation. We also compare to the base model (dashed line). This figure is similar
to Figure 21, except we plot every learning rate, with a line representing a fixed learning rate.

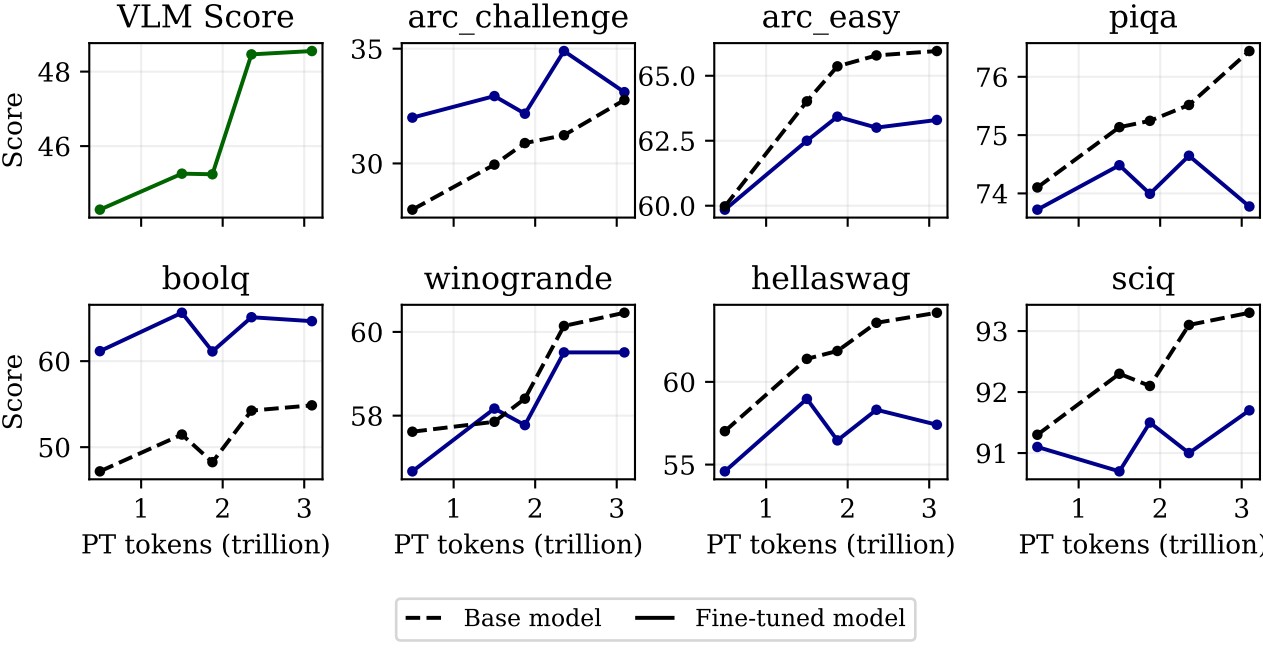

*Figure 23.* **Evaluation OLMo-1B post-trained on VLM as a function of the number of pre-trained tokens, with tuned learning rates.** We report the scores on eight different datasets: VLM Score is considered to be the main evaluation of interest (corresponding with the downstream performance), and the other datasets are considered out-of-distribution (corresponding with the generalist performance). We use the intermediate checkpoints from Table 1 for the evaluation. We tune the learning rate for each checkpoint to maximize the main evaluation (VLM Score). This figure is analogous to Figure 2.

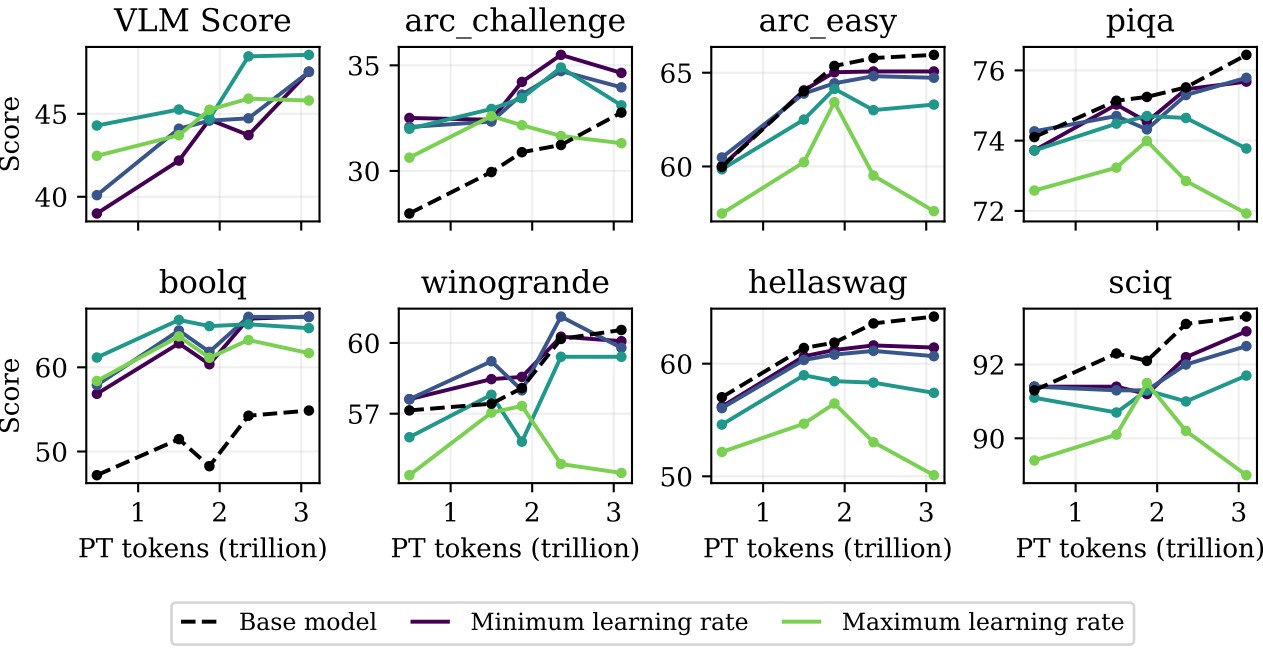

*Figure 24.* **Evaluation OLMo-1B post-trained on VLM as a function of the number of pre-trained tokens, for all learning rates.** We report the scores on eight different datasets: VLM Score is considered to be the main evaluation of interest (corresponding with the downstream performance), and the other datasets are considered out-of-distribution (corresponding with the generalist performance). We use the intermediate checkpoints from Table 1 for the evaluation. We also compare to the base model (dashed line). This figure is similar to Figure 23, except we plot every learning rate, with a line representing a fixed learning rate.

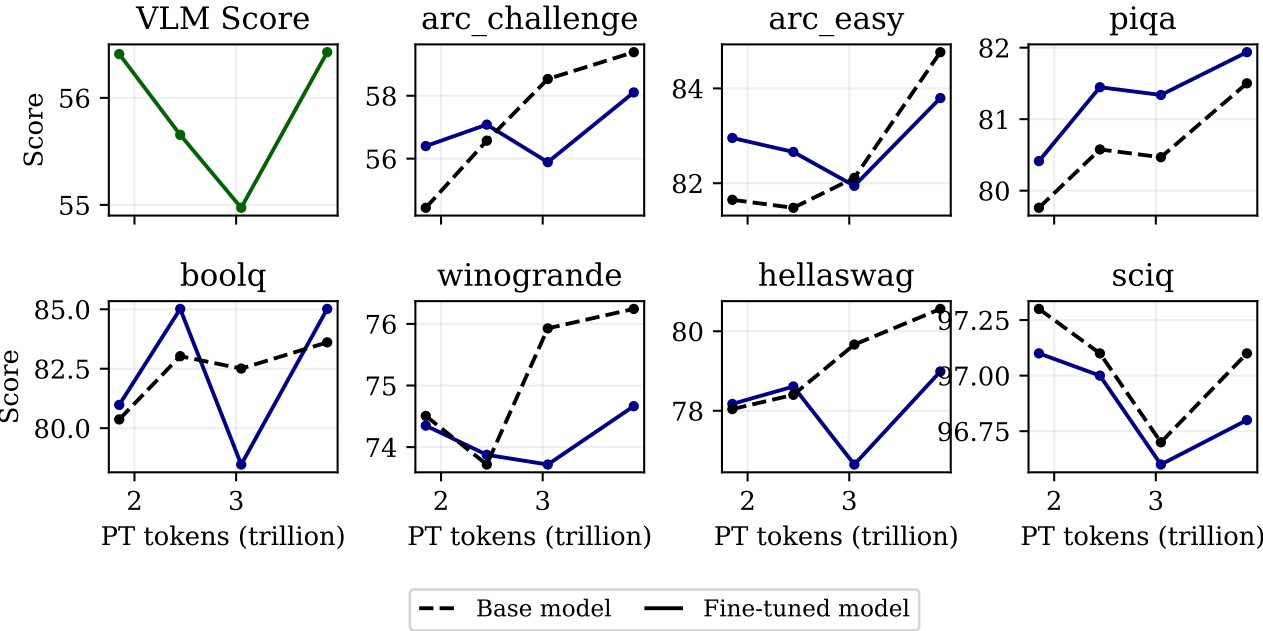

*Figure 25.* **Evaluation OLMo-2-7B post-trained on VLM as a function of the number of pre-trained tokens, with tuned learning rates.** We report the scores on eight different datasets: VLM Score is considered to be the main evaluation of interest (corresponding with the downstream performance), and the other datasets are considered out-of-distribution (corresponding with the generalist performance). We use the intermediate checkpoints from Table 1 for the evaluation. We tune the learning rate for each checkpoint to maximize the main evaluation (VLM Score). This figure is analogous to Figure 2.

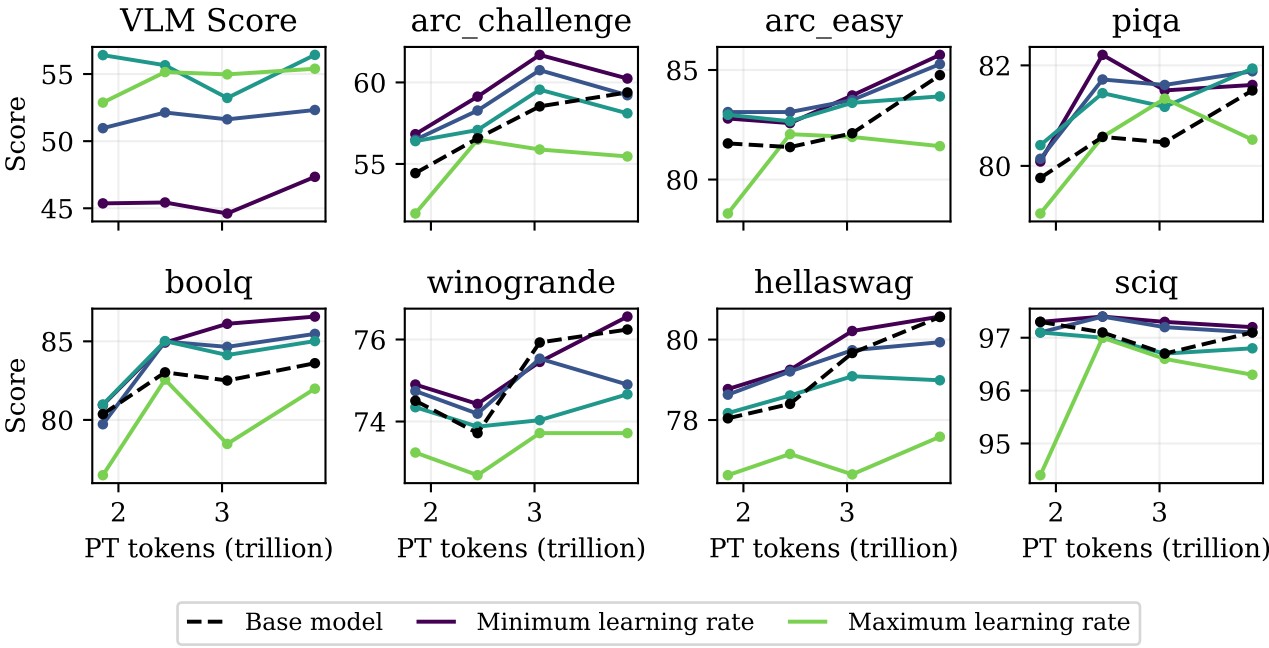

*Figure 26.* **Evaluation OLMo-2-7B post-trained on VLM as a function of the number of pre-trained tokens, for all learning rates.** We report the scores on eight different datasets: VLM Score is considered to be the main evaluation of interest (corresponding with the downstream performance), and the other datasets are considered out-of-distribution (corresponding with the generalist performance). We use the intermediate checkpoints from Table 1 for the evaluation. We also compare to the base model (dashed line). This figure is similar to Figure 25, except we plot every learning rate, with a line representing a fixed learning rate.

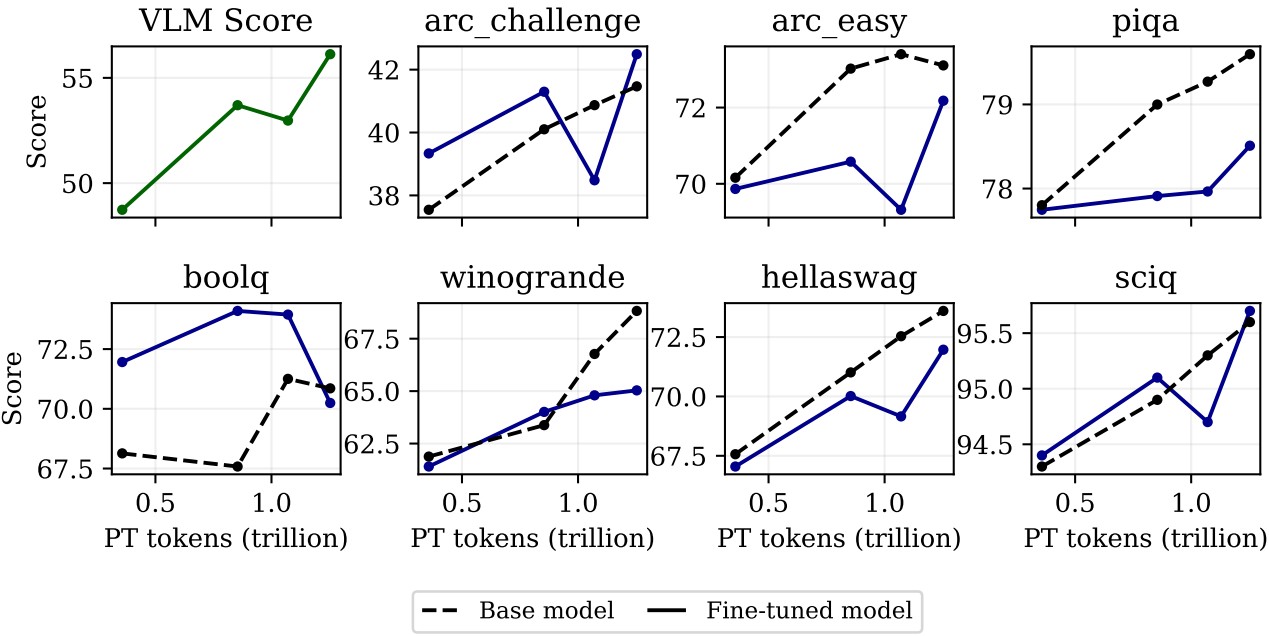

*Figure 27.* **Evaluation LLM360-7B post-trained on VLM as a function of the number of pre-trained tokens, with tuned learning rates.** We report the scores on eight different datasets: VLM Score is considered to be the main evaluation of interest (corresponding with the downstream performance), and the other datasets are considered out-of-distribution (corresponding with the generalist performance). We use the intermediate checkpoints from Table 1 for the evaluation. We tune the learning rate for each checkpoint to maximize the main evaluation (VLM Score). This figure is analogous to Figure 2.

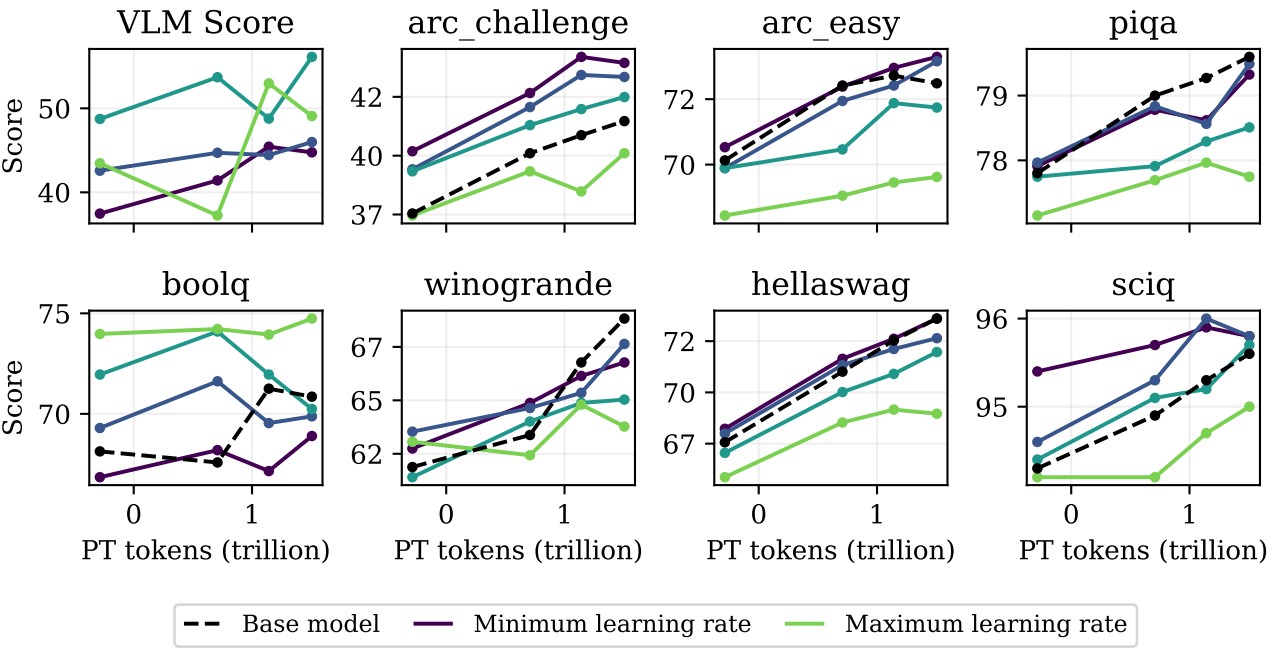

*Figure 28.* **Evaluation LLM360-7B post-trained on VLM as a function of the number of pre-trained tokens, for all learning rates.** We report the scores on eight different datasets: VLM Score is considered to be the main evaluation of interest (corresponding with the downstream performance), and the other datasets are considered out-of-distribution (corresponding with the generalist performance). We use the intermediate checkpoints from Table 1 for the evaluation. We also compare to the base model (dashed line). This figure is similar to Figure 27, except we plot every learning rate, with a line representing a fixed learning rate.

# G. Omitted Figures from Section 3: Controlled Experiments

In this section, we provide the omitted figures from Section 3 that show the results of the extended controlled experiments.

## G.1. Sensitivity

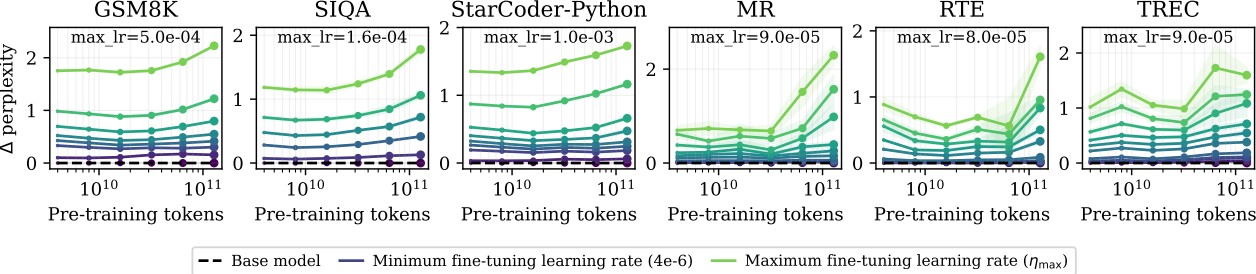

*Figure 29.* **Sensitivity of fine-tuned models with fixed learning rate in our controlled setup.** This figure is analogous to Figure 5 from the main paper, but plots the difference in perplexity between the fine-tuned model and the base model for OLMo-30M. This figure illustrates that sensitivity increases progressively throughout training.

To supplement Figure 6 from the main paper, we plot the sensitivity of fine-tuned models with fixed learning rate in our controlled setup as a function of the number of pre-training tokens in Figure 29. We find, across all datasets, that sensitivity progressively increases throughout training. Since this figure is sufficiently similar to Figure 6, we omit the corresponding sensitivity figures for the other settings we consider.

## G.2. Extended fine-tuning experiments.

We now plot the extended fine-tuning experiments. We ablate the batch size, learning rate scheduler, and model size. Table 6 provides a reference to the figures that show the results of the extended controlled experiments.

| Setting | Pre-training perplexity | Fine-tuning perplexity | Tuned pre-training perplexity | Tuned fine-tuning perplexity | Optimal LR |
|---|---|---|---|---|---|
| Batch size: 256 | Figure 30 | Figure 31 | Figure 32 | Figure 33 | Figure 34 |
| Batch size: 32 | Figure 35 | Figure 36 | Figure 37 | Figure 38 | Figure 39 |
| LR schedule: Constant | Figure 40 | Figure 41 | Figure 42 | Figure 43 | Figure 44 |
| LR schedule: constant with warmup | Figure 45 | Figure 46 | Figure 47 | Figure 48 | Figure 49 |
| OLMo-15M | Figure 50 | Figure 51 | Figure 52 | Figure 53 | Figure 54 |
| OLMo-30M (extended) | Figure 55 | Figure 56 | Figure 57 | Figure 58 | Figure 59 |
| OLMo-90M | Figure 60 | Figure 61 | Figure 62 | Figure 63 | Figure 64 |

*Table 6.* **Table of contents for extended experimental settings.** This table provides a reference to the figures that show the results of the extended controlled experiments.

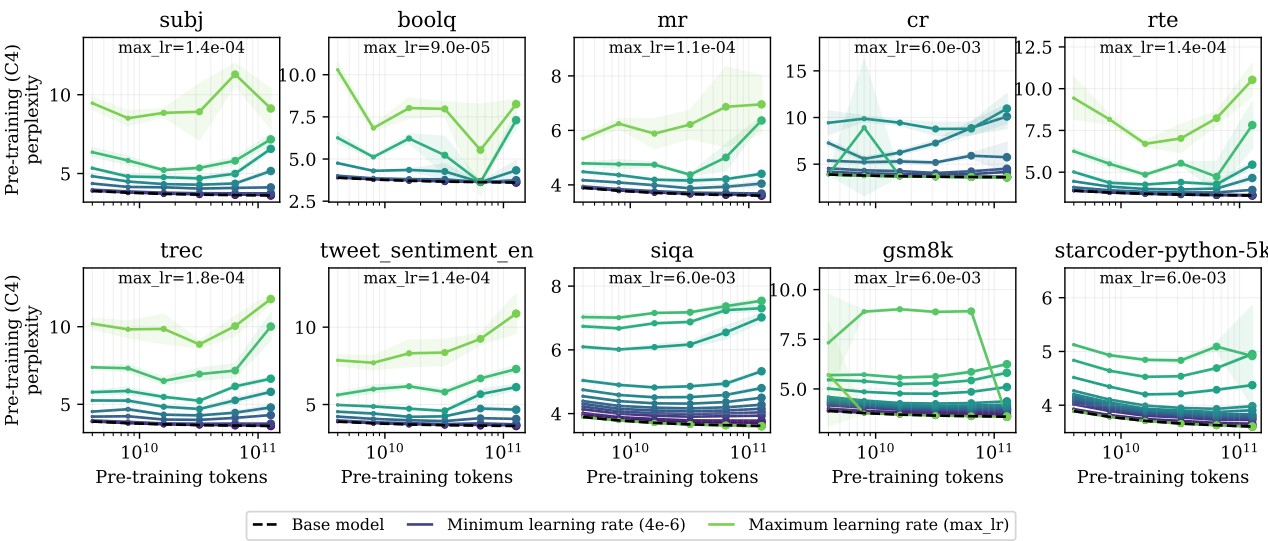

*Figure 30.* **Pre-training perplexity after fine-tuning as a function of the pre-training budget using the configuration specified in Table 3 but with batch size 256 for the OLMo-30M model.** Each connected line reflects a series of models trained with fixed hyperparameters.

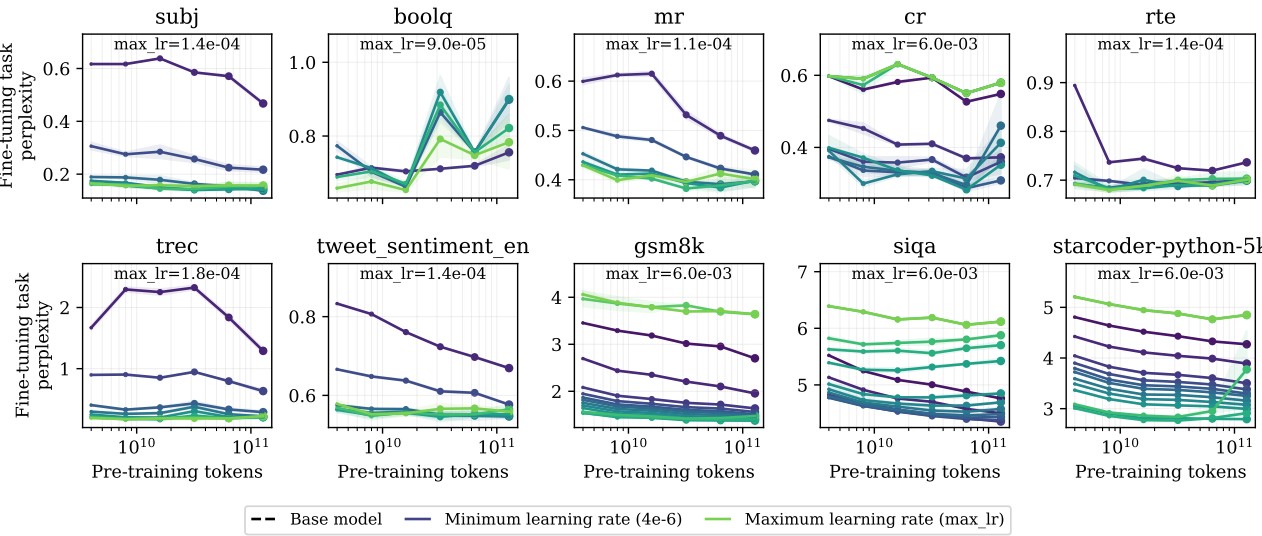

*Figure 31.* **Fine-tuning perplexity after fine-tuning as a function of the pre-training budget using the configuration specified in Table 3 but with batch size 256 for the OLMo-30M model.** Each connected line reflects a series of models trained with fixed hyperparameters.

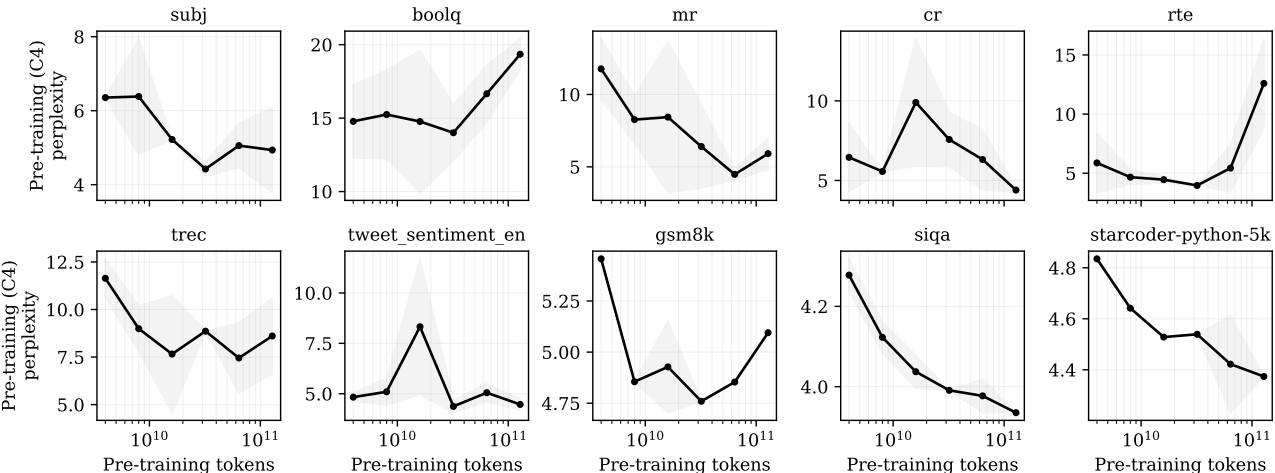

*Figure 32.* **Pre-training perplexity after fine-tuning as a function of the pre-training budget with a tuned learning rate to optimize fine-tuning performance using the configuration specified in Table 3 but with batch size 256 for the OLMo-30M model.** Similar to the untuned version but with the fine-tuning-optimal learning rate.

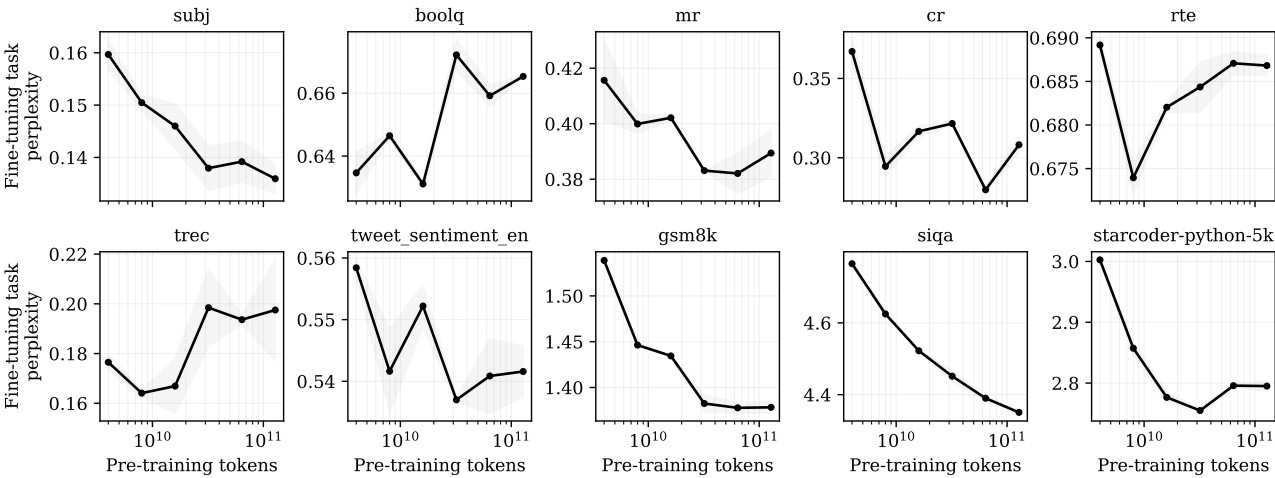

*Figure 33.* **Fine-tuning perplexity after fine-tuning as a function of the pre-training budget with a tuned learning rate to optimize fine-tuning performance using the configuration specified in Table 3 but with batch size 256 for the OLMo-30M model.** Similar to the untuned version but with the fine-tuning-optimal learning rate.

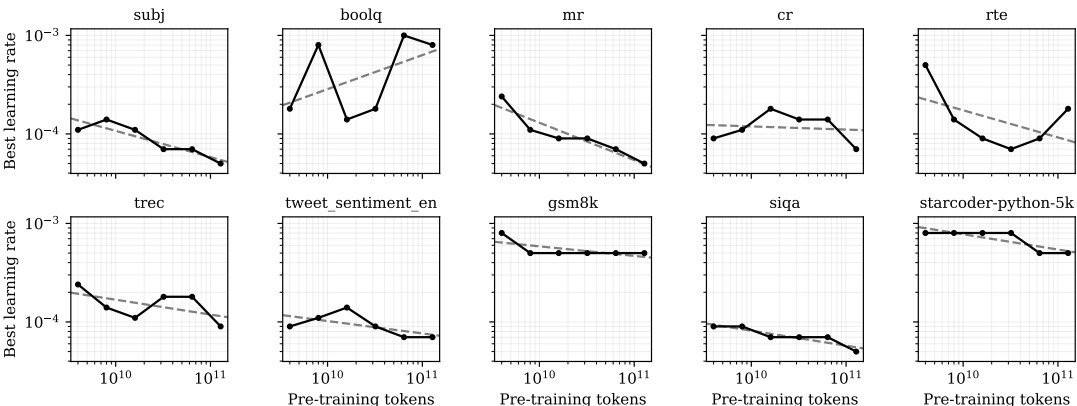

*Figure 34.* **The optimal learning rate for best fine-tuning performance as a function of the pre-training budget using the configuration specified in Table 3 but with batch size 256 for the OLMo-30M model.** The learning rate shown corresponds with those chosen in Figures 32 and 33.

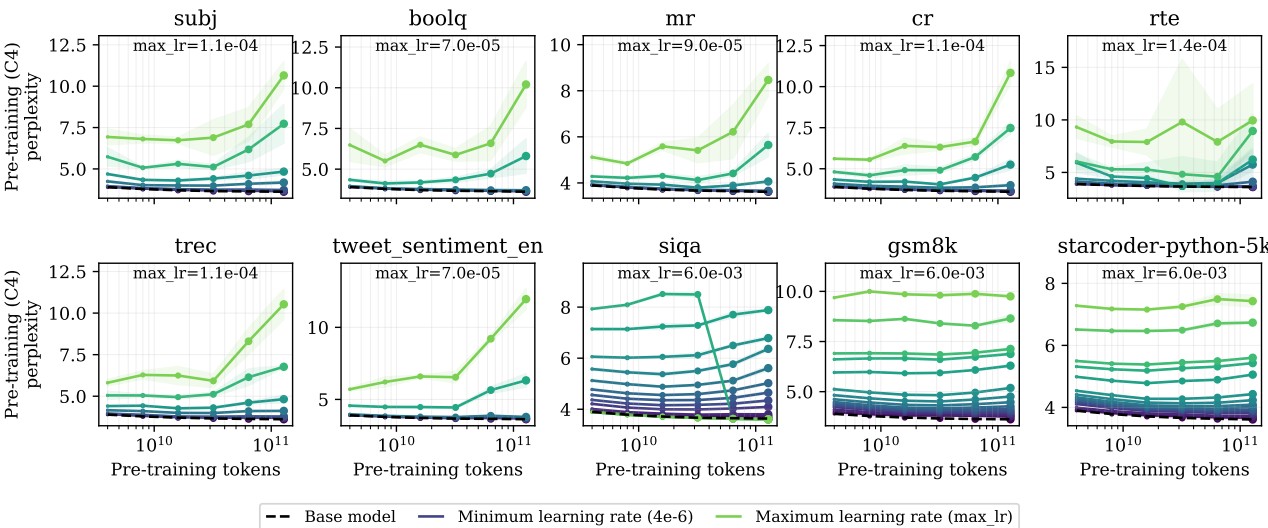

*Figure 35.* **Pre-training perplexity after fine-tuning as a function of the pre-training budget using the configuration specified in Table 3 but with batch size 32 for the OLMo-30M model.** Each connected line reflects a series of models trained with fixed hyperparameters.

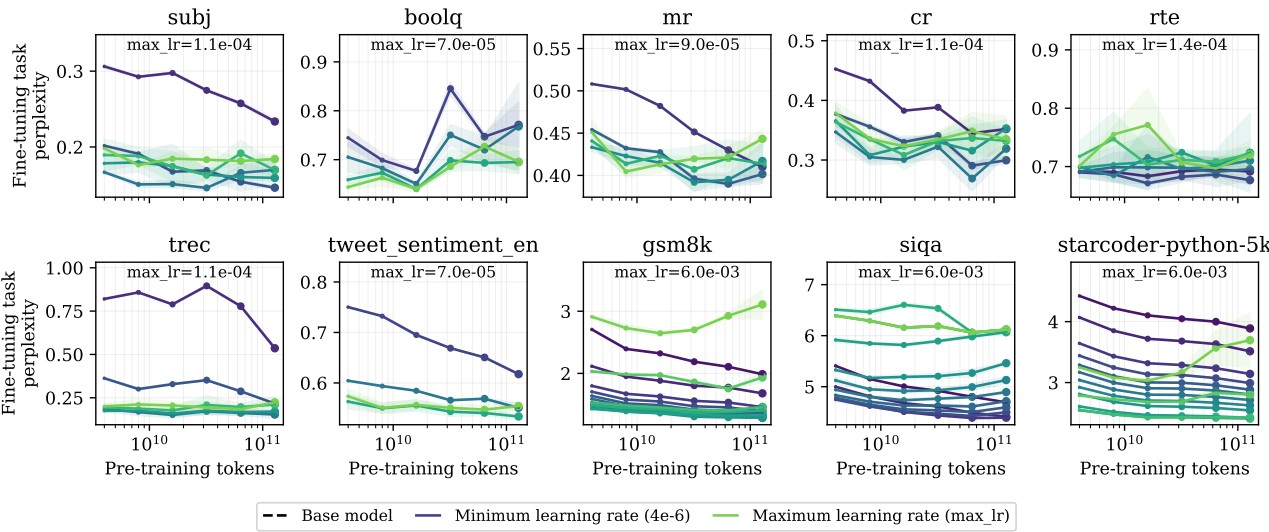

*Figure 36.* **Fine-tuning perplexity after fine-tuning as a function of the pre-training budget using the configuration specified in Table 3 but with batch size 32 for the OLMo-30M model.** Each connected line reflects a series of models trained with fixed hyperparameters.

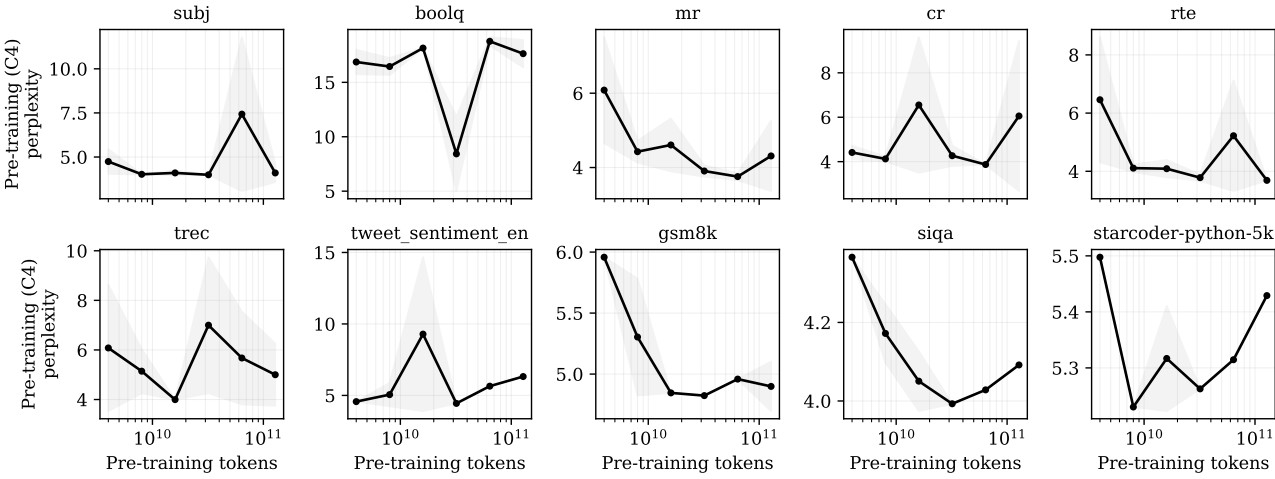

*Figure 37.* **Pre-training perplexity after fine-tuning as a function of the pre-training budget with a tuned learning rate to optimize fine-tuning performance using the configuration specified in Table 3 but with batch size 32 for the OLMo-30M model.** Similar to the untuned version but with the fine-tuning-optimal learning rate.

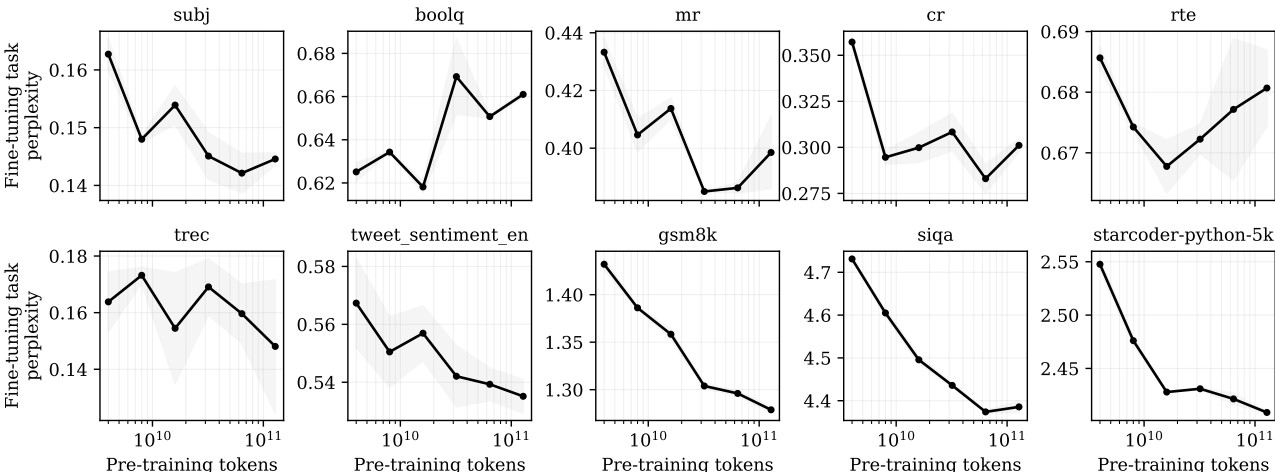

*Figure 38.* **Fine-tuning perplexity after fine-tuning as a function of the pre-training budget with a tuned learning rate to optimize fine-tuning performance using the configuration specified in Table 3 but with batch size 32 for the OLMo-30M model.** Similar to the untuned version but with the fine-tuning-optimal learning rate.

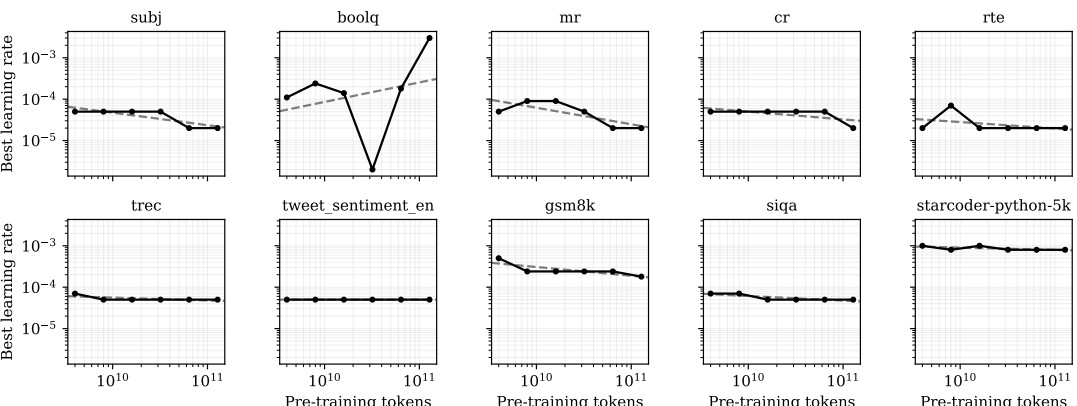

*Figure 39.* **The optimal learning rate for best fine-tuning performance as a function of the pre-training budget using the configuration specified in Table 3 but with batch size 32 for the OLMo-30M model.** The learning rate shown corresponds with those chosen in Figures 37 and 38.

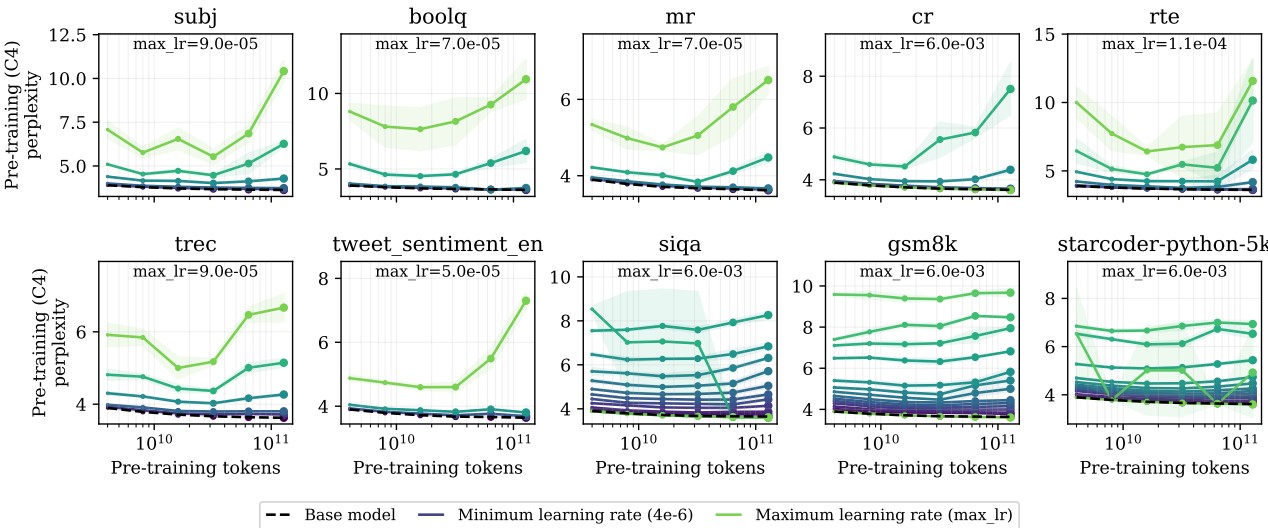

*Figure 40.* **Pre-training perplexity after fine-tuning as a function of the pre-training budget using a constant learning rate scheduler (instead of Cosine) with the configuration specified in Table 3 for the OLMo-30M model.** Each connected line reflects a series of models trained with fixed hyperparameters.

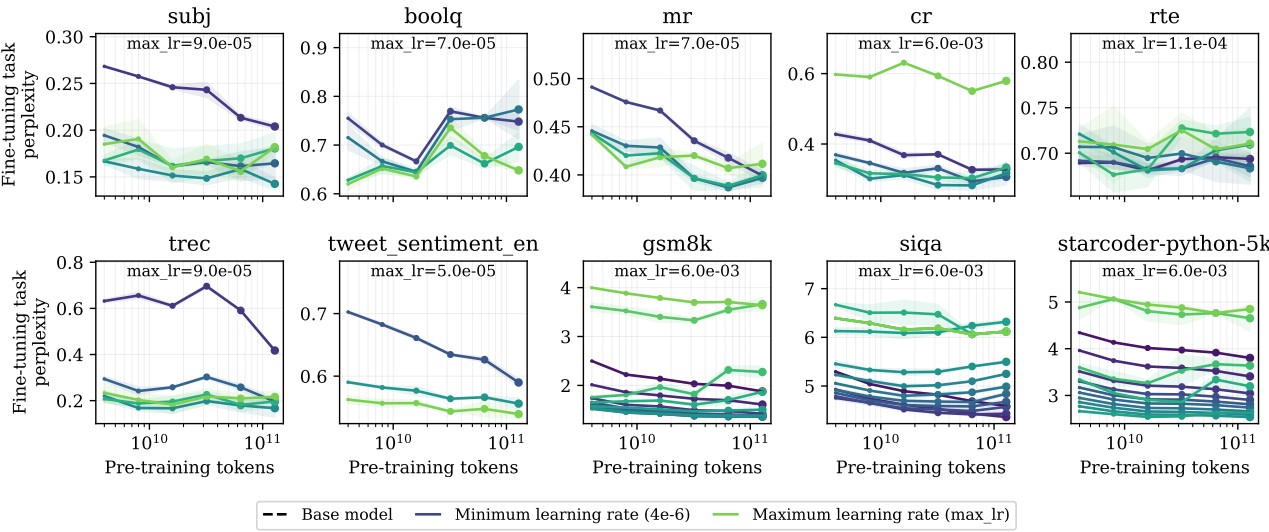

*Figure 41.* **Fine-tuning perplexity after fine-tuning as a function of the pre-training budget using a constant learning rate scheduler (instead of Cosine) with the configuration specified in Table 3 for the OLMo-30M model.** Each connected line reflects a series of models trained with fixed hyperparameters.

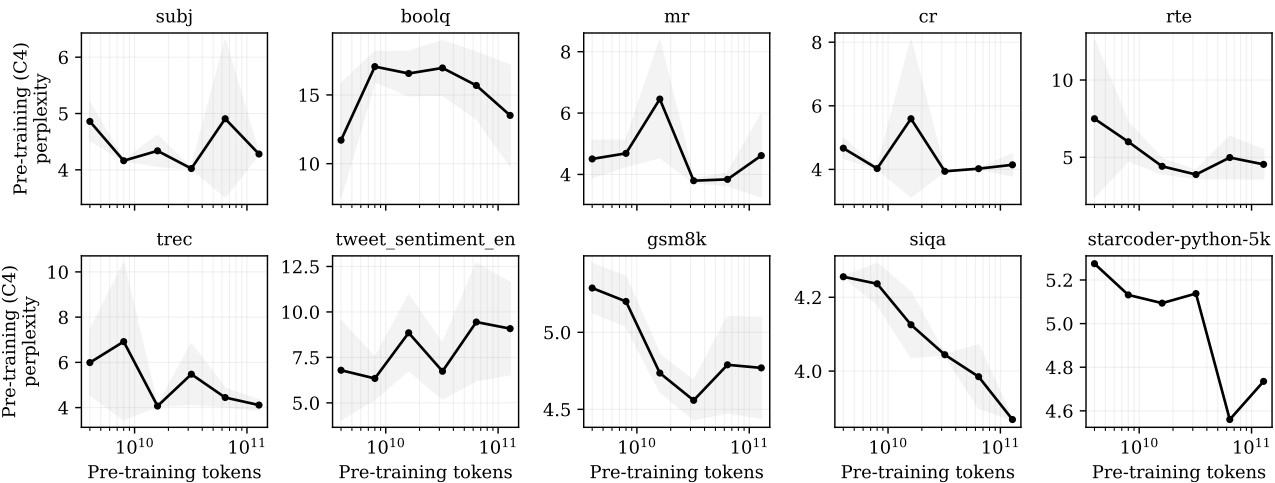

*Figure 42.* **Pre-training perplexity after fine-tuning as a function of the pre-training budget with a tuned learning rate to optimize fine-tuning performance using a constant learning rate scheduler for the OLMo-30M model.** Similar to the untuned version but showing the performance with the fine-tuning-optimal learning rate.

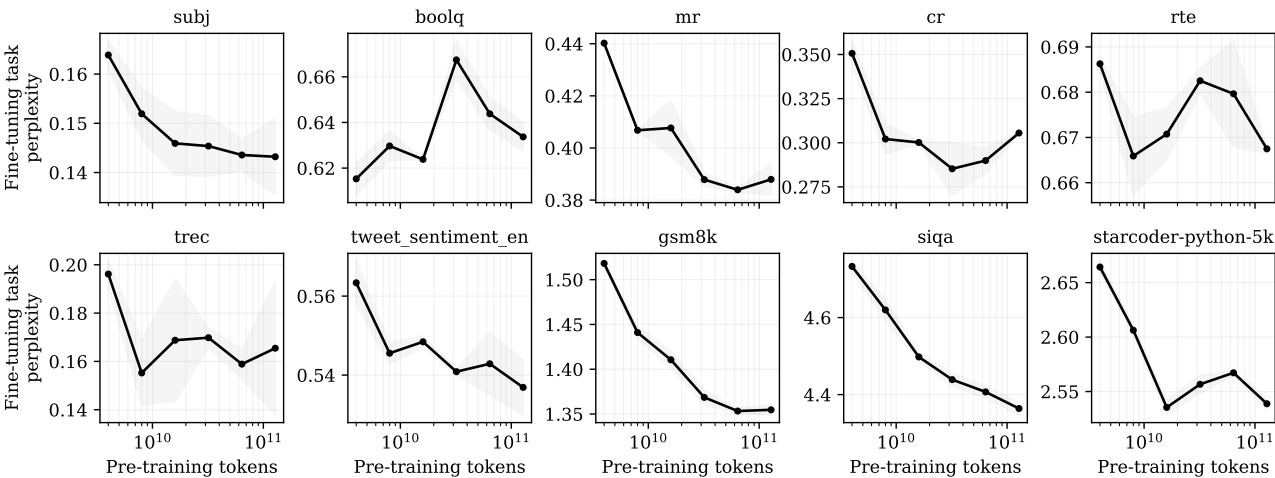

*Figure 43.* **Fine-tuning perplexity after fine-tuning as a function of the pre-training budget with a tuned learning rate to optimize fine-tuning performance using a constant learning rate scheduler for the OLMo-30M model.** Similar to the untuned version but showing the performance with the fine-tuning-optimal learning rate.

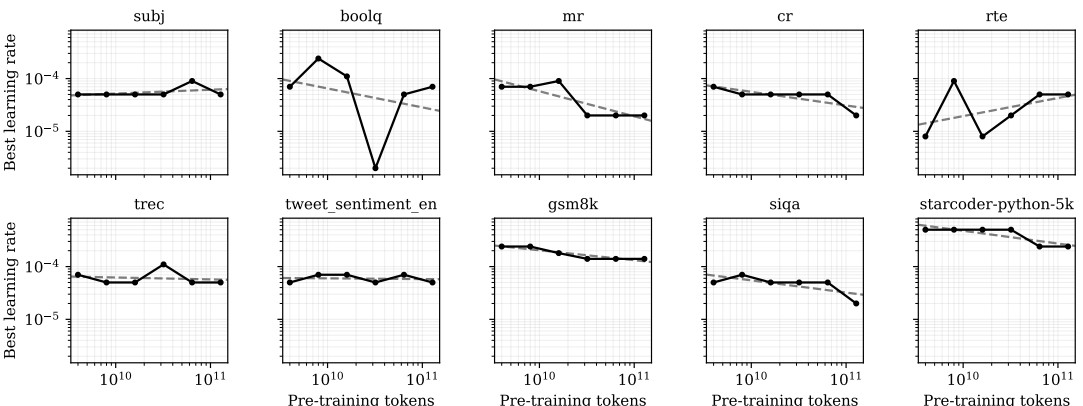

*Figure 44.* **The optimal learning rate for best fine-tuning performance as a function of the pre-training budget using a constant learning rate scheduler for the OLMo-30M model.** The learning rate shown corresponds with those chosen in Figures 42 and 43.

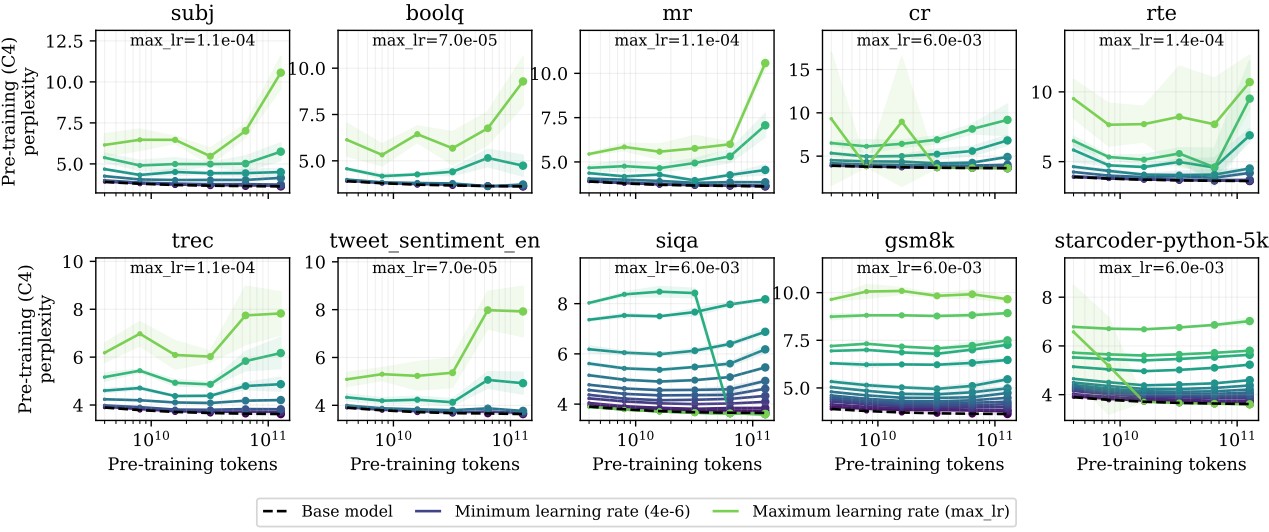

*Figure 45.* **Pre-training perplexity after fine-tuning as a function of the pre-training budget using a constant learning rate scheduler with warmup with the configuration specified in Table 3 for the OLMo-30M model.** Each connected line reflects a series of models trained with fixed hyperparameters.

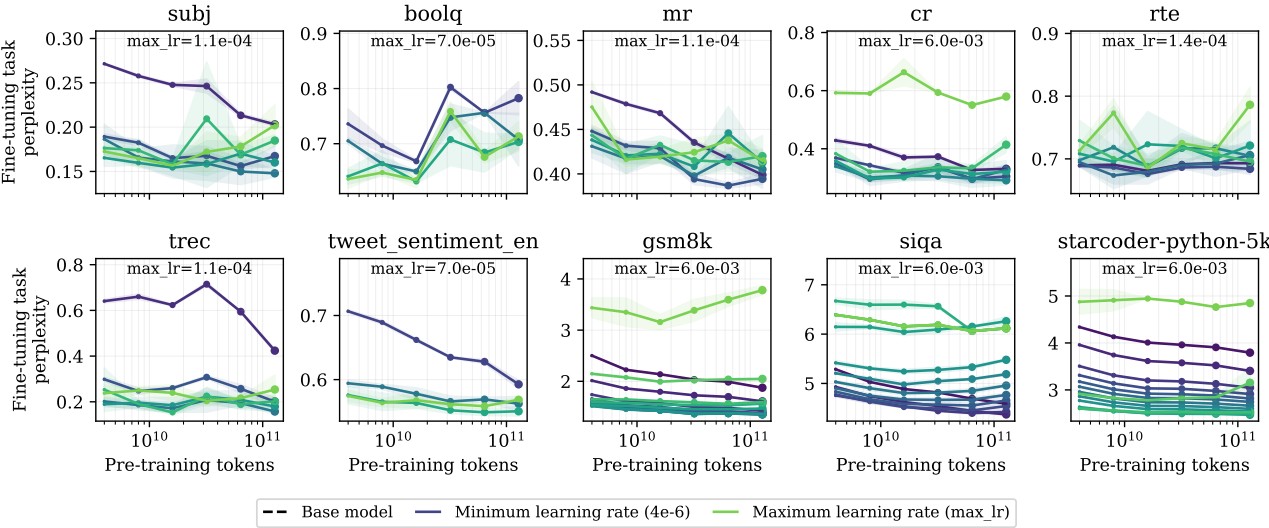

*Figure 46.* **Fine-tuning perplexity after fine-tuning as a function of the pre-training budget using a constant learning rate scheduler with warmup with the configuration specified in Table 3 for the OLMo-30M model.** Each connected line reflects a series of models trained with fixed hyperparameters.

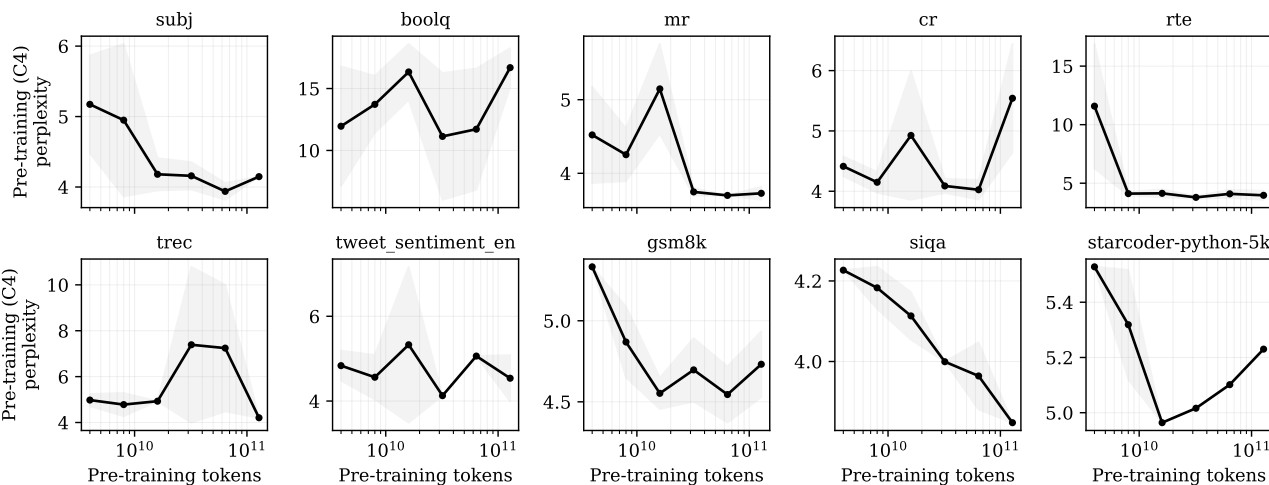

*Figure 47.* **Pre-training perplexity after fine-tuning as a function of the pre-training budget with a tuned learning rate to optimize fine-tuning performance using a constant learning rate scheduler with warmup for the OLMo-30M model.** Similar to the untuned version but showing the performance with the fine-tuning-optimal learning rate.

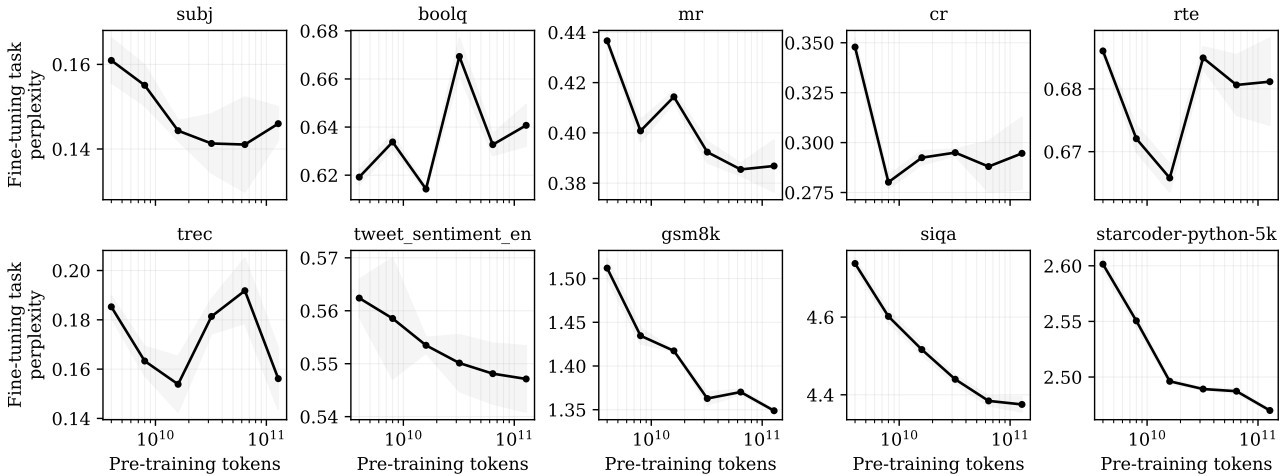

*Figure 48.* **Fine-tuning perplexity after fine-tuning as a function of the pre-training budget with a tuned learning rate to optimize fine-tuning performance using a constant learning rate scheduler with warmup for the OLMo-30M model.** Similar to the untuned version but showing the performance with the fine-tuning-optimal learning rate.

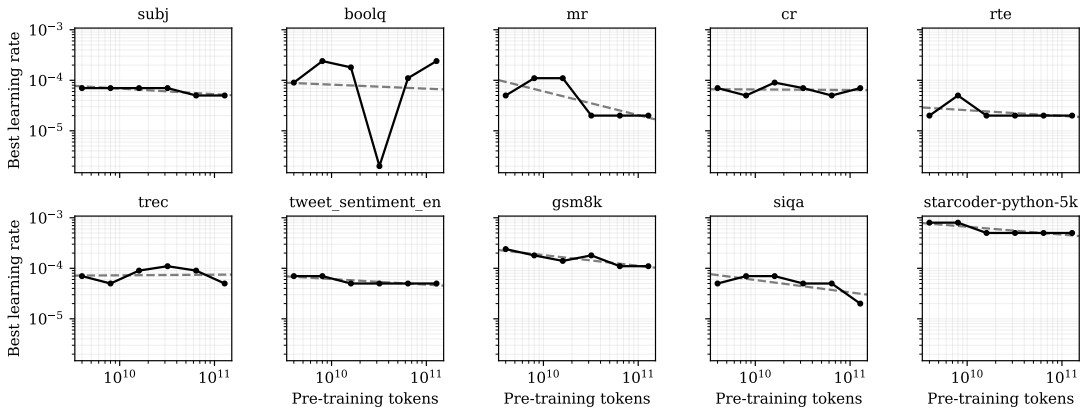

*Figure 49.* **The optimal learning rate for best fine-tuning performance as a function of the pre-training budget using a constant learning rate scheduler with warmup for the OLMo-30M model.** The learning rate shown corresponds with those chosen in Figures 47 and 48.

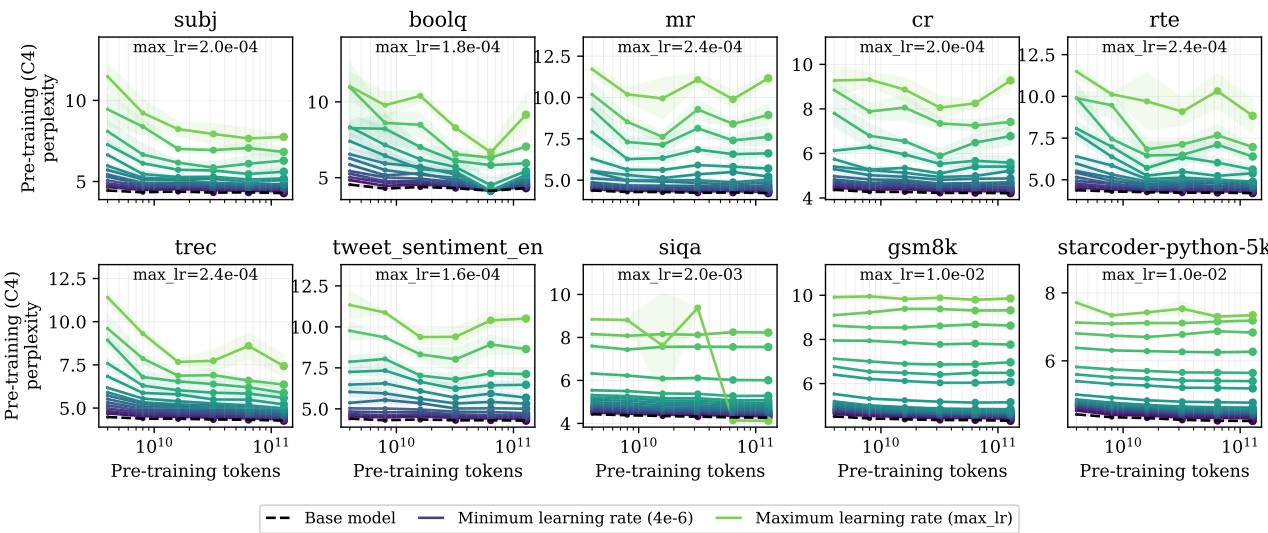

*Figure 50.* **Pre-training perplexity after fine-tuning as a function of the pre-training budget using the configuration specified in Table 3 for OLMo-15M.** Each connected line reflects a series of models trained with fixed hyperparameters. Analogous to Figure 5 (top) from the main paper.

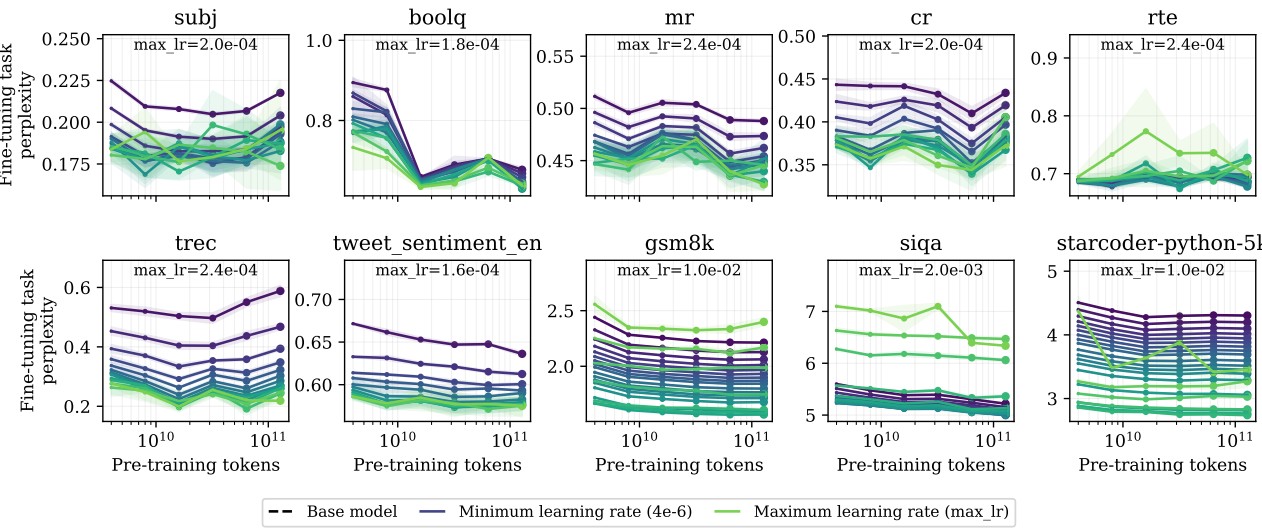

*Figure 51.* **Fine-tuning perplexity after fine-tuning as a function of the pre-training budget using the configuration specified in Table 3 for OLMo-15M.** Each connected line reflects a series of models trained with fixed hyperparameters. Analogous to Figure 5 (bottom) from the main paper.

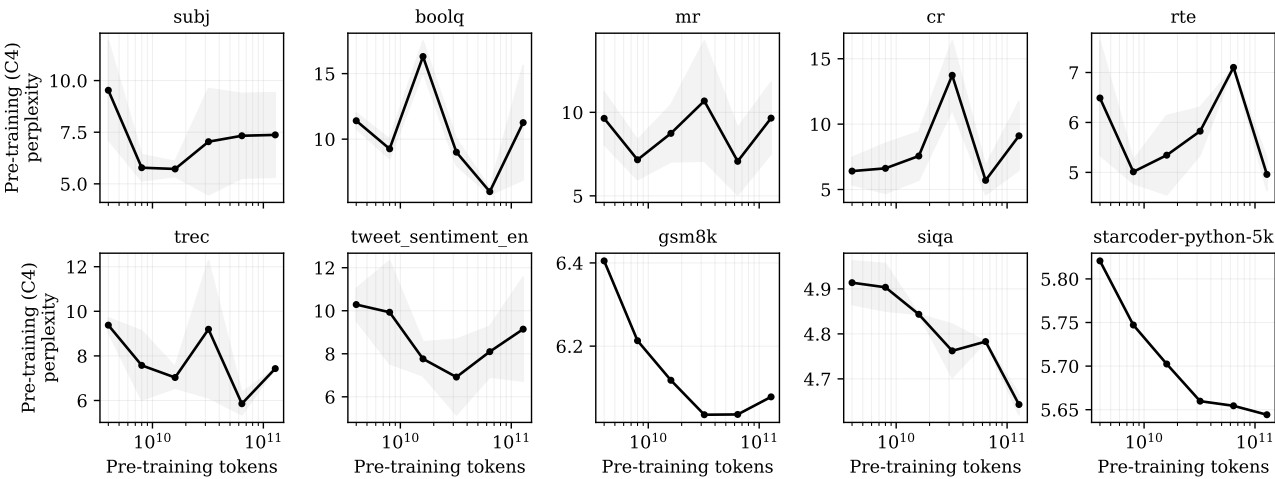

*Figure 52.* **Pre-training perplexity after fine-tuning as a function of the pre-training budget with a tuned learning rate to optimize fine-tuning performance using the configuration specified in Table 3 for OLMo-15M.** Similar to the untuned version but showing the performance obtained with the fine-tuning-optimal learning rate, analogous to Figure 6 (bottom) from the main paper.

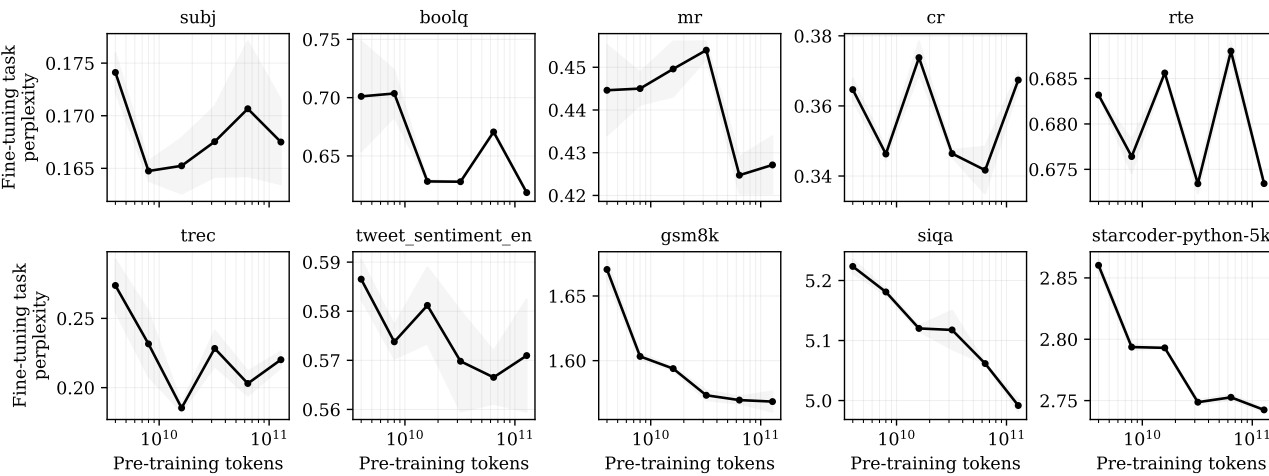

*Figure 53.* **Fine-tuning perplexity after fine-tuning as a function of the pre-training budget with a tuned learning rate to optimize fine-tuning performance using the configuration specified in Table 3 for OLMo-15M.** Similar to the untuned version but showing the performance obtained with the fine-tuning-optimal learning rate, analogous to Figure 6 (top) from the main paper.

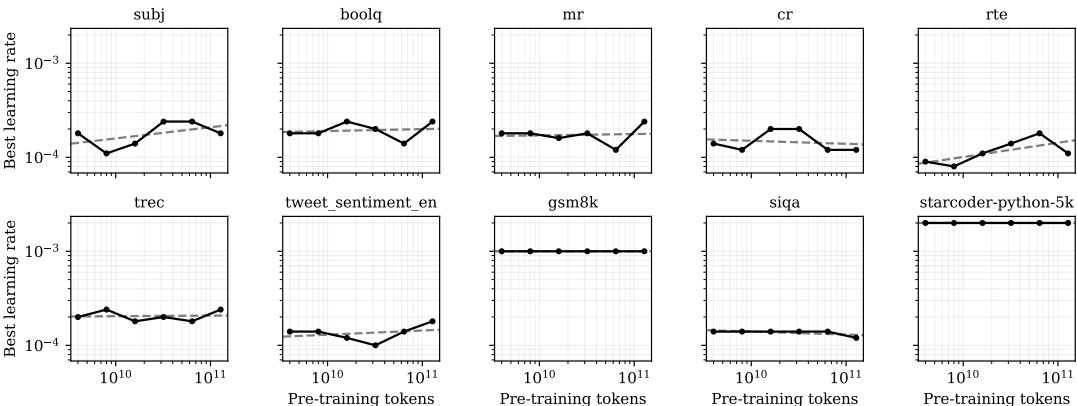

*Figure 54.* **The optimal learning rate for best fine-tuning performance as a function of the pre-training budget using the configuration specified in Table 3 for OLMo-15M.** The learning rate shown corresponds with those chosen in Figures 52 and 53.

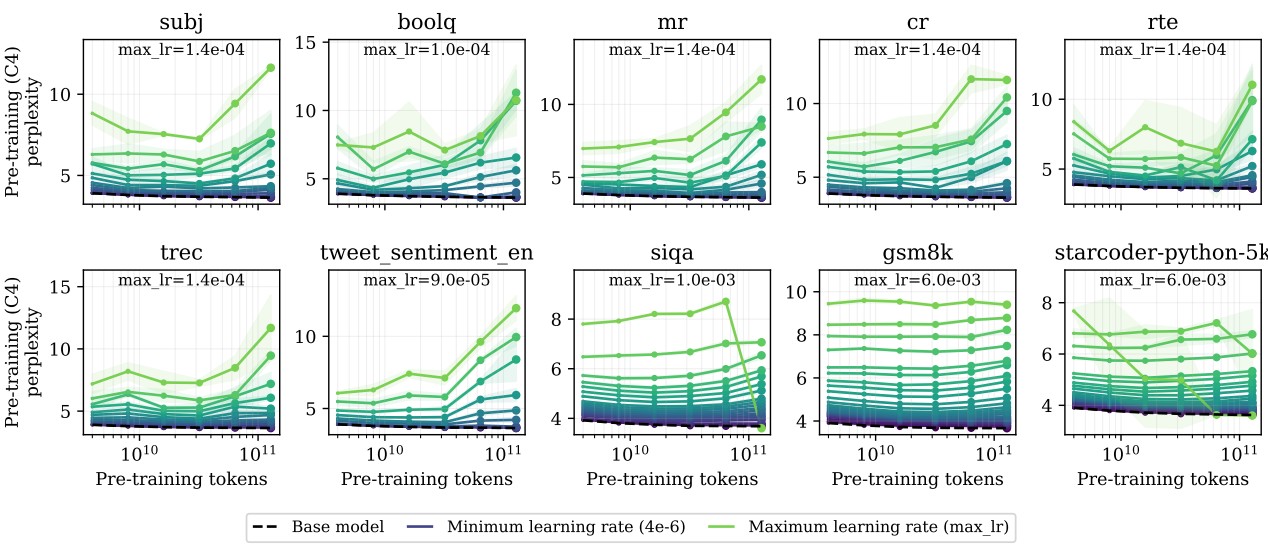

*Figure 55.* **Pre-training perplexity after fine-tuning as a function of the pre-training budget using the configuration specified in Table 3 for OLMo-30M.** Each connected line reflects a series of models trained with fixed hyperparameters. Extended version of Figure 5 (top) from the main paper.

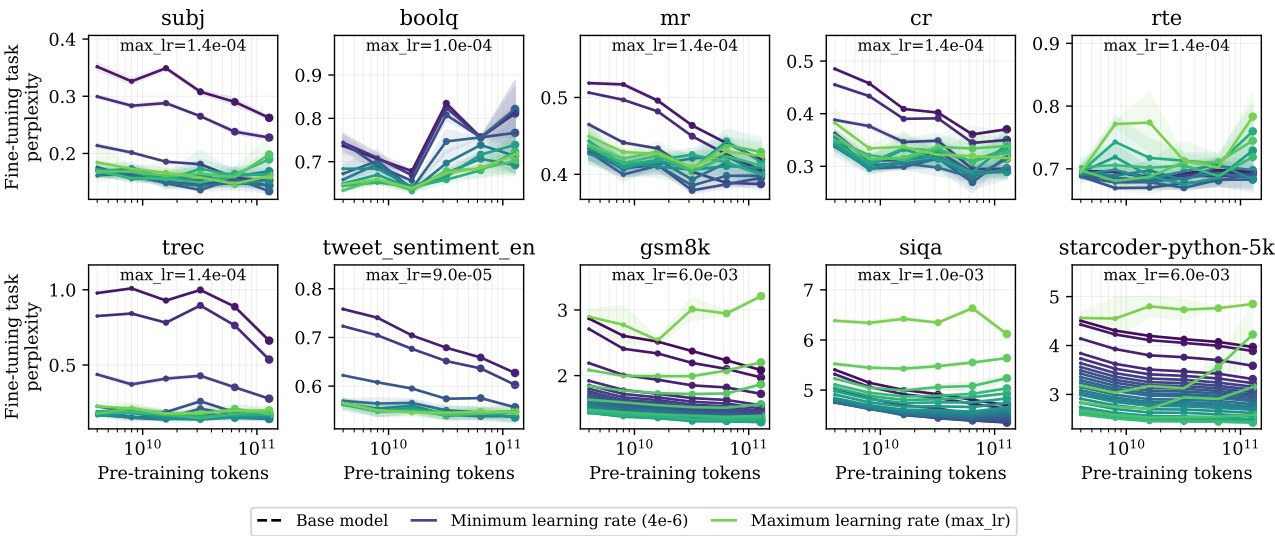

Figure 56. **Fine-tuning perplexity after fine-tuning as a function of the pre-training budget using the configuration specified in Table 3 for OLMo-30M.** Each connected line reflects a series of models trained with fixed hyperparameters. Extended version of Figure 5 (bottom) from the main paper.

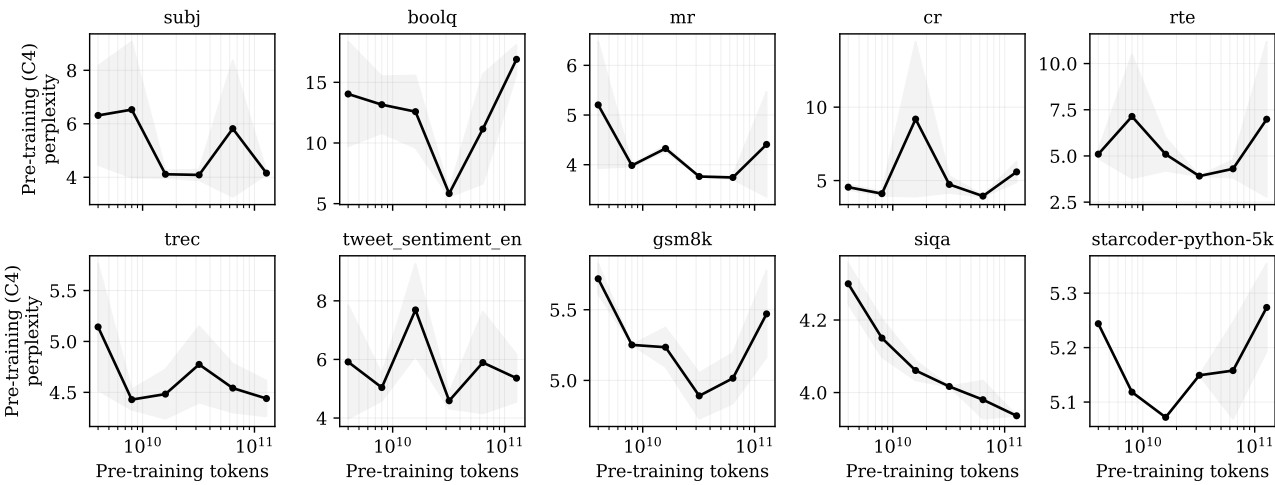

Figure 57. **Pre-training perplexity after fine-tuning as a function of the pre-training budget with a tuned learning rate to optimize fine-tuning performance using the configuration specified in Table 3 for OLMo-30M.** Similar to the untuned version but showing the performance with the fine-tuning-optimal learning rate, analogous to Figure 6 (bottom) from the main paper. Extended version of Figure 55 from the main paper.

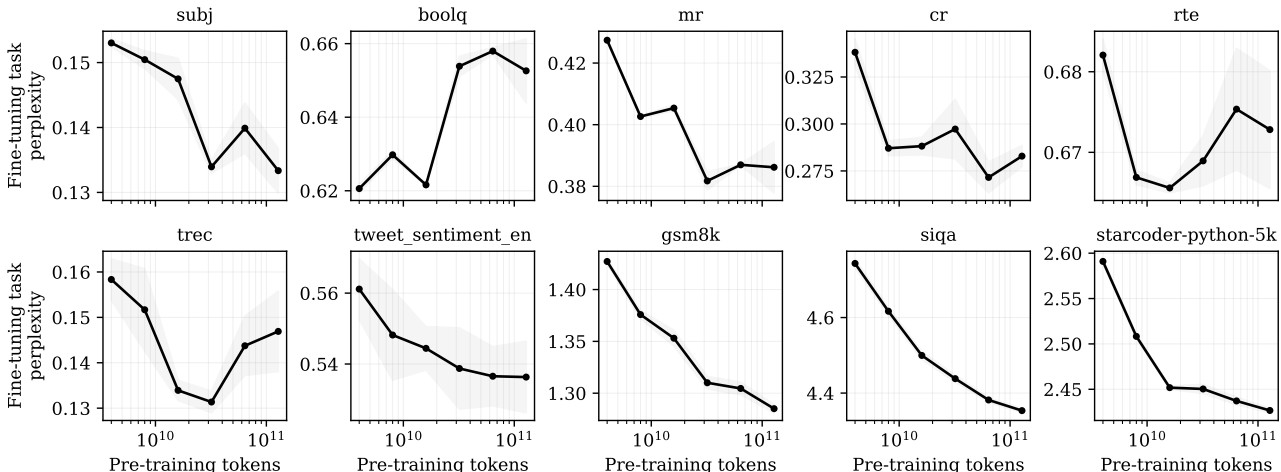

*Figure 58.* **Fine-tuning perplexity after fine-tuning as a function of the pre-training budget with a tuned learning rate to optimize fine-tuning performance using the configuration specified in Table 3 for OLMo-30M.** Similar to the untuned version but showing the performance with the fine-tuning-optimal learning rate, analogous to Figure 6 (top) from the main paper. Extended version of Figure 56 from the main paper.

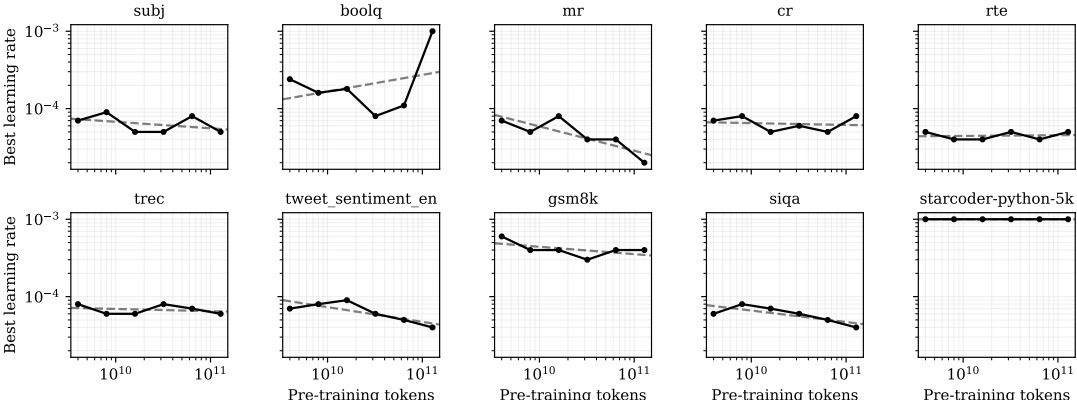

*Figure 59.* **The optimal learning rate for best fine-tuning performance as a function of the pre-training budget using the configuration specified in Table 3 for OLMo-30M.** The learning rate shown corresponds with those chosen in Figures 57 and 58.

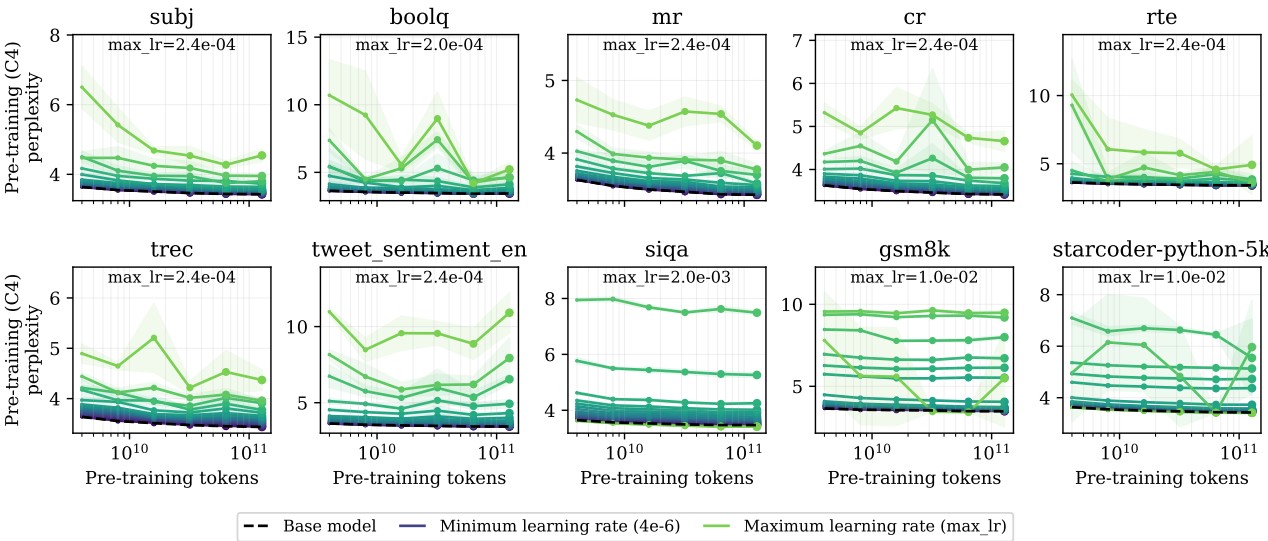

*Figure 60.* **Pre-training perplexity after fine-tuning as a function of the pre-training budget using the configuration specified in Table 3 for OLMo-90M.** Each connected line reflects a series of models trained with fixed hyperparameters. Analogous to Figure 5 (top) from the main paper.

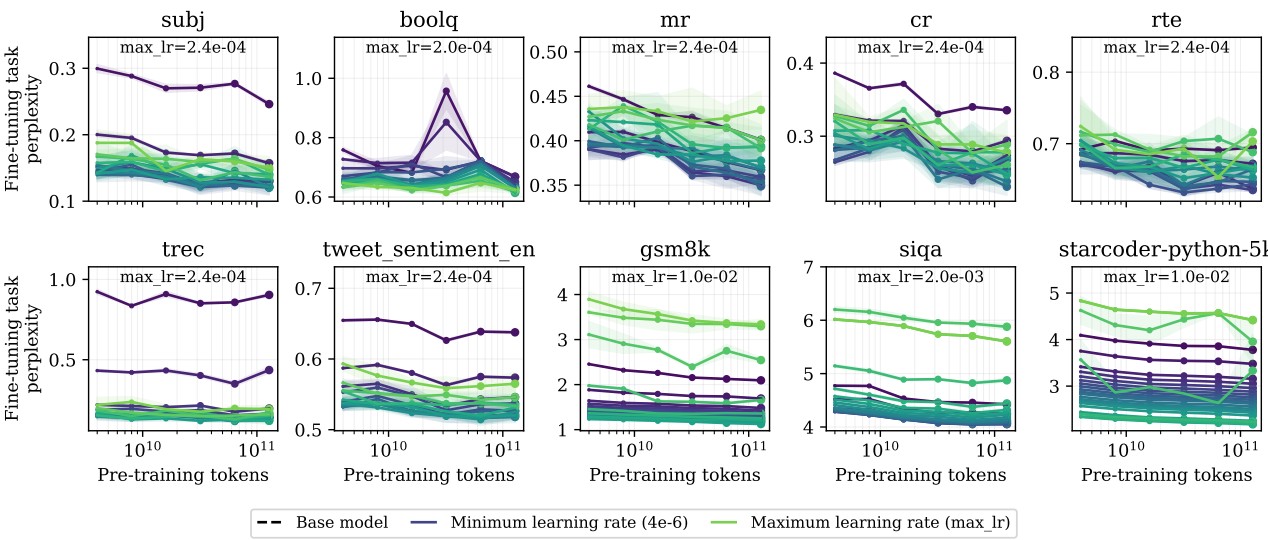

*Figure 61.* **Fine-tuning perplexity after fine-tuning as a function of the pre-training budget using the configuration specified in Table 3 for OLMo-90M.** Each connected line reflects a series of models trained with fixed hyperparameters. Analogous to Figure 5 (bottom) from the main paper.

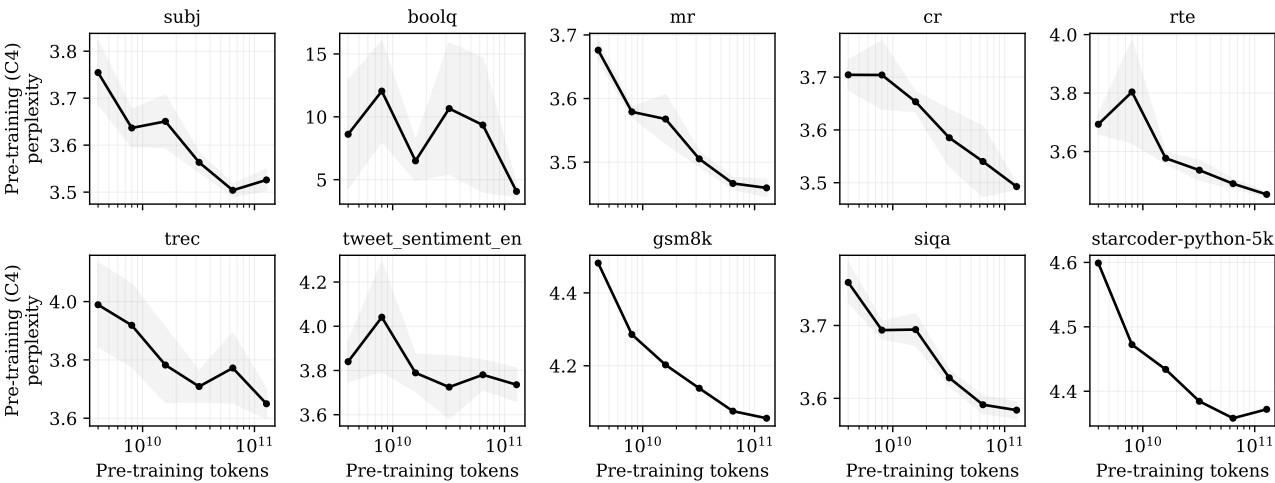

*Figure 62.* **Pre-training perplexity after fine-tuning as a function of the pre-training budget with a tuned learning rate to optimize fine-tuning performance using the configuration specified in Table 3 for OLMo-90M.** Similar to the untuned version but showing the performance with the fine-tuning-optimal learning rate, analogous to Figure 6 (bottom) from the main paper.

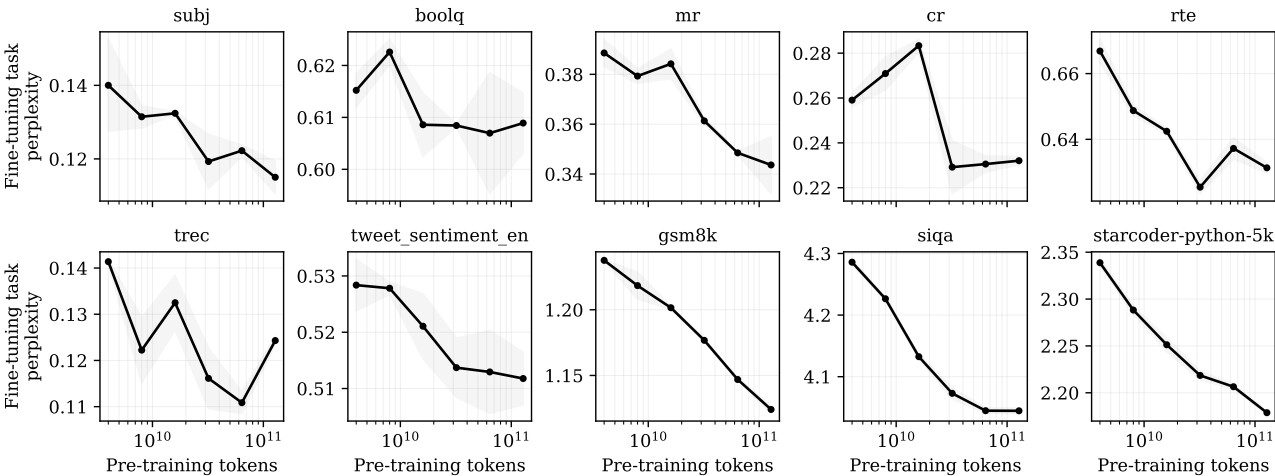

*Figure 63.* **Fine-tuning perplexity after fine-tuning as a function of the pre-training budget with a tuned learning rate to optimize fine-tuning performance using the configuration specified in Table 3 for OLMo-90M.** Similar to the untuned version but showing the performance with the fine-tuning-optimal learning rate, analogous to Figure 6 (top) from the main paper.

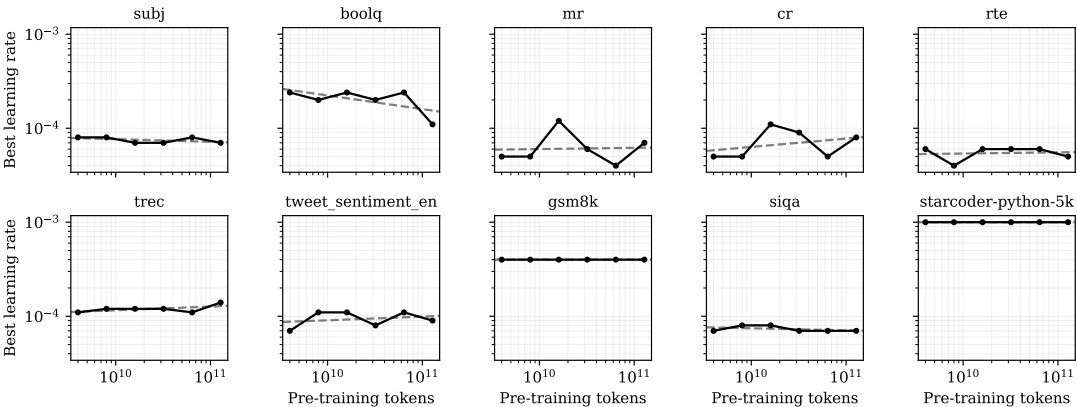

*Figure 64.* **The optimal learning rate for best fine-tuning performance as a function of the pre-training budget using the configuration specified in Table 3 for OLMo-90M.** The learning rate shown corresponds with those chosen in Figures 62 and 63.

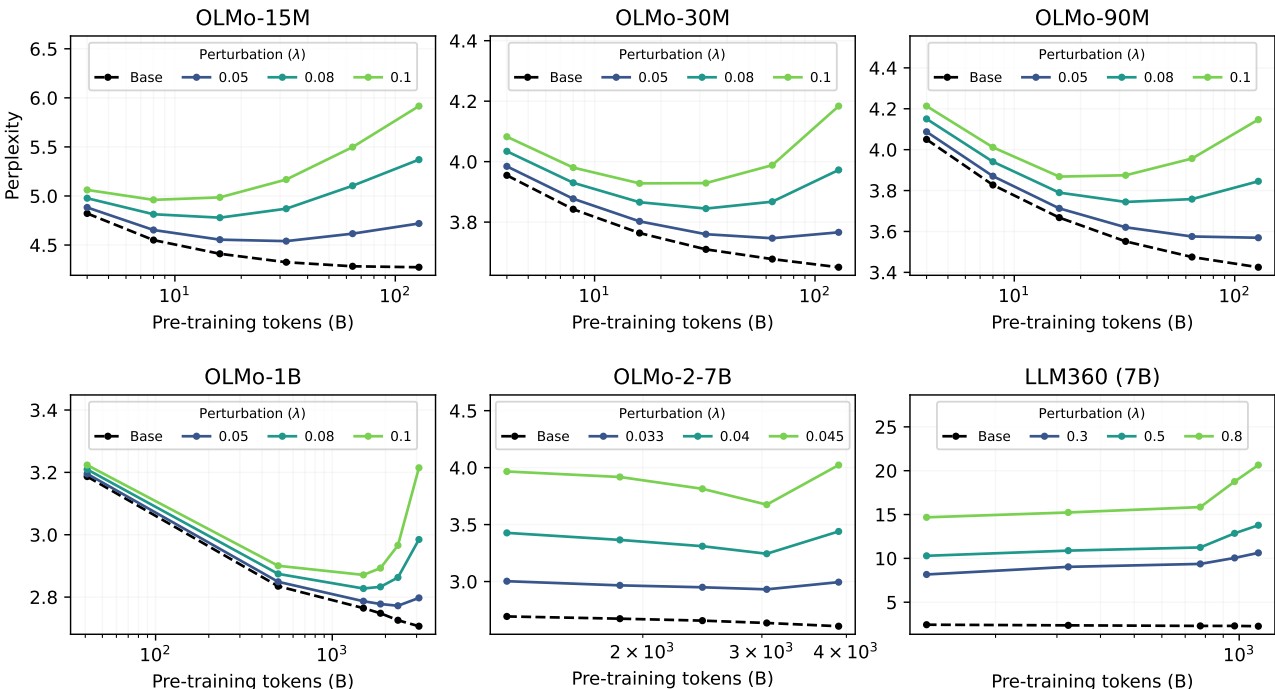

*Figure 65.* **Pre-training perplexity of models with parameters perturbed by Gaussian noise, as a function of the number of pre-training tokens.** We report the C4 web data perplexity of different models where each parameter is perturbed by Gaussian noise scaled by the factor $\lambda$ (color). This figures is an extension of Figure 3 to additional models: OLMo-15M, OLMo-90M, OLMo-1B, OLMo-2-7B, and LLM360-Amber (7B).

### G.3. Extended Gaussian perturbations experiments.

Here, we present extended experiments with Gaussian perturbations on additional models: OLMo-15M, OLMo-90M, OLMo-1B, OLMo-2-7B, and LLM360-Amber (7B). We perturb each parameter by Gaussian noise scaled by the factor $\lambda$. Figure 65 shows the pre-training perplexity of models with parameters perturbed by Gaussian noise as a function of the number of pre-training tokens. Refer to Appendix D for more details on the experimental setup.

