# OpenReview forum: "Overtrained Language Models Are Harder to Fine-Tune"
_ICML.cc/2025/Conference — ICML 2025 poster_

### Official Review · Reviewer_YWVH · 2025-03-12

**Overall Recommendation:** 4

**Summary:**

This paper investigates the phenomenon of "catastrophic overtraining" in language models, where models trained on significantly more tokens than compute-optimal regimes exhibit degraded performance after fine-tuning, despite showing continued improvement in pre-training loss.  The authors present both empirical and theoretical evidence to support this claim.

## update after rebuttal

Satisfied with responses to questions : Rating remains "4: Accept"

**Claims And Evidence:**

* Claim: Overtrained language models are harder to fine-tune.
  + This is well supported by experiments
* Claim: Catastrophic overtraining exists, characterized by thresholds $T_{ft}$ and $T_{pre}$ (fine-tuning and pre-training performance, respectively) degrade when these thresholds are exceeded.
  + $T_{ft}$ well supported by experiments, $T_{pre}$ less so.
* Claim: Theoretical characterization in linear models supports the empirical findings.
  + Given the simplifications involved in the linear models, the theoretical results supply intuitions about the mechanism, rather than directly supporting the observed results

**Essential References Not Discussed:**

N/a

**Experimental Designs Or Analyses:**

* By varying pre-training token budgets and keeping model size constant, the authors effectively isolate the variable of interest (overtraining). Using intermediate checkpoints of pre-trained models and training models from scratch for controlled experiments further strengthens the design.
* Evaluating on both downstream task performance and generalist performance provides a holistic picture of the impact of overtraining. Using multiple diverse benchmarks strengthens the generalizability of the findings.

**Methods And Evaluation Criteria:**

The methods and evaluation criteria are appropriate, comprehensive, and designed to rigorously investigate the $T_{ft}$ research question.  The evaluations for the $T_{pre}$ element are weaker.

**Other Comments Or Suggestions:**

N/a - and I usually find typos...

**Other Strengths And Weaknesses:**

* The paper addresses a highly relevant and timely question in the field of large language models. The finding that overtraining can degrade fine-tuning performance is novel and counter-intuitive, challenging conventional wisdom.
* The claim for $L_{pre}$ (base model capabilities) is a bigger and less definitively backed claim than the claim for $L_{ft}$ (fine-tuning performance).

**Questions For Authors:**

1. Mitigation strategies (Section 6 and future work): Beyond regularization, are there other potential techniques that could mitigate catastrophic overtraining and improve the fine-tuning adaptability of overtrained models? For example, could techniques like parameter resetting, learning rate scheduling adjustments during fine-tuning, or architectural modifications be explored?

2. In the "optimal learning rate trend" analysis (Section 3.4, Figure 5): You categorize datasets based on whether the optimal learning rate is constant, slowly decreasing, or quickly decreasing.  Could you provide further intuition or hypotheses about why certain datasets exhibit each trend? What characteristics of the datasets or fine-tuning tasks might determine the optimal learning rate trend and its relation to overtraining degradation?

**Relation To Broader Scientific Literature:**

* Scaling Laws: The paper investigates a regime that deviates from compute-optimal scaling.
* Overtraining and Generalization: The paper shows that overtraining in pre-training can negatively impact transfer learning ability, which is a novel and significant finding in the current paradigm of large-scale pre-training.

**Theoretical Claims:**

While I have not rigorously checked the proofs line-by-line, the informal statements and the overall theoretical approach seem reasonable.  The theoretical framework provides a valuable conceptual understanding of the overtraining phenomenon.

---

> ### Author Rebuttal · Authors · 2025-03-31
>
> Thank you for your detailed feedback! We are happy to hear your positive comments such as that our methodology is “appropriate, comprehensive, and designed [...] rigorously”, that our paper addresses a problem that is “highly relevant and timely”, and that our results are “novel and counter-intuitive, challenging conventional wisdom”.
>
>
> > Given the simplifications involved in the linear models, the theoretical results supply intuitions about the mechanism, rather than directly supporting the observed results
>
> We aren’t aware of any theoretical tools that would enable us to study the dynamics of LLMs without at least some degree of simplification such as the ones we describe. We fully acknowledge that there are limitations that arise from these simplifications but we hope our analysis is a starting point to understand this phenomenon that may lead to principled mitigation strategies in the future.
>
> Additionally, since catastrophic overtraining is so counterintuitive (to us), seeing it emerge from incremental learning in a simpler setting reassures us it's not just an artifact of our LLM setup and offers intuition for a possible mechanism.
>
> >The claim for $L_{\mathrm{pre}}$ (base model capabilities) is a bigger and less definitively backed claim than the claim for $L_{\mathrm{ft}}$ (fine-tuning performance).
>
> We want clarify our claim regarding how extending pre-training can reduce the base model capabilities:
>
> Our core finding is that there are some important settings where additional pre-training hurts the fine-tuned model’s capabilities, despite helping the base model’s performance. We don’t intend to claim that all settings necessarily exhibit catastrophic overtraining after hyperparameter tuning.
>
> In Figure 2, we find that the fine-tuned model’s “general capabilities” (blue lines) are degraded for PIQA, ARC Challenge, ARC Easy when instruction tuning, and PIQA, ARC Challenge when multimodal tuning. In Figure 3, we find that the fine-tuned model’s “general capabilities” (bottom) degrade for GSM8k, Starcoder-Python, MR, and RTE.
>
> Hopefully this clarifies our evidence that the general capabilities can degrade with overtraining.
>
>
> > Beyond regularization, are there other potential techniques that could mitigate catastrophic overtraining and improve the fine-tuning adaptability of overtrained models? For example, could techniques like parameter resetting, learning rate scheduling adjustments during fine-tuning, or architectural modifications be explored?
>
> Great question! We do explore various learning rate schedules during fine-tuning (constant, constant + warmup, cosine schedules), as detailed in Appendix C.2, and found catastrophic overtraining persisted in each case. Exploring additional mitigation techniques (e.g., parameter resets, architectural modifications) would indeed be valuable future work.
>
>
> > In the "optimal learning rate trend" analysis (Section 3.4, Figure 5): You categorize datasets based on whether the optimal learning rate is constant, slowly decreasing, or quickly decreasing. Could you provide further intuition or hypotheses about why certain datasets exhibit each trend? What characteristics of the datasets or fine-tuning tasks might determine the optimal learning rate trend and its relation to overtraining degradation?
>
> Great question! While empirical datasets are difficult to analyze precisely, our theoretical results offer helpful intuition:
>
> Theorem 4.3 shows that extending pre-training increases the sensitivity (or forgetting) of pre-trained features during fine-tuning. This sensitivity is larger for tasks whose feature distributions significantly differ from pre-training (i.e., larger eigenvalue differences)—fine-tuning tasks which are much different from the pre-training task are more prone to catastrophic overtraining. In our revised version of the paper, we will clarify this with an appropriately updated theorem statement.
>
> Our intuition is that this theoretical insight translates roughly to practice. Empirically, tasks very dissimilar to pre-training likely require larger changes in model features, and larger learning rates naturally induce larger weight changes (even controlling for steps). We’ve empirically verified that larger learning rates lead to larger weight changes and will add a figure illustrating it clearly. Thus, tasks needing significant feature adaptation typically benefit from larger learning rates and, as a result, exhibit more pronounced catastrophic overtraining.

---

> > ### Comment · Reviewer_YWVH · 2025-04-04
> >
> > Thanks for the answers to my questions.  My rating remains "4: Accept"

---

### Official Review · Reviewer_isiV · 2025-03-13

**Overall Recommendation:** 4

**Summary:**

This work challenges the widely held belief in the field that scaling pre-training robustly improves LM performance. The authors find that increasing token budgets during pre-training can actually lead to suboptimal performance on downstream fine-tuned tasks. They leverage popular open-source models and datasets to provide empirical evidence supporting this claim.

The authors conjecture that the suboptimal fine-tuning performance can be attributed to overly complex features of the pre-training distribution, which are learned toward the end of training. Drawing insights from the transfer learning literature, they provide theoretical evidence for this hypothesis using small models.

Taken together, this work sheds light on how design decisions made during pre-training can have ripple effects on post-training outcomes.

**Claims And Evidence:**

The authors provide compelling empirical evidence supporting their claims. In particular, the studies involving intermediate OLMo checkpoints and newly trained models from scratch are especially persuasive.

**Essential References Not Discussed:**

I am not aware of any essential references that have been excluded from this work.

**Experimental Designs Or Analyses:**

Yes

**Methods And Evaluation Criteria:**

The methods and evaluation criteria suite the research questions.

**Other Comments Or Suggestions:**

During pre-training, it may be difficult to identify precisely when overtraining has occurred. One approach practitioners might adopt is applying post-training to intermediate checkpoints, similarly to the OLMo experiments. However, repeated post-training runs might be infeasible or prohibitively expensive. A helpful step the authors could take to operationalize their findings would be to suggest heuristics based on model weights or activations. These heuristics could help practitioners predict whether training should halt due to overtraining, without needing to rely on repeated post-training experiments.

In Section 6, the authors discuss their rationale for not conducting any staged pre-training (also known as annealing or mid-training) [1, 2]. It would be valuable to further expand on this point in the discussion or include additional experiments in the appendix that explore scenarios involving additional domain-oriented pre-training. It is possible that concluding pre-training with domain-specific data might mitigate much of the fine-tuning performance penalty associated with overtraining.

[1] - OLMo, Team et al. “2 OLMo 2 Furious.” ArXiv abs/2501.00656 (2024): n. pag.

[2] - Dubey, Abhimanyu et al. “The Llama 3 Herd of Models.” ArXiv abs/2407.21783 (2024): n. pag.

**Other Strengths And Weaknesses:**

NA

**Questions For Authors:**

NA

**Relation To Broader Scientific Literature:**

This paper contributes to the literature on LM pre-training dynamics. There is growing interest in better understanding how decisions made during pre-training affect final model behavior after post-training. Through the lens of fine-tuning performance, this work suggests that overtrained base models may become more difficult to fine-tune for downstream tasks.

This paper relates to the literature examining the degree to which post-training robustly modifies LM behavior [1]. The authors' contributions suggest that the robustness of post-training behavior may be improved by mitigating overtraining.

[1] - Qi, X., Panda, A., Lyu, K., Ma, X., Roy, S., Beirami, A., Mittal, P., & Henderson, P. (2024). Safety Alignment Should Be Made More Than Just a Few Tokens Deep. ArXiv, abs/2406.05946.

**Theoretical Claims:**

Yes

---

> ### Author Rebuttal · Authors · 2025-03-31
>
> Thank you for your detailed feedback! We are happy to hear your positive comments that there is “compelling empirical evidence”, that “the studies involving intermediate OLMo checkpoints and newly trained models from scratch are especially persuasive”.
>
>
> > During pre-training, it may be difficult to identify precisely when overtraining has occurred.  [...] A helpful step the authors could take to operationalize their findings would be to suggest heuristics based on model weights or activations.
>
> Great suggestion! Developing heuristics based on model weights or activations is indeed a valuable direction for future work. Another promising avenue would be creating scaling laws specifically for catastrophic overtraining. With accurate scaling laws, practitioners could significantly reduce computational overhead by performing fewer post-training experiments to identify optimal training durations.
>
>
> > In Section 6, the authors discuss their rationale for not conducting any staged pre-training (also known as annealing or mid-training) [1, 2]. It would be valuable to further expand on this point in the discussion or include additional experiments in the appendix that explore scenarios involving additional domain-oriented pre-training. It is possible that concluding pre-training with domain-specific data might mitigate much of the fine-tuning performance penalty associated with overtraining.
>
> This is a great point. In our current experiments, our smaller-scale models are pre-trained on the C4 dataset (which includes some code), and we demonstrate catastrophic overtraining on Starcoder-Python fine-tuning (Figures 2, 3). This suggests domain-oriented data doesn't completely eliminate the problem.
>
> However, systematically understanding how the choice of pre-training distribution affects catastrophic overtraining remains important future work. Still, we believe adding domain-specific data cannot universally solve this issue: it’s practically impossible to include data from all potential fine-tuning domains. For example, catastrophic overtraining even arises in multimodal fine-tuning scenarios, where adding image data during pre-training (to a text-only model) simply isn't possible.

---

### Official Review · Reviewer_fdKm · 2025-03-13

**Overall Recommendation:** 3

**Summary:**

The authors study a phenomenon they observe where the more overtrained a pretrained language model is (as a function of pretraining tokens per parameter, w.r.t. training-compute optimal amounts), the more difficult it is to fine-tune the model. The study is motivated by an initial example for OLMo-1B, where the authors tune the SFT LR to optimize quality for a primary task (e.g. AlpacaEval) and use this LR rate to SFT other tasks (e.g. ARC, BoolQ). When doing this, they see that the strongest performance on all tasks do not come from the final overtrained checkpoint, but instead come from earlier checkpoints, calling into question the value of overtraining in the SFT regime. The authors recreate this behavior in smaller scale controlled settings (to up 90M model, 128B tokens) and observe similar behavior, across a large sweep of learning rates, schedules, and batch size. Finally, the authors provide a theoretical explanation for the phenomenon proved for a 2 layer linear network.

**Claims And Evidence:**

The claims in the paper are clear. This paper makes a relatively bold and counter-intuitive claim: that pretraining more can make SFT harder (or in some cases hurt SFT), and the paper provides a wealth of empirical results that provide significant insights into the interaction between overtraining and SFT. The findings here are interesting and would be of interest to the community.

A criticism of the work might be that the author's claim is too broad and underspecified, as to what "Harder" means. I believe while there's significant insight in this paper, the framing of the work suggests that overtraining might be bad as it can hurt post-training -- but this broad claim is very difficult to exhaustively verify. It is also unclear to what level practitioners already expect to have to tune basic SFT hyperparameters to maximize performance given a fixed pretrained model. Whether this is "harder" than what's expected is ill defined. This paper would in general benefit from a more straightforward framing of the results.

An example: In Figure 1, we see that the optimal LR was selected based on AlpacaEval -- which truly benefits from less pretraining under the experimental settings of this paper, at OLMo-1B scale. However, in Figure 7 (Appendix) we get to see the full LR sweep of each task. If the authors decided to pick a different reference task to optimize LR over, the model would have looked significantly more predictable to fine-tune with more pretraining tokens. The choice to optimize AlpacaEval, on one hand is motivated as general, but ultimately a bit arbitrary -- especially as it uses an imperfect autorater (and maybe not tuned well for 1B model outputs).

Similarly, in Figure 9, we see that this difficult to SFT behavior doesn't quite exist for LLM360-7B as the model is less overtrained. However we don't quite see the same AlpacaEval-based result for OLMo-7B to prove this hypothesis. Is it because of the overtraining, or does this phenomena improve as we scale to 7B, for these particular datasets?

Finally I particularly have issues with how the authors sometime conflate SFT for instruction tuning, SFT-until convergence over a single task, and general "post-training" -- especially as the intention and training dynamics for these vary widely (post-training typically also includes RL these days.)

**Essential References Not Discussed:**

A relevant work might be https://openreview.net/pdf?id=vPOMTkmSiu.

**Experimental Designs Or Analyses:**

Yes, see "Claims And Evidence" and "Methods And Evaluation Criteria" for particular cases I have issues with. Aside from these, the authors do indeed do a very thorough job of providing exhaustive empirical results and provide nice analysis of the results. My issues with this work mostly lay in the claims made about the results, given the experimental settings, rather than the results themselves.

**Methods And Evaluation Criteria:**

The methods and evaluation are insightful as we get to see the full behavior of the model and how pretraining and finetuning interact over a wide set of hyperparameters. In some sense, given these empirical results alone -- the reader could come up with their own valuable conclusions as to what the relationship between overtraining and SFT is.

Some of the methodology is contentious: Figure 1 selects a checkpoint based on IT performance, with the expectation that it generalizes to SFT-until-convergence settings. It's unclear how valid this is as the intent of the two types of SFT are quite different (IT tries to train to generalize over many topics, SFT on a single task tries to maximize just that task -- in both cases it is typical to assume that they would need different LR / hypers.)

In Figure 3, the controlled experiment fixes this issue by focusing on single-task settings. However the "harder to SFT" claim is still hard to verify: for most of the tasks, there does indeed exist an optimal hyper-parameter that maximizes performance, and is predictable with more pretraining. There are a few where this is not true, and this is interesting -- however it is unclear if this implies that overtraining makes the model harder to SFT in general. Figure 4 basically shows this, and if theres a widely accepted optimal learning rate per-task, I'm not sure how much "harder" this is. This seems quite reasonable and standard.

**Other Comments Or Suggestions:**

N/A

**Other Strengths And Weaknesses:**

Other strengths:
- The paper is well written free of mistakes. It is easy to follow and read, although including more pertinent information from the Appendix in the main text would improve the writing and let the reader assess the claims easier.

Other weaknesses:
- The theoretical argument is quite limited (two-layer linear model), and is unclear what value it adds to the work. The space could be more useful to provide more detailed discussion of the results, methodology, and relationship to scaling model size.

**Questions For Authors:**

How would you define what it means to be harder to fine-tune, and what is the main evidence in your paper that supports this specifically?

**Relation To Broader Scientific Literature:**

The authors provide thorough connections to related work. So far, no previous work has found that downstream performance could be harmed by overtraining, although there have been works that show that pretraining evaluations can improve with overtraining and in particular Gadre et al. 2024 show that a scaling does exist between overtraining and downstream performance, but this is more aggregated across a larger set of models and tasks.

**Theoretical Claims:**

I followed the theoretical argument, but did not check the correctness of the proofs.

---

> ### Author Rebuttal · Authors · 2025-03-31
>
> Thank you for your detailed feedback! We are happy to hear your positive comments that there is “significant insight in this paper”, that it “would be of interest to the community”, and that we include a “wealth of empirical results”.
>
> We were also glad to read:
>
> > My issues with this work mostly lay in the claims made about the results [...] rather than the results themselves.
>
> To address this, we’ve revised our framing to clarify our claims. These changes will be reflected in the final version.
>
> Summary of changes: To address your main concerns, we will: (1) rewrite to clarify the definition of “harder” and the scope of our findings; (2) further highlight how checkpoint selection (tuning) affects results; (3) make specific changes based on the comments below.
>
>
> > How would you define what it means to be harder to fine-tune, and what is the main evidence in your paper that supports this specifically?
>
> We will clarify in the updated manuscript—by "harder" we mean specifically that extending the pre-training phase, and then fine-tuning, can lead to worse performance in-distribution (fine-tuning task) and also out-of-distribution (unrelated tasks) compared to a shorter pre-training stage.
>
> Our evidence:
> * After instruction tuning OLMo-1B on Anthropic-HH, models trained on 3T tokens underperform those trained on 2.3T on AlpacaEval (in-distribution) and ARC/PIQA (out-of-distribution) — see Figure 1.
> * After multimodal tuning, the 3T-token model also performs worse on ARC and PIQA (Figure 1).
> * After instruction tuning on TULU, the 3T-token model scores lower on AlpacaEval (in-distribution) and on multiple OOD benchmarks (ARC, PIQA, Winogrande, HellaSwag, OpenBookQA) — see Appendix Figure 7 (right).
> * For OLMo-30M, models trained beyond ~32B tokens show degraded in-distribution performance on several fine-tuning tasks (RTE, TREC) and degraded C4 web-data performance when fine-tuning with GSM8K, Starcoder-Python, MR, RTE — see Figure 2.
>
> Apologies for the confusion, we will clear this up in our revision.
>
>
> > This broad claim [overtraining => harder to fine-tune] is very difficult to exhaustively verify
>
> Thanks for bringing this up—in our revised version we will properly clarify the scope of our paper.
>
> To summarize: Our core finding is that there are some important settings where additional pre-training hurts fine-tuned performance, despite helping the base model performance (Figures 1 and 2). We believe that finding *any* example of this phenomenon is surprising. We do not claim that *all* settings exhibit this degradation with overtraining.
>
>
>
> > If the authors decided to pick a different reference task to optimize LR over [other than AlpacaEval], the model would have looked significantly more predictable to fine-tune with more pretraining tokens.
>
> Agreed! A key contribution of our work is precisely to analyze how the optimal learning rate chosen for one task impacts observed degradation on others — this is the focus of Section 3.4.
>
> In this section, we argue that we specifically observe performance (ID or OOD) degradation when optimizing for a task that requires a larger learning rate for good performance — empirically, this is the case for AlpacaEval, and many other evaluation settings (see response below).
>
>
> > The choice to optimize AlpacaEval, on one hand is motivated as general, but ultimately a bit arbitrary.
>
> For instruction tuning, we pick AlpacaEval because it is a standard benchmark to measure response quality—improving this is our goal during fine-tuning. We agree that any single metric could seem arbitrary, but we also evaluate on a collection of other evaluation metrics:
>
> We consider 12 total evaluations across various settings:
> * AlpacaEval (for instruction-tuned models)
> * VLM score (average over 5 VLM benchmarks; for multimodal tuned models)
> * 10 individual fine-tuning datasets where we optimize for ID performance: SUBJ, BoolQ, MR, CR, RTE, TREC, Tweet sentiment, GSM8k, SIQA, Starcoder-Python (Figure 2, plus more in Appendix C.2)
>
> We believe that, collectively, these experiments robustly illustrate the occurrence of degradation across many practical fine-tuning scenarios.
>
>
> > This difficult to SFT behavior doesn't quite exist for LLM360-7B [and] OLMo-7B [...]. Is it because of the overtraining?
>
> Great question! This difference is likely explained by how many tokens per parameter each model has seen:
> * OLMo-1B (3T tokens ≈ 3000 tokens per param) experiences degradation by ~2.3T tokens.
> * LLM360-7B (1.3T tokens ≈ 185 tokens per param) and OLMo-7B (4T tokens ≈ 571 tokens per param) have seen significantly fewer tokens per parameter relative to OLMo-1B.
>
> If degradation scales linearly with model size, we would only expect degradation for a 7B model after ~16T tokens (7B × 2300 tokens per param). Thus, current 7B models simply haven't been trained long enough to reach that regime yet.

---

> > ### Comment · Reviewer_fdKm · 2025-04-04
> >
> > Thank you authors for the clarifications and openness in adjusting the framing and claims of the work. I believe if incorporated your proposed changes will strengthen the work and provide more clarity to the reader. I have increased my rating accordingly.

---

> > > ### Author Response · Authors · 2025-04-08
> > >
> > > Thank you for the quick update! Sorry to follow up, we just wanted to check what improvements could turn this into an "4: Accept" recommendation? We would love to get feedback to improve our work going forward!
> > >
> > > So far it looks like the review indicated that we propose a "relatively bold and counter-intuitive claim" that is justified by a "wealth of empirical results", “provide[s] significant insights”, and is “of interest to the community”, and the main weakness was the scoping of our claims, which we believe we have resolved—is there anything else we can improve along this front?
> > >
> > > ------
> > >
> > > Many of your questions revolved around the high level question “when does catastrophic overtraining occur?"—also, you were wondering how our theory fits into our story.
> > >
> > > We’ve updated our theory section to answer a question that is challenging to answer precisely empirically: which attributes of the fine-tuning (and pre-training) data determine whether or not we observe catastrophic overtraining, and its severity?
> > >
> > > In particular, we’ve updated our theorem statements to make precise exact conditions on the pre-training and fine-tuning distributions for which catastrophic overtraining occurs. We show that degradation from catastrophic overtraining begins to occur earlier in pre-training, and also to a greater extent, when the eigenvalues of the pre-training and fine-tuning tasks differ more substantially.
> > >
> > > Our intuition is that this theoretical insight translates roughly to practice. Empirically, tasks very dissimilar to the pre-training distribution likely require larger changes in model features, and therefore will exhibit faster degradation with overtraining.
> > >
> > > (This is based on comments from Reviewer YWVH, so please refer to our conversation there as well.)
> > >
> > > ------
> > >
> > > Again, we really appreciate you taking the time to write a thoughtful review—our paper has already benefited substantially from it—so we apologize for bothering you about this.

---

### Official Review · Reviewer_gBdh · 2025-03-14

**Overall Recommendation:** 4

**Summary:**

The paper demonstrates how overtraining language models (training on more than the compute-optimal number of tokens) affects their ability to be fine-tuned on new data. For example, the authors perform experiments on OLMo-1B models and show that models pretrained on 3T tokens performed 3% worse on AlpacaEval and 2% worse on ARC reasoning after instruction fine-tuning compared to models trained on 2.3T tokens. Similar degradation is observed over several other evals (PIQA, BoolQ, Winogrande) and fine-tuning setups ((instruction tuning on TULU and multimodal fine-tuning). They perform further controlled experiments on 15M-90M parameter models and provide a theoretical linear model to characterize "catastrophic overtraining," where fine-tuning performance first improves and then degrades with excessive pre-training.

**Claims And Evidence:**

- Experiments on 1B-7B parameter models and two fine-tuning tasks (Anthropic-HH SL on instruction-tuning data, multimodal fine-tuning) support the main finding that overtrained language models can perform worse after fine-tuning.
- The authors provide a theoretical model that helps explain the observed phenomenon in a simplified setting.
- The paper investigates how the learning rate during fine-tuning affects the degradation, showing that different learning rates are optimal for models with different pre-training durations.
- To eliminate the confounder of the learning rate schedule being at different points for the different OLMo model checkpoints, they perform controlled experiments and train models from scratch for various pre-training budgets with a cosine annealing schedule.

However, while the authors describe "catastrophic overtraining" as universal, the evidence is limited to only two fine-tuning setups. Both fail to produce consistent improvements in the evals measured compared to the base model, which could indicate issues in the choices of fine-tuning data mix.

**Essential References Not Discussed:**

N/A

**Experimental Designs Or Analyses:**

The experimental design is generally sound although I have some concern about the choices of fine-tuning datasets as mentioned in the other sections.

**Methods And Evaluation Criteria:**

Overall the methods and benchmark datasets used make sense. However, the use of perplexity in the small pretraining experiments might be misleading (i.e. the results in Figure 2). For downstream tasks like GSM8K, we don't care about perplexity on the answer but rather accuracy. Because it may be hard to get non-trivial accuracy in small models, you could look at average per-token prediction accuracy instead of loss/perplexity. This would help decouple token prediction calibration/effective temperature from downstream task performance.

**Other Comments Or Suggestions:**

N/A

**Other Strengths And Weaknesses:**

N/A

**Questions For Authors:**

1. I am confused by the results in Figure 1. It looks like for most evaluations, fine-tuning hurts instead of helps performance on the eval measured (solid line is below dotted line). Possibly the fine-tuning datasets used are too narrow and would benefit from additional regularization (e.g. by mixing in a wider range of diverse examples from other distributions). Do you have empirical evidence showing that your results generalize to settings where the fine-tuning datamix robustly improves performance on the target evals beyond the capability of the pretrained model?
2. For your perplexity-based experiments, do your findings still hold if you measure token prediction accuracy instead of perplexity?

**Relation To Broader Scientific Literature:**

The paper connects to:
- Research on pretraining scaling laws (Hoffmann et al., 2022; Kaplan et al., 2020) that established optimal token-per-parameter ratios
- Research on model plasticity, particularly studies in reinforcement learning that showed models can become less adaptable after extensive training (Kumar et al., 2020; Lyle et al., 2022) and work on feature rank and inactivity as mechanisms for reduced adaptability (Gulcehre et al., 2022; Dohare et al., 2021)

**Theoretical Claims:**

The theoretical model described in section 4 looks sound. However, the simplification makes a number of assumptions that may not hold for LLMs (e.g. linearity, assuming the pretraining and fine-tuning datasets share the same singular vectors, not accounting for discrete gradient descent algorithm).

---

> ### Author Rebuttal · Authors · 2025-03-31
>
> Thank you for your detailed feedback! We are happy to hear your positive comments that our experiments “support the main finding” and that our theory "helps explain the observed phenomenon”.
>
> Summary of changes: To address your main concerns: (1) we clarify that our fine-tuning setups do lead to consistent improvements on the target task; (2) we will rewrite to clarify the scope of our findings; (3) we will add the per-token accuracy experiment you propose; (4) we will make specific changes as outlined below.
>
> Please find our responses to each concern:
>
>
> > The authors describe "catastrophic overtraining" as universal
>
> Thanks for bringing this up—in our revised version we will clarify the scope and remove the use of “universal”.
>
> To summarize: Our core finding is that there are some important settings where additional pre-training hurts fine-tuned performance, despite helping the base model (Figures 1 and 2). We believe that finding *any* example of this is surprising. We do not claim that *all* settings exhibit this degradation with overtraining.
>
>
> > The evidence is limited to only two fine-tuning setups
>
> We want to clarify—we consider more than just two fine-tuning settings.
>
> For the our large model results (OLMo-1B):
> * Anthropic-HH (Instruction tuning)
> * Multimodal tuning
> * TULU (instruction tuning)
> * Alpaca (instruction tuning)
>
> (Figure 1 and Appendix B). We observe catastrophic overtraining for Anthropic-HH, multimodal tuning, and TULU.
>
> For the small model results (OLMo-30M):
> * GSM8k
> * SIQA
> * Starcoder-Python
> * MR
> * RTE
> * TREC
> * CR
> * Tweet sentiment
> * SUBJ
> * BoolQ
>
> (Figures 2 & 3, plus Appendix C).
>
>
> > Both [IFT & VLM fine-tuning setups] fail to produce consistent improvements in the evals measured compared to the base model.
>
> For Anthropic-HH, we aim to improve response quality (measured via AlpacaEval), and fine-tuning boosts scores by ~35–45% over the base model (see table below).
>
> For VLM fine-tuning, the goal is to enable image input via an adapter—something the base model can’t do. The fine-tuned model reaches ~45% VLM score (Figure 1).
>
> We also evaluate OOD tasks (ARC, PIQA, BoolQ, Winogrande, HellaSwag, and OpenBookQA). Fine-tuning on an unrelated task isn’t expected to help performance on these tasks and can often harm performance [1].
>
> Together, these evaluations help us assess how overtraining impacts the ability to learn new tasks (measured by main eval) and the tendency to degrade unrelated capabilities (measured by OOD eval).
>
>
> Here are the base model (and instruction-fine-tuned) scores for AlpacaEval:
>
> | Tokens | Base model AlpacaEval (%) | Instruction-tuned AlpacaEval (%) |
> |---|-|---|
> | 0.5e12      | 10.57                 | **47.10**                            |
> | 1.5e12      | 14.43                 | **54.97**                            |
> | 1.8e12     | 11.94                 | **56.03**                            |
> | 2.3e12      | 12.39                 | **56.05**                            |
> | 3e12      | 11.07                 | **53.52**                            |
>
> Table: Comparing AlpacaEval for OLMo-1B base model vs instruction tuning with Anthropic-HH. We will add this to the main paper to clarify, apologies for the confusion.
>
>
> > Do your findings still hold if you measure token prediction accuracy instead of perplexity?
>
> We ran this experiment, and yes, our findings hold. Below report per-token accuracy of GSM8k as a function of pre-training tokens (analogous to Figure 3).
>
> | Pre-training Tokens (B) | Pre-training data per-token accuracy (%) | GSM8k per-token accuracy (%) |
> |---|----|---|
> | 4 | 15.4     | 65.9    |
> | 8  | 15.8  | 67.2    |
> | 16  | 16.7    | 67.7    |
> | 32 | 16.4  | 68.0  |
> | 64   | 15.4  | 67.8 |
> | 128  | 13.4   | 67.1   |
>
> Note: GSM8k per-token accuracy is more predictable than web data because the examples are quite structured.
>
> Results corresponding to other figures & datasets similarly hold true.
>
>
> > The simplification [of the theoretical model] makes a number of assumptions that may not hold for LLMs.
>
> To clarify—We do model discrete gradient descent explicitly during fine-tuning (see Appendix A.3).
>
> We aren’t aware of any theoretical tools that would enable us to study the dynamics of LLMs without at least some degree of simplification such as the ones we describe. We fully acknowledge that there are limitations that arise from these simplifications but we hope our analysis is a starting point to understand this phenomenon that may lead to principled mitigation strategies in the future.
>
> Additionally, since catastrophic overtraining is so counterintuitive (to us), seeing it emerge from incremental learning in a simpler setting reassures us it's not just an artifact of our LLM setup and offers intuition for a possible mechanism.
>
> [1] Goodfellow, Ian J., et al. An empirical investigation of catastrophic forgetting in gradient-based neural networks. 2013.

---

> > ### Comment · Reviewer_gBdh · 2025-04-05
> >
> > Thank you for the additional information, this clarifies my understanding. I will increase my score.

---

### Decision · Program_Chairs · 2025-05-01

**Decision:**

Accept (poster)

**Comment:**

Summary: This paper studies how overtraining LLMs impacts their fine-tuning effectiveness. Through extensive analysis of small scale models, the authors find that pretraining on too many tokens can degrade downstream fine-tuning performance, even as pretraining loss continues to improve. The authors call this as “catastrophic overtraining” and support their claim with both experiments and theoretical framework.

Review summary:
All reviewers agree the paper presents an important insight into LLM training dynamics. The paper provides extensive empirical evidence across model sizes and tasks that overtraining can degrade fine-tuning performance.

Recommendation:
I recommend acceptance.